# DRIFT2MATRIX: KERNEL-INDUCED SELF REPRESENTATION FOR CONCEPT DRIFT ADAPTATION IN CO-EVOLVING TIME SERIES

## ABSTRACT

In the realm of time series analysis, tackling the phenomenon of concept drift poses a significant challenge. Concept drift – characterized by the evolving statistical properties of time series data, affects the reliability and accuracy of conventional analysis models. This is particularly evident in co-evolving scenarios where interactions among variables are crucial. This paper presents Drift2Matrix, a novel framework that leverages kernel-induced self-representation for adaptive responses to concept drift in time series. Drift2Matrix employs a kernel-based learning mechanism to generate a representation matrix, encapsulating the inherent dynamics of co-evolving time series. This matrix serves as a key tool for identification and adaptation to concept drift by observing its temporal variations. Furthermore, Drift2Matrix effectively identifies prevailing patterns and offers insights into emerging trends through pattern evolution analysis. Our empirical evaluation of Drift2Matrix across various datasets demonstrates its effectiveness in handling the complexities of concept drift. This approach introduces a novel perspective in the theoretical domain of co-evolving time series analysis, enhancing adaptability and accuracy in the face of dynamic data environments. Code is available at GitHub[1].

## 1 INTRODUCTION

Co-evolving time series data analysis plays a crucial role in diverse sectors including finance, healthcare, and meteorology. Within these areas, multiple time series evolve simultaneously and interact with one another, forming complex, dynamic systems. The evolving statistical properties of such data present significant analytical challenges. A particularly pervasive issue is concept drift Lu et al. (2018b); Yu et al. (2024), which refers to shifts in the underlying data distribution over time, thereby undermining the effectiveness of static models. Miyaguchi & Kajino (2019); You et al. (2021).

Traditional time series approaches commonly rely on the assumptions of stationarity and linear relationships. Methods such as ARIMA and VAR Box (2013), for instance, perform well in circumstances with stable and predictable dynamics. However, their effectiveness decreases when dealing with non-stationary data, particularly in the presence of concept drift. Conversely, machine learning methodologies Li et al. (2022); Wen et al. (2020), such as diverse neural network architectures Ho et al. (2022); Li et al. (2023); Yang et al. (2024), offer more flexibility but often require large amounts of data and face difficulties in terms of interpretability and adaptability, especially in dynamic contexts.

The evolving study has steered the field towards more adaptive and dynamic models. Methods like change point detection Deldari et al. (2021); Liu et al. (2023) and online learning algorithms Huang et al. (2022); Zhang et al. (2024) are designed to detect shifts in patterns. Nonetheless, these methods are typically restricted to detecting structural breaks or focusing on univariate series, rather than tracking and predicting subtle, ongoing changes in concepts, which limits their applicability in real-world co-evolving time series. In the complex environments, where multiple time series evolve and interact simultaneously, capturing the nonlinear relationships among variables is criticalMarcotte et al. (2023); Bayram et al. (2022). Despite recent advancements Matsubara & Sakurai (2019); Li

---

[1] https://anonymous.4open.science/r/Drift2Matrix-main-86B7

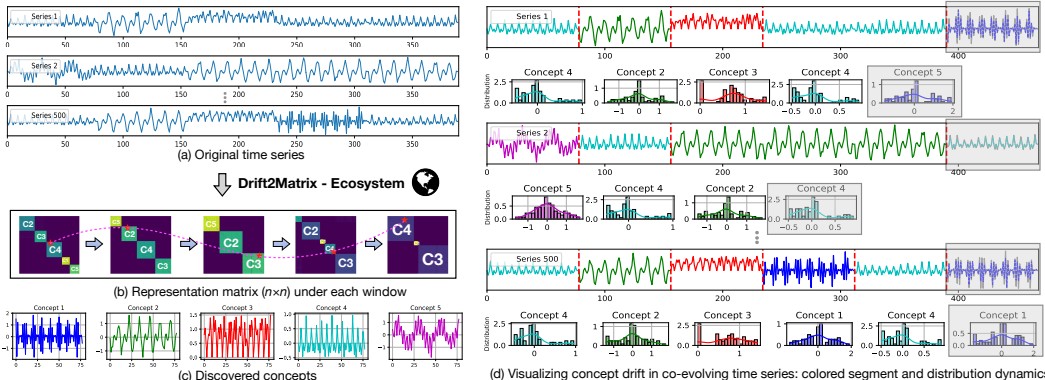

Figure 1: Modeling power of Drift2Matrix for co-evolving time series: Drift2Matrix treats the time series as an ecosystem - 🌐, and automatically identifies, tracks, and predicts dynamic concepts without prior knowledge. (a) Original time series; (b) Drift2Matrix-generated representation matrix with distinct concepts (C1-C5) in block diagonal form. The red star marks S1, and purple dashed lines trace S1's concept drift over time; (c) Identified concepts within the series; (d) Concept drift (red dashed lines) and forecasted trends (grey areas).

et al. (2022); Wen et al. (2024), most multivariate models define the concept as a collective behavior of streaming data, falling short in their ability to capture the dynamics of individual series and their interactions. This limitation impairs the models' capability to discover and interpret the complex interdependencies among variables, thereby constraining their effectiveness in scenarios that involve multiple time series, such as city-wide electricity usage or road traffic forecasting. This raises a critical question – ***Can we identify underlying concepts from co-evolving time series and leverage their nonlinear relationships to predict concepts that have not appeared in a single series?***

To tackle these challenges, we introduce Drift2Matrix, a framework designed for the dynamic complexities of co-evolving time series data. Drift2Matrix employs a kernel-induced self-representation method, adept at capturing the intricate interdependencies and the evolving natures of such data. Our approach, which transforms time series into a matrix format capable of adapting to concept drift, offers a robust and flexible solution for co-evolving time series analysis amidst non-linear interactions and shifting distributions.

**Preview of Our Results.** Fig. 1 showcases our Drift2Matrix results to a 500-dimensional ($n = 500$) co-evolving time series (Fig. 1 (a)). Treating the dataset as an interconnected ecosystem 🌐, Drift2Matrix allows for comprehensive analysis through three key objectives: identifying concepts, tracking their drift, and forecasting future trend.

*(O1) Concept identification*: Fig. 1 (b) displays the heatmap of Drift2Matrix-generated representation matrix ($n \times n$) over time[2]. This matrix showcases a block diagonal structure, where each block represents a unique concept (e.g., C1 - C5), with brightness in the heatmap indicating series correlation strength. Variations of the block structure over time underscore the dynamic correlations within the time series, indicating the presence and evolution of concepts. An extended analysis of these matrices facilitates the identification of the total number of concepts and their pattern (Fig. 1 (c)).

*(O2) Dynamic concept drift*: Tracking the transitions of time series within the Drift2Matrix-generated matrices provides insights into the trajectory of concept drift. Fig. 1 (d) illustrates the concept drift process over time for each of the 3 example series. Red dashed lines in the figure mark the points where concept drift occurs, exemplified by the transition of Series 1 from Concept 4 to Concept 2 and subsequent shifts. This process can also be observed in the representation matrix shown in Fig. 1 (b), where the red star marks the position of Series 1 (S1), and the purple dashed lines trace how S1 shifts over time, further illustrating the process of tracking its transition between concepts.

*(O3) Forecasting*: The grey areas in Fig. 1 (d) represent the forecasted series distribution and values. These forecasts, denoted by dashed lines matching the colors of the concepts, align closely

---

[2]Drift2Matrix autonomously determines the optimal segments in a domain-agnostic manner.

with the actual series trend (solid grey lines). ***A significant forecast for Series 1 includes the emergence of Concept 1, previously undetected.*** This predictive capability stems from Drift2Matrix's ecosystem perspective of the co-evolving time series, leveraging a probabilistic model of the nonlinear interactions among series to anticipate the emergence and evolution of new concepts.

**Contributions.** Drift2Matrix represents more than a mere solution – it is a paradigm shift in co-evolving time series analysis. This framework introduces a new perspective on modeling and interpreting complex, interrelated co-evolving time series. Drift2Matrix has the following desirable properties:

1. Drift2Matrix introduces a novel, kernel-induced approach for modeling complex interdependencies, enhancing understanding of underlying dynamics, and can be easily integrated into most deep learning backbones.

2. Drift2Matrix is adaptive, with the capability to identify and respond to concept drift autonomously, without prior knowledge about concept.

3. Drift2Matrix transforms concept drift into a matrix optimization problem and enhances interpretability, offering a new perspective into the evolving dynamics of co-evolving time series.

## 2 THE LANDSCAPE OF CONCEPT DRIFT

**Concept Drift.** Concept drift in time series refers to the scenario where the statistical properties of the target variable, or the joint distribution of the input-output pairs, change over time. These drifts primarily exhibit in two manners Ren et al. (2018); Kim et al. (2021): the first is characterized by subtle, ongoing changes, reflecting the evolving dynamics of the time series, while the second arises from sudden shifts caused by structural breaks in the relationships among time series. Both gradual and abrupt changes can significantly disrupt model performance if not detected and adapted to in a timely manner, as they challenge the stability and accuracy of predictive models.

**Challenges in Co-evolving Scenarios.** Recent advances in time series analysis have led to progress in addressing concept drift. Dish-TS Fan et al. (2023) offers a general approach for alleviating distribution shift in time series forecasting by normalizing model inputs and outputs to better handle distribution changes. Similarly, Cogra's application of the Sequential Mean Tracker (SMT) adjusts to changes in data distribution, improving forecast accuracy Miyaguchi & Kajino (2019). Despite these strides, these methodologies exhibit limitations when applied to co-evolving time series, where interdependencies between series introduce additional complexity. In such scenarios, a shift in one variable can propagate through the network of interrelations, affecting the entire system. DDG-DA Li et al. (2022) for data distribution generation has been adapted to better suit co-evolving scenarios, addressing the unique challenges presented by the interplay of multiple data streams under concept drift conditions. However, this method defines the concept as a collective behavior represented by co-evolving time series rather than capturing the dynamics of individual series and their interactions. Notably, even the most recent deep learning methods that mention concept drift, such as OneNet Wen et al. (2024) and FSNet Pham et al. (2022), primarily aim to mitigate the impact of concept drift on forecasting rather than addressing the challenges of adaptive concept identification and dynamic concept drift. They achieve this by incorporating an ensemble of models with diverse data biases or by refining network parameters for better adaptability. Due to the space limit, more related works about concept-drift, representation learning on times series and motivation are left in the Appendix A.

## 3 PRELIMINARIES

**Problem Definition.** Consider a co-evolving time series dataset $\mathbf{S} = S_1, S_2, \ldots, S_N \in \mathbb{R}^{T \times N}$, with $N$ being the number of variables and $T$ represents the total number of time steps. Our goal is to (1) to automatically identify a set of latent concepts $\mathbf{C} = \mathbf{C}_1, \mathbf{C}_2, \ldots, \mathbf{C}_k$, where $k$ represents the total number of distinct concepts; (2) to track the evolution and drift of these concepts across time; and (3) to predict future concepts.

**Concept.** Throughout this paper, a concept is defined as the profile pattern of a cluster of similar subseries, observed within a specific segment/window. Here, the term "profile pattern" refers to a subseries, the vector representation of which aligns with the centroid of similar subseries. We

use a tunable hyperparameter $\rho$, to differentiate profile patterns and modulate whether concept drift is gradual (smaller $\rho$, more concepts) or abrupt (larger $\rho$, fewer concepts). For a more detailed mathematical definition, please refer to Appendix A.

**Notation.** We denote matrices by boldface capital letters, e.g., $\mathbf{M}$. $\mathbf{M}^{\mathrm{T}}$, $\mathbf{M}^{-1}$, $\mathrm{Tr}(\mathbf{M})$ indicate the transpose, inverse and trace of matrix $\mathbf{M}$, respectively. $\mathrm{diag}(\mathbf{M})$ refers to a vector with its $i$-$th$ element being the $i$-$th$ diagonal element of $\mathbf{M}$.

### 3.1 SELF-REPRESENTATION LEARNING IN TIME SERIES

Time series often manifest recurring patterns. One feasible way to capture these inherent patterns is through self-representation learning Bai & Liang (2020). This approach models each series as a linear combination of others, formulated as $\mathbf{S} = \mathbf{SZ}$ or $S_i = \sum_j S_j Z_{ij}$, where $\mathbf{Z}$ is the self-representation coefficient matrix. In multiple time series, high $Z_{ij}$ values indicate similar behaviors or concepts between $S_i$ and $S_j$. The learning objective function is:

$$\min_{\mathbf{Z}} \frac{1}{2}||\mathbf{S} - \mathbf{SZ}||^2 + \Omega(\mathbf{Z}), \ s.t. \ \mathbf{Z} = \mathbf{Z}^{\mathrm{T}} \geq 0, \mathrm{diag}(\mathbf{Z}) = 0 \tag{1}$$

where $\Omega(\cdot)$ is a regularization term on $\mathbf{Z}$. The ideal representation $\mathbf{Z}$ should group data points with similar patterns, represented as block diagonals in $\mathbf{Z}$, each block signifying a specific concept. The number of blocks, $k$, corresponds to the distinct concepts.

### 3.2 KERNEL TRICK FOR MODELING TIME SERIES

Addressing nonlinear relationships in co-evolving time series, especially in the presence of concept drift, can be challenging for linear models. Kernelization techniques overcome this by mapping data into higher-dimensional spaces using suitable kernel functions. This facilitates the identification of concepts within these transformed spaces. The process is facilitated by the "kernel trick", which employs a nonlinear feature mapping, $\Phi(\mathbf{S})$: $\mathcal{R}^d \rightarrow \mathcal{H}$, to project data $\mathbf{S}$ into a kernel Hilbert space $\mathcal{H}$. Direct knowledge of the transformation $\Phi$ is not required; instead, a kernel Gram matrix $\mathcal{K} = \Phi(\mathbf{S})^{\top}\Phi(\mathbf{S})$ is used.

## 4 DRIFT2MATRIX

This section introduces the fundamental concepts and design philosophy of Drift2Matrix. Our objective is to identify significant concept trends and encapsulate them into a succinct yet powerful and adaptive representative model.

### 4.1 KERNEL-INDUCED REPRESENTATION LEARNING

To model concepts, we propose kernel-induced representation learning to cluster subseries retrieved using a sliding window technique. We begin with a simple case, where we treat the entire series as a single window. Given a collection of time series $\mathbf{S} = (S_1, \ldots, S_N) \in \mathcal{R}^{T \times N}$ as described in Eq. 1, its linear self-representation $\mathbf{Z}$ would make the inner product $\mathbf{SZ}$ come close to $\mathbf{S}$. Nevertheless, the objective function in Eq. 1 may not efficiently handle nonlinear relationships inherent in time series. A solution involves employing "kernel tricks" to project the time series into a high-dimensional RKHS. Building upon this kernel mapping, we present a new kernel representation learning strategy, with the ensuing self-representation objective:

$$\min_{\mathbf{Z}} \frac{1}{2}||\Phi(\mathbf{S}) - \frac{\alpha}{2}\Phi(\mathbf{S})\mathbf{Z}||^2 = \min_{\mathbf{Z}} \frac{1}{2}\mathrm{Tr}(\mathcal{K} - \alpha\mathcal{K}\mathbf{Z} + \mathbf{Z}^{\mathrm{T}}\mathcal{K}\mathbf{Z}), \quad s.t. \ \mathbf{Z} = \mathbf{Z}^{\mathrm{T}} \geq 0, \mathrm{diag}(\mathbf{Z}) = 0 \tag{2}$$

Here, the mapping function $\Phi(\cdot)$ needs not be explicitly identified and is typically replaced by a kernel $\mathcal{K}$ subject to $\mathcal{K} = \Phi(\mathbf{S})^{\top}\Phi(\mathbf{S})$. It's noteworthy that the parameter $\alpha$ is key to preserving the local manifold structure of time series during this projection, further explained in Sec. 5.2.

Ideally, we aspire to achieve the matrix $\mathbf{Z}$ having $k$ block diagonals under some proper permutations if time series $\mathbf{S}$ contains $k$ concepts. To this end, we add a regularization term to $\mathbf{Z}$ and define the kernel objective function as:

$$\min_{\mathbf{Z}} \frac{1}{2}\mathrm{Tr}(\mathcal{K} - \alpha\mathcal{K}\mathbf{Z} + \mathbf{Z}^{\mathrm{T}}\mathcal{K}\mathbf{Z}) + \gamma||\mathbf{Z}||_{\boxed{k}}, \quad s.t. \ \mathbf{Z} = \mathbf{Z}^{\mathrm{T}} \geq 0, \mathrm{diag}(\mathbf{Z}) = 0 \tag{3}$$

where $\gamma > 0$ balances the loss function with regularization term, $||\mathbf{Z}||_{\boxed{k}} = \sum_{i=N-k+1}^{N} \lambda_i(\mathbf{L_Z})$ and $\lambda_i(\mathbf{L_Z})$ contains the eigenvalues of Laplacian matrix $\mathbf{L_Z}$ corresponding to $\mathbf{Z}$ in decreasing order. Here, the regularization term is equal to 0 if and only if $\mathbf{Z}$ is $k$-block diagonal (see **Theorem 4.1** for details). Based on the learned high-quality matrix $\mathbf{Z}$ (containing the block diagonal structure), we can easily group the time series into $k$ concepts using traditional spectral clustering technology Ng et al. (2001). The detailed method of estimating the number of concepts $k$ is provided in Appendix B. To solve Eq. 3, which is a nonconvex optimization problem, we leverage the Augmented Lagrange method with Alternating Direction Minimization strategy to propose a specialized method for solving the nonconvex kernel self-representation optimization (see Appendix C.3).

**Theorem 4.1** *If the multiple time series $\mathbf{S}$ contains $k$ distinct concepts, then $\min \sum_{i=N-k+1}^{N} \lambda_i(\mathbf{L_Z})$ is equivalent to $\mathbf{Z}$ being $k$-block diagonal.*

**Proof 4.2** *Due to the fact that $\mathbf{Z} = \mathbf{Z}^{\mathrm{T}} \geq 0$, the corresponding Laplacian matrix $\mathbf{L_Z}$ is positive semidefinite, i.e., $\mathbf{L_Z} \succeq 0$, and thus $\lambda_i(\mathbf{L_Z}) \geq 0$ for all $i$. The optimal solution of $\min \sum_{i=N-k+1}^{N} \lambda_i(\mathbf{L_Z})$ is that all elements of $\lambda_i(\mathbf{L_Z})$ are equal to 0, which means that the $k$ smallest eigenvalues are 0. Combined with the Laplacian matrix property, it's evident that the multiplicity $k$ of the zero eigenvalues in Laplacian matrix $\mathbf{L_Z}$ matches the count of connected components (or blocks) present in $\mathbf{Z}$, and thus the soundness of **Theorem 1** has been proved.*

### 4.2 ADAPTATION TO CONCEPT DRIFT

For $b$ sliding windows $\{W_1, \ldots, W_b\}$[3], Drift2Matrix constructs individual kernel representations for each window. Let's consider $k$ distinct concepts identified across these windows, denoted as $\mathbf{C}_{c \in [1,k]} = \{\mathbf{C}_1, \cdots, \mathbf{C}_k\}$. It is important to know that the concepts identified from the subseries $\mathbf{S}_p$ within the $p$-$th$ ($p \in [1, b]$) window may differ from those in other windows. This reveals the variety of concepts in time series and the demand for a dynamic representation.

For two consecutive windows $W_p$ and $W_{p+1}$, the effective probability of a suddenly switching in concept from $\mathbf{C}_r$ to $\mathbf{C}_m$ can be calculated for series $S_i$ as follows:

$$P(\mathbf{C}r \to \mathbf{C}m | W_p \to W_{p+1}, S_i) = \frac{\sum_{\zeta} \Psi_{p,p+1}^{r,\zeta} \Lambda_{p,p+1}^{\zeta,m}}{\sum_{\zeta_1} \sum_{\zeta_2} \Psi_{p,p+1}^{\zeta_1,\zeta_2} \Lambda_{p,p+1}^{\zeta_1,\zeta_2}} \tag{4}$$

where $\zeta_1, \zeta_2 \in \{1, \cdots, k\}$ and

$$\Psi_{p,p+1}^{r,m} = \frac{\eta(\mathbf{C}_r \to \mathbf{C}_m | \mathcal{T}r(S_i|W_p))}{|\mathcal{T}r(S_i|W_p)|},$$

$$\Lambda_{p,p+1}^{r,m} = \sum_{l=1}^{p-1} \frac{\min\{\eta(\mathbf{C}_r, W_l), \eta(\mathbf{C}_m, W_{l+1})\}}{\max\{\eta(\mathbf{C}_r, W_l), \eta(\mathbf{C}_m, W_{l+1})\}} \tag{5}$$

Here, the trajectory $\mathcal{T}r(S_i|W_p)$ represents the sequence of concepts exhibited by series $S_i$ over time. The term $\eta(\mathbf{C}_r \to \mathbf{C}_m | \mathcal{T}r(S_i|Wp))$ counts the occurrences of the sequence $\mathbf{C}_r, \mathbf{C}_m$ within this trajectory. $\eta(\mathbf{C}_r, W_l)$ (resp. $\eta(\mathbf{C}_m, W_{l+1})$) denotes the number of series exhibiting concept $\mathbf{C}_r$ (resp. $\mathbf{C}_m$) at window $W_l$ (resp. $W_{l+1}$).

Notably, the component $\Psi_{p,p+1}^{r,m}$ gauges the immediate risk of observing concept $\mathbf{C}_m$ after the prior concept $\mathbf{C}r$. Meanwhile, $\Lambda_{p,p+1}^{r,m}$ quantifies the likelihood of transitions between concepts within the entire dataset $\mathbf{S}$. Consequently, Eq. 4 integrates both the immediate risk for a single series and the collective concept of series in $\mathbf{S}$.

In the event that the exhibited concept of $S_i$ in $W_p$ is $\mathbf{C}_r$ and the most probable concept switch goes to one of the concepts $\mathbf{C}_m$, we can estimate the series value based on the previous realized value observed and the concept predicted. The predicted values of $S_i$ under the window $W_{p+1}$ can be calculated as:

$$Pre\_S_i = \sum_{l=1}^{p} \Delta(\mathbf{R}_m | S_i, W_l) \cdot \tau^{p-l+1} \cdot \{S_i | W_l\} \tag{6}$$

---

[3]The window size is determined through a heuristic method that balances the granularity of concept identification with the representational capacity of Drift2Matrix. For details, see Sec. 6.1 and Appendix D.

where the indicator function $\Delta(\mathbf{C}_m | S_i, W_l)$ indicates whether $S_i$ belongs to $\mathbf{C}_m$ under window $W_l$, and $\{S_i | W_l\}$ is the subseries value of $S_i$ within window $W_l$. $\tau^{p-l+1} \in (0, 1)$ is the weight value that modulates the contribution of $\{S_i | W_l\}$ to generate predicted values. In this context, we simply require $\sum_{l=1}^{p} \tau^{p-l+1} = 1$, which implies that the subseries closer to the predicted window is deemed more significant.

### 4.3 Integration into Deep Learning Backbones

One of the key strengths of Drift2Matrix is its flexibility, which allows it to be easily integrated into most modern deep learning backbones. Here, we take Autoencoder-Drift2Matrix (Auto-D2M) as an example, which comprises an Encoder, a Kernel Representation Layer, and a Decoder.

**Encoder**: The encoder maps input $\mathbf{S}$ into a latent representation space. Specifically, the encoder performs a nonlinear transformation $\mathbf{H}_{\Theta_e} = \text{Encoder}_{\Theta_e}(\mathbf{S})$, where $\mathbf{H}_{\Theta_e}$ represents the latent representations.

**Kernel Representation Layer**: Implemented as a fully connected layer without bias and non-linear activations, this layer captures intrinsic relationships among the latent representations and ensures that each latent representation can be expressed as a combination of others $\Phi(\mathbf{H}_{\Theta_e}) = \Phi(\mathbf{H}_{\Theta_e})\Theta_\mathbf{s}$, where $\Theta_\mathbf{s} \in \mathbb{R}^{n \times n}$ is the self-representation coefficient matrix. Each column $\theta_{s,i}$ of $\Theta_\mathbf{s}$ represents the weights used to reconstruct the $i$-th latent representation from all latent representations. To promote sparsity in $\Theta_\mathbf{s}$ and highlight the most significant relationships, we introduce an $\ell_1$ norm regularization: $\mathcal{L}_{\text{kernel}}(\Theta_\mathbf{s}) = \|\Theta_\mathbf{s}\|_1$.

**Decoder**: The decoder reconstructs the input from the refined latent representations $\hat{\mathbf{S}}_{\Theta_d} = \text{Decoder}_{\Theta_d}(\hat{\mathbf{H}}_{\Theta_e})$, where $\hat{\mathbf{S}}_{\Theta_d}$ represents the reconstructed time series segments.

**Loss Function**: Training involves minimizing a loss function that combines reconstruction loss, self-representation regularization, and a temporal smoothness constraint:

$$\mathcal{L}(\Theta) = \frac{1}{2}\|\mathbf{S} - \hat{\mathbf{S}}_{\Theta_d}\|_F^2 + \lambda_1 \|\Theta_\mathbf{s}\|_1 + \lambda_2 \|\Phi(\mathbf{H}_{\Theta_e}) - \Phi(\mathbf{H}_{\Theta_e})\Theta_\mathbf{s}\|_F^2,$$

where $\Theta = \{\Theta_\mathbf{e}, \Theta_\mathbf{s}, \Theta_\mathbf{d}\}$ includes all learnable parameters, with $\lambda_1$, $\lambda_2$, and $\lambda_3$ balancing the different loss components. Specifically, $\lambda_1$ promotes sparsity in the self-representation $\Theta_\mathbf{s}$, and $\lambda_2$ preserves the self-representation property.

## 5 Theoretical Analysis

### 5.1 Behavior of the Representation Matrix

The core of Drift2Matrix is the kernel representation matrix $\mathbf{Z}$, which encapsulates the relationships and concepts within time series. Without loss of generality, let $\mathbf{S} = [\mathbf{S}^{(1)}, \mathbf{S}^{(2)}, \cdots, \mathbf{S}^{(k)}]$ be ordered according to their concept. Ideally, we wish to obtain a representation $\mathbf{Z}$ such that each point is represented as a combination of points belonging to the same concept, i.e., $\mathbf{S}^{(i)} = \mathbf{S}^{(i)}\mathbf{Z}^{(i)}$. In this case, $\mathbf{Z}$ in Eq. 1 has the $k$-block diagonal structure (up to permutations), i.e.,

$$\mathbf{Z} = \begin{bmatrix} \mathbf{Z}^{(1)} & 0 & \cdots & 0 \\ 0 & \mathbf{Z}^{(2)} & \cdots & 0 \\ \vdots & \vdots & \ddots & \vdots \\ 0 & 0 & \cdots & \mathbf{Z}^{(k)} \end{bmatrix} \tag{7}$$

This representation reveals the underlying structure of $\mathbf{S}$, with each block $\mathbf{Z}^{(i)}$ in the diagonal representing a specific concept. $k$ represents the number of blocks, which is directly associated with the number of distinct concept. Though we assume that $\mathbf{S} = [\mathbf{S}^{(1)}, \mathbf{S}^{(2)}, \cdots, \mathbf{S}^{(k)}]$ is ordered according to the true membership for the simplicity of discussion, the input matrix in Eq. 2 can be $\tilde{\mathbf{S}} = \mathbf{S}\mathbf{P}$, where $\mathbf{P}$ can be any permutation matrix which reorders the columns of $\mathbf{S}$.

**Theorem 5.1** *In Drift2Matrix, the representation matrix obtained for a permuted input data is equivalent to the permutation-transformed original representation matrix. Let $\mathbf{Z}$ be feasible to $\Phi(\mathbf{S}) = \Phi(\mathbf{S})\mathbf{Z}$, then $\tilde{\mathbf{Z}} = \mathbf{P^T}\mathbf{Z}\mathbf{P}$ is feasible to $\Phi(\tilde{\mathbf{S}}) = \Phi(\tilde{\mathbf{S}})\tilde{\mathbf{Z}}$. (See Appendix C.1 for proof.)*

## 5.2 KERNEL-INDUCED REPRESENTATION

The kernel-induced representation in Drift2Matrix is a key step that reveals nonlinear relationships among co-evolving time series in a high-dimensional space. This approach transforms complex, intertwined patterns in the original time series, which may not be discernible in low-dimensional spaces, into linearly separable entities in the transformed space. Essentially, it allows for a deeper and more nuanced understanding of the dynamics hidden within complex time series structures. Through kernel transformation, previously obscured correlations and patterns become discernible, enabling more precise and insightful analysis of co-evolving time series data.

Additionally, our kernel-induced representation not only projects the time series into a high-dimensional space but also preserves the local manifold structure of the time series in the original space. This preservation ensures that the intrinsic geometric and topological characteristics of the data are not lost during transformation. Our goal is to ensure that, once the time series are mapped into a higher-dimensional space, the integrity of the identified concepts remains consistent with the structure of the original space, without altering the distribution or shape of these concepts.

**Theorem 5.2** *Drift2Matrix reveals nonlinear relationships among series in a high-dimensional space while simultaneously preserving the local manifold structure of series. (See Appendix C.2 for proof.)*

## 6 EXPERIMENTS

This section presents our experiments to evaluate the effectiveness of Drift2Matrix. The experiments were designed to answer the following questions:

**(Q1) Effectiveness:** How well does Drift2Matrix identify and track concept drift?

**(Q2) Accuracy:** How accurately does Drift2Matrix forecast future concept and series value?

**(Q3) Scalability:** How does Drift2Matrix perform in online forecasting scenarios?

### 6.1 DATA AND EXPERIMENTAL SETUP

The data utilized in our experiments consists of a synthetic dataset (SyD) constructed to allow the controllability of the structures/numbers of concepts and the availability of ground truth, as well as several real-life datasets: GoogleTrend Music Player dataset (MSP), Customer electricity load (ELD) data, Chlorine concentration data (CCD), Earthquake data (EQD), Electrooculography signal (EOG), Rock dataset (RDS), two financial datasets (Stock1 & Stock2), four ETT (ETTh1, ETTh2, ETTm1, ETTm2), Traffic and Weather datasets. In our kernel representation learning process, we chose the Gaussian kernel function. Detailed information about these datasets and the setup of the kernel function can be found in Appendix F. The source code is public to the research community[4].

In the learning process, a fixed window slides over all the series and generates subseries under different windows. Then, we learn a kernel representation for the subseries in each window. Given our focus on identifying concepts and concept drift, varying window sizes actually demonstrate Drift2Matrix's ability to extend across multi-scale time series. Specifically, smaller window sizes represent a low-scale perspective, uncovering short-term subtle concept variations (fluctuations), while larger window sizes (high-scale) reflect overall concept trends (moderation). In this paper, we employ MDL (Minimum Description Length) techniques Rissanen (1998) on the segment-score obtained through our kernel-induced representation to determine a window size [5] that establishes highly similar or repetitive concepts across different windows (see Appendix D). Although adaptively determining a domain-agnostic window size forms a part of our work, to ensure fairness in comparison, all models, including Drift2Matrix, were evaluated using the same window size settings in our experiments.

### 6.2 Q1: EFFECTIVENESS

Drift2Matrix's forecasting effectiveness is evaluated through its ability to identify important concepts. Due to space limitations, here we only describe our results for the SyD and Stock1 datasets, the outputs with the other datasets are shown in Appendix H.2. Our method for automatically estimating

---

[4]https://anonymous.4open.science/r/Drift2Matrix-main-86B7

[5]For ETT, Traffic, and Weather datasets, we consider sizes of 96, 192, 336, and 720, following the standard settings used by most methods.

the optimal size of the sliding windows to obtain suitable concepts is detailed in Appendix D. The method allows obtaining window sizes of 78 and 17 for `Synthetic` and `Stock1`, respectively (see Fig 5). The Drift2Matrix output of `SyD` has already been presented in Fig. 1 of Sec. 1. As already seen, our method automatically captures typical concepts in a given co-evolving time series (Fig. 1 (b-c)), and dynamic concept drifts (Fig. 1 (d)). Drift2Matrix views co-evolving time series as an ecosystem, enabling precise detection of interconnected dynamics and drifts. This also facilitates forecasting future concepts and values (The grey areas in Fig. 1 (d)).

Fig. 2(a) illustrates various concepts identified in the `Stock1` dataset, revealing patterns of market volatility. These discovered concepts are meaningful as they encapsulate different market behaviors, such as steady trends, spikes, or dips, corresponding to distinct phases in market activity. For example, Concept 4 represents stable periods with low volatility, while Concepts 2 and 5 capture moments of sudden market fluctuations or high-risk events. Additionally, the subtle differences between Concepts 2 and 5 highlight the model's ability to detect gradual drifts in market behavior. Using these concepts, we trace series exhibiting concept drift over different windows. For the purpose of illustra-

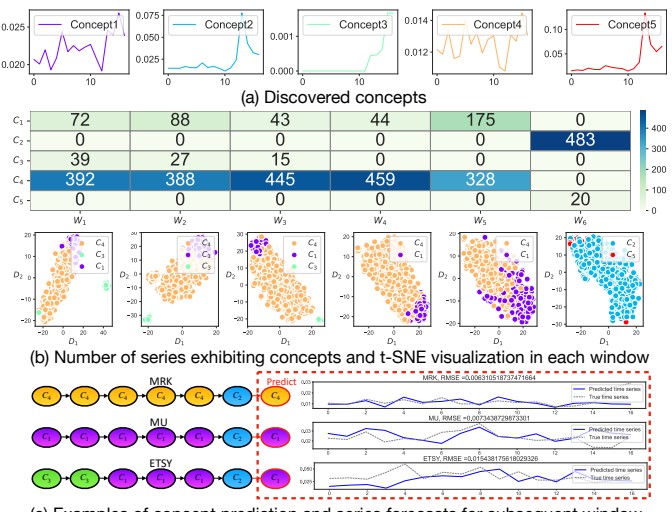

Figure 2: Visualized results on `Stock1`.

tion, in Fig. 2(b), we plot the heatmap of each concept across different windows and their corresponding 2D visualizations using t-Distributed Stochastic Neighbor Embedding (t-SNE) Van der Maaten & Hinton (2008). Darker heatmap cells indicate more prevalent concepts. The exclusive appearance of $C_2$ and $C_5$ in the $6^{th}$ window, absent in the preceding ones, denotes significant changes and fluctuations in the financial markets – i.e., possibly induced by the COVID-19 pandemic. Fig. 2(c) shows the concept drifts and predictions for three random stocks (NASDAQ: MRK, MU and ETSY) from `Stock1`, demonstrating Drift2Matrix's ability to track and forecast concept drifts.

### 6.3 Q2: ACCURACY

For real datasets, we lack the ground truth for validating the obtained concepts. Instead, we validate the value and gain of the discovered concepts for time series forecasting as they are employed in the forecasting formula Eq. 6. In this section, we evaluate the forecasting performance of the proposed model against seventeen different models, utilizing the Root Mean Square Error (RMSE) as an evaluative metric. Due to space limitations, we only present results for seven comparison models here; the complete experimental results can be found in the Appendix H.3. These seven models include four forecasting models (ARIMA Box (2013), KNNR Chen & Paschalidis (2019), INFORMER Zhou et al. (2021), and a ensemble model N-BEATS Oreshkin et al. (2019)), and three are concept-drift models (Cogra Miyaguchi & Kajino (2019), OneNet Wen et al. (2024) and OrBitMap Matsubara & Sakurai (2019)). For the existing methods, we use the codes released by the authors, and the details of the parameter settings can be found in Appendix G.

Table 1 shows the forecasting performance of the models. We see that our model consistently outperforms the other models, achieving the lowest forecasting error on most datasets. ARIMA has the ability to capture seasonality patterns within time series; however, when the various seasonalities are noncontiguous, the models face difficulties in capturing complex, nonlinear dynamic interactions between time series. N-BEATS, a state-of-the-art deep learning model, while generally effective due to its ensemble-based architecture, does not consistently perform as well as Drift2Matrix or OneNet, particularly in capturing concept drift across multiple time series. OneNet, like N-BEATS, achieves good results due to its ensemble-based strengths. However, Drift2Matrix achieves comparable results. Notably, Drift2Matrix is not primarily designed as a forecasting model; rather, it focuses

Table 1: Models' forecasting performance, in terms of RMSE

| Datasets | Horizon | Forecasting models | | | | Concept-aware models | | | | |
|---|---|---|---|---|---|---|---|---|---|---|
| | | ARIMA | KNNR | INFORMER | N-BEATS | Cogra | OneNet | OrbitMap | Drift2Matrix | Auto-D2M |
| SyD | 78 | 1.761 | 1.954 | 0.966 | 0.319 | 1.251 | 0.317 | 0.635 | **0.315** | *0.313* |
| MSP | 31 | 6.571 | 4.021 | 2.562 | 0.956 | 2.898 | 0.751 | 1.244 | **0.663** | *0.659* |
| ELD | 227 | 2.458 | 2.683 | 2.735 | 1.593 | 2.587 | **1.101** | 1.835 | 1.644 | *1.669* |
| CCD | 583 | 8.361 | 6.831 | 3.746 | 1.692 | 3.604 | **1.298** | 1.753 | 1.387 | *1.392* |
| EQD | 50 | 5.271 | 3.874 | 4.326 | 1.681 | 3.949 | **1.386** | **1.386** | 1.392 | *1.388* |
| EOG | 183 | 3.561 | 3.452 | 4.562 | 2.487 | 4.067 | **1.337** | 3.251 | **1.198** | *1.191* |
| RDS | 69 | 6.836 | 6.043 | 5.682 | 2.854 | 5.135 | 1.865 | 4.571 | **1.699** | *1.689* |
| Stock1 ($\times 10^{-2}$) | 17 | 2.635 | 2.348 | 2.127 | 1.035 | 2.137 | 0.923 | 1.003 | **0.878** | *0.902* |
| Stock2 ($\times 10^{-2}$) | 11 | 2.918 | 2.761 | 1.064 | 0.607 | 1.367 | 0.312 | 0.747 | **0.303** | *0.317* |
| ETTh1 | 96 | 1.209 | 0.997 | 0.966 | 0.933 | **0.909** | 0.916 | **0.909** | 0.913 | *0.907* |
| | 192 | 1.267 | 1.034 | 1.005 | 1.023 | 0.996 | **0.975** | 0.991 | 0.979 | *0.977* |
| | 336 | 1.297 | 1.057 | 1.035 | 1.048 | 1.041 | 1.028 | 1.039 | **1.018** | *1.015* |
| | 720 | 1.347 | 1.108 | 1.088 | 1.115 | 1.095 | 1.082 | 1.083 | **1.073** | *1.085* |
| ETTh2 | 96 | 1.216 | 0.944 | 0.943 | 0.892 | 0.901 | 0.889 | 0.894 | **0.885** | *0.879* |
| | 192 | 1.250 | 1.027 | 1.015 | 0.979 | 0.987 | **0.968** | 0.976 | 0.977 | *0.970* |
| | 336 | 1.335 | 1.111 | 1.088 | 1.040 | 1.065 | **1.039** | 1.052 | 1.044 | *1.037* |
| | 720 | 1.410 | 1.210 | 1.146 | **1.101** | 1.131 | 1.119 | 1.120 | 1.115 | *1.121* |
| ETTm1 | 96 | 0.997 | 0.841 | 0.853 | 0.806 | 0.780 | **0.777** | 0.778 | 0.781 | *0.777* |
| | 192 | 1.088 | 0.898 | 0.898 | 0.827 | 0.819 | 0.813 | 0.810 | **0.805** | *0.801* |
| | 336 | 1.025 | 0.886 | 0.885 | 0.852 | 0.838 | **0.819** | 0.820 | 0.822 | *0.819* |
| | 720 | 1.070 | 0.921 | 0.910 | 0.903 | 0.890 | **0.859** | 0.868 | 0.864 | *0.854* |
| ETTm2 | 96 | 0.999 | 0.820 | 0.852 | **0.804** | 0.824 | 0.812 | 0.821 | 0.810 | *0.802* |
| | 192 | 1.072 | 0.874 | 0.902 | 0.829 | 0.849 | 0.830 | 0.832 | **0.825** | *0.824* |
| | 336 | 1.117 | 0.905 | 0.892 | 0.852 | 0.854 | **0.841** | 0.842 | 0.847 | *0.839* |
| | 720 | 1.176 | 0.963 | 0.965 | 0.897 | 0.921 | 0.896 | 0.906 | **0.886** | *0.876* |
| Traffic | 96 | 1.243 | 1.006 | 0.895 | 0.893 | 0.898 | 0.884 | 0.883 | **0.880** | *0.874* |
| | 192 | 1.253 | 1.021 | 0.910 | 0.920 | 0.908 | **0.883** | 0.895 | 0.888 | *0.880* |
| | 336 | 1.260 | 1.028 | 0.916 | **0.895** | 0.922 | 0.901 | 0.908 | 0.937 | *0.926* |
| | 720 | 1.285 | 1.060 | 0.968 | 0.949 | 0.964 | 0.940 | 0.946 | **0.932** | *0.923* |
| Weather | 96 | 1.013 | 0.814 | 0.800 | 0.752 | 0.759 | 0.745 | 0.744 | **0.737** | *0.742* |
| | 192 | 1.021 | 0.867 | 0.861 | 0.798 | 0.793 | 0.776 | 0.775 | **0.771** | *0.769* |
| | 336 | 1.043 | 0.872 | 0.865 | 0.828 | 0.825 | 0.801 | 0.806 | **0.791** | *0.786* |
| | 720 | 1.096 | 0.917 | 0.938 | 0.867 | 0.863 | **0.833** | 0.841 | 0.840 | *0.832* |

While forecasting series task is not our main focus, we provide a comparison of Drift2Matrix with other models. Results for the extended Auto-D2M, are included but not part of the comparison. Complete experimental results can be found in the Appendix H.3.

on uncovering concepts and tracking concept drift. This unique focus allows Drift2Matrix to excel in identifying complex, dynamic interactions within time series data. Meanwhile, OrbitMap, while also concept-aware, is hindered by its necessity for predefined concepts and struggles with handling multiple time series.

### 6.4 Q3: SCALABILITY

To further illustrate the predictive scalability of Drift2Matrix, we employed it for one of the most challenging tasks in time series analysis – i.e., online forecasting, leveraging the discovered concepts. For this task, our objective is to forecast upcoming unknown future events, at any given moment, while discarding redundant information. This approach is inherently aligned with online learning paradigms, where the model continually learns and adapts to new data points, making it highly pertinent in the dynamic landscape of financial markets. We conducted tests on the Stock2 dataset.

Fig. 3 illustrates the online forecasting examples on four stocks (NASDAQ: CSX, ULTA, UNP and BK) and showcases snapshots at several time-stamps. The original data at the top of Fig. 3(a)

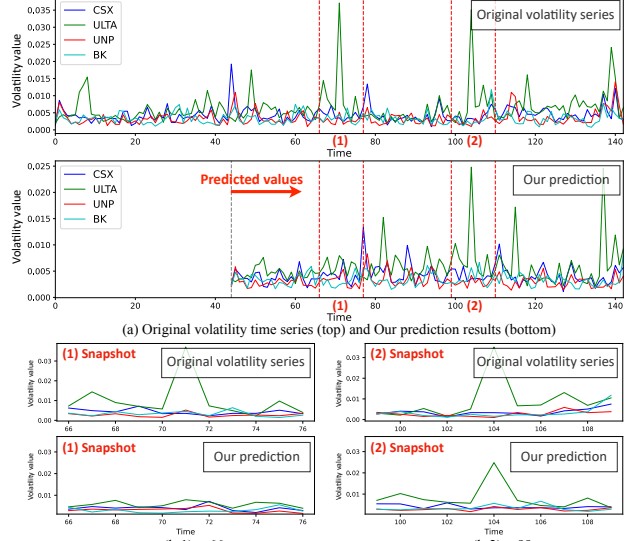

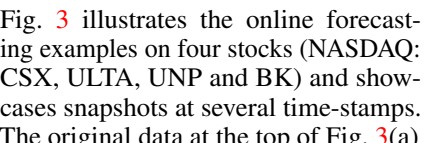

Figure 3: Online forecasting results on Stock2

elucidates the daily volatility fluctuations for these four stocks. The lower part of Fig. 3(a) unveils the outcomes for online forecasting, showcasing how our model anticipates series behavior over

time. For stock ULTA, depicted by the green line, all the compared models, including our model, encounter challenges in predicting the abnormal behavior of the first high volatility (Fig. 3(b-1)) due to the absence of antecedent knowledge. However, post encountering this anomalous behavior, our model accurately anticipates the timing of the second anomalous high-volatility behavior (Fig. 3(b-2)) and the ensuing volatility behavior, attributing to Drift2Matrix's ability to model concept drift by leveraging the interrelations among multiple time series. Specifically, for a single-series concept drift model, it is impossible to predict the second anomalous behavior accurately if there is no periodic pattern in it; whereas for our model, since our concept drift is based on correlations between series, when other series start to show some anomalous volatility behavior (albeit small), our model is also able to predict the next volatility of multiple time series in a holistic way.

### 6.5 ADDITIONAL EXPERIMENTS

Further experiments and ablation studies can be found in Appendix H, including

- Appendix H.1: We evaluate Drift2Matrix's ability to handle noise or outliers, demonstrating its robustness.
- Appendix H.2: We expanded the model's evaluation to include the other datasets like MSP, ELD, CCD, EQD, EOG, and RDS. For each dataset, distinct concepts exhibited by co-evolving series were identified, demonstrating the model's robustness in concept identification across various datasets.
- Appendix H.3: A complete result comparing our model with other models, providing a comprehensive view of the model's performance across all datasets.
- Appendix H.4: A comparative analysis was conducted between our model and the N-BEATS model on Stock1 and Stock2 datasets. This comparison highlighted the limitations of N-BEATS in capturing complex concept transitions within multiple time series.
- Appendix H.5: The model's application to motion segmentation on the Hopkins155 database was explored. This case study demonstrated the model's effectiveness in handling different types of sequences and its adaptability to various motion concepts.
- Appendix H.6: A comprehensive analysis of RMSE values across all datasets was presented, showcasing our model's superior performance in comparison to other models.
- Appendix H.7: A comprehensive analysis of complexity and execution time Evaluation.
- Appendix H.8: Detailed ablation studies were conducted to validate the efficacy of various components of the Drift2Matrix's kernel representation learning, including regularizations, kernel-based methods, and different kernel functions. These studies provided insights into the model's performance under different configurations and conditions.

## 7 CONCLUSION

In this work, we devised a principled method for identifying and modeling intricate, non-linear interactions within an ecosystem of multiple time series. The method enables us to predict both concept drift and future values of the series within this ecosystem. One noteworthy feature of the proposed method is its ability to identify and handle multiple time series dominated by concepts. This is accomplished by devising a kernel-induced representation learning, from which the time-varying kernel self-representation matrices and the block-diagonal property are utilized to determine concept drift. The proposed method adeptly reveals diverse concepts in the series under investigation without requiring prior knowledge. This work opens up avenues for further research into time series analysis, particularly regarding concept drift mechanisms in multi-series ecosystems.

Despite its strengths, Drift2Matrix has a limitation when applied to time series with few variables. For example, converting a dataset with five variables into a 5x5 matrix makes block diagonal regularization less effective. Conversely, larger datasets, like those with 500 variables, benefit significantly from our method, enabling the identification of nonlinear relationships and concept drift. This characteristic is somewhat counterintuitive compared to most existing time series models that often focus on single or low-dimensional (few variables) time series forecasting, such as sensor data streams. Despite this limitation, we believe it underscores Drift2Matrix's unique appeal. It addresses a gap in handling concept drift in time series with a large number of variables, offering excellent interpretability and reduced computational complexity.

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

# Supplementary Materials for: Drift2Matrix

In this document, we have gathered all the results and discussions that, due to page limitations, were not included in the main manuscript.

# Appendix

## A    EXTENDED RELATED WORK AND MOTIVATION

**Concept drift models.** Co-evolving time series analysis, by its nature, entails the simultaneous observation and interpretation of interdependent data streams Cavalcante et al. (2016). This complexity is further heightened when concept drift is introduced into the model Webb et al. (2016). In such scenarios, a shift in one variable can propagate through the network of interrelations, affecting the entire co-evolving system. Matsubara et al. Matsubara & Sakurai (2016) put forth the RegimeCast model, which learns potential patterns within a designated time interval in a co-evolving environment and predicts the subsequent pattern most likely to emerge. While the approach can forecast following patterns, it is not designed to account for any interdependencies between them. In their subsequent work Matsubara & Sakurai (2019), the authors introduced the deterministic OrbitMap model to capture the temporal transitions across displayed concepts. Notably, this approach relies on pre-labeled concepts (known beforehand). DDG-DA Li et al. (2022) for data distribution generation has been adapted to better suit co-evolving scenarios, addressing the unique challenges presented by the interplay of multiple data streams under concept drift conditions. However, this method defines the concept as a collective behavior represented by co-evolving time series, rather than capturing the dynamics of individual series and their interactions. While acknowledging that deep learning has made significant advances in time series field, we must also note that most of these progress aims at improving accuracy. For example, OneNet Wen et al. (2024) addresses the concept drift problem by integrating an ensemble of models that share different data biases and learning to dynamically combine forecasts from these models for enhanced prediction. It maintains two forecasting models focusing on temporal correlation and cross-variable dependency, trained independently and dynamically adjusted during testing; FSNet Pham et al. (2022), on the other hand, is designed to quickly adapt to new or recurring patterns in non-stationary environments by enhancing a neural network backbone with two key components: an adapter for recent changes and an associative memory for recurrent patterns. Dish-TS Fan et al. (2023) offers a general approach for alleviating distribution shift in time series forecasting by normalizing model inputs and outputs to better handle distribution changes. Similarly, Cogra's application of the Sequential Mean Tracker (SMT) adjusts to changes in data distribution, improving forecast accuracy Miyaguchi & Kajino (2019).

**Representation Learning on TS.** Representation learning on time series (TS) has gained significant attention due to its potential in uncovering underlying patterns and features essential for various downstream tasks. T-Rep Fraikin et al. (2023) leverages time-embeddings for time series representation. This method focuses on capturing temporal dependencies and variations through time-specific embeddings. TimesURL Liu & Chen (2024) employs self-supervised contrastive learning to create representations for time series. BTSF Yang & Hong (2022) is an unsupervised method for time series

representation learning that iteratively fuses temporal and spectral information. The bilinear fusion mechanism allows the model to capture both temporal dynamics and spectral characteristics of the time series. Timemae Cheng et al. (2023) leverages self-supervised learning to learn representations of time series using decoupled masked autoencoders. This method focuses on reconstructing the time series data by masking certain parts of the input and learning to predict the missing information. While the aforementioned methods have advanced time series representation learning, they have several limitations. Many approaches assume linear relationships, limiting their ability to capture complex, non-linear dependencies inherent in co-evolving time series. Additionally, techniques heavily reliant on specific features, such as spectral characteristics, may not generalize well across diverse datasets. The computational complexity of some advanced representation learning methods poses challenges for scalability, especially when applied to large, co-evolving datasets. Moreover, these methods often focus on representation learning for single time series or treat co-evolving time series as a data stream, rather than uncovering the intricate non-linear relationships among series.

**Motivation.** Our work aims to propose a novel perspective on concept evaluation in co-evolving time series. Eschewing traditional methods that rely on latent variable dynamics, we delve into the inherent behavior of the time series. Our proposed Drift2Matrix, with its nonlinear mapping, is adept at capturing the ever-changing concepts, offering insights into their intricacies and forecasting potential concept drift. The comparison of our Drift2Matrix framework with the recent advances in deep learning methods, is given from the following three perspectives:

Table 2: Capabilities of approaches.

| | HMM/++ | ARIMA/++ | WCPD-RS | ORBITMAP | LSTM/N-BEATS/INFORMER | COGRA | ONENET | FSNET | Drift2Matrix |
|---|---|---|---|---|---|---|---|---|---|
| Multiple time series | - | - | - | - | ✓ | - | ✓ | - | ✓ |
| Time series compression | ✓ | - | ✓ | ✓ | - | ✓ | - | - | ✓ |
| Domain agnostic segmentation | - | - | - | ✓ | - | - | - | - | ✓ |
| Concept identification (non-linear interaction) | - | - | - | - | - | - | - | - | ✓ |
| Concept trajectory tracking | - | - | ✓ | ✓ | - | - | - | - | ✓ |
| Mitigating Concept Drift Impact | - | - | ✓ | ✓ | - | ✓ | ✓ | ✓ | ✓ |
| Forecasting | - | ✓ | - | ✓ | ✓ | ✓ | ✓ | ✓ | ✓ |

- **Focus of Research.** While acknowledging the rapid developments in deep learning for time series forecasting, it's crucial to point out that most of these advancements concentrate predominantly on forecasting accuracy. Notably, even the most recent deep learning methods that mention concept drift, such as OneNet Wen et al. (2024) and FSNet Pham et al. (2022), primarily aim to mitigate the impact of concept drift on forecasting. They achieve this by incorporating an ensemble of models with diverse data biases or by refining network parameters for better adaptability. In contrast, Drift2Matrix focuses on the challenges of adaptive concept identification and dynamic concept drift in co-evolving time series. Our model delves deeper into the inherent structure of time series data, enabling a more nuanced understanding and handling of concept drift by dynamically identifying and adapting to new concepts as they emerge.

- **Interpretability.** Drift2Matrix introduces kernel-induced representation to reveal nonlinear relationships in time series, substantially boosting both adaptability and interpretability of the model. In particular, Drift2Matrix transforms time series data into a matrix format, where its block diagonal structure intuitively maps out distinct concepts. Conversely, while methods like feature importance scoring and attention mechanisms aim to improve deep learning models' interpretability, they often rely on post-hoc analysis of the model's internal mechanisms.

- **Example of Financial Market.** Financial time series analysis transcends mere forecasting; it demands interpretability that builds trust and supports applications like portfolio management. Drift2Matrix excels in this regard, offering clear insights into market dynamics beyond the conventional categories of bull, bear, or sideways markets. It adeptly captures a wide array of market scenarios, identifying distinct concepts driven by various factors—whether it's value versus growth or the interplay between small and large caps. For in-depth case studies, including analyses of the Stock1 and Stock2 datasets, please refer to Sec 6.2 & 6.4.

**Formal Mathematical Definition of Concepts.** For window $W_p$, the $r$-th concept is defined as the vector representation of subseries corresponding to the $r$-th block, $\mathbf{Z}_p^{(r)}$, in the representation matrix $\mathbf{Z}_p$. Specifically, it aligns with the centroid of similar subseries, represented as $C_{r,p} = \text{Centroid}\left(\{\mathbf{S}_i \mid \mathbf{S}_i \in \mathbf{Z}_p^{(r)}\}\right)$. To differentiate and refine similar or repeated concepts across different

windows, two concepts $C_{r,p}$ and $C_{s,p+1}$ are considered distinct if $||C_{r,p} - C_{s,p+1}||\_F^2 > \rho$, where $\rho$ is a tunable hyperparameter that regulates the granularity of concept.

## B ESTIMATING THE NUMBER OF CONCEPTS

In time series analysis, accurately identifying the number of concepts, such as periods of varying volatility in financial market, stages of a disease in medical monitoring, or climatic patterns in meteorology, is crucial. These concepts offer insights for data-driven decision-making, understanding underlying dynamics, and predicting future behaviors. While estimating the number of concepts is generally challenging, our kernel-induced representation learning approach offers a promising solution. Leveraging the block-diagonal structure of the self-representation matrix produced by our method, we can effectively estimate the number of concepts. According to the Laplacian matrix property Von Luxburg (2007), a strictly block-diagonal matrix $\mathbf{Z}$ allows us to determine the number of concepts $k$ by first calculating the Laplacian matrix of $\mathbf{Z}$ ($\mathbf{L_Z}$) and then counting the number of zero eigenvalues of $\mathbf{L_Z}$. Although the dataset is not always clean or noise-free (as is often the case in practice), we propose an eigengap thresholding approach to estimating the number of concepts. This approach estimates the number of concepts $\hat{k}$ as:

$$\hat{k} = \arg\min_{i} \{i | g(\sigma_i) \leq \tau\}_{i=1}^{N-1} \tag{8}$$

Where $0 < \tau < 1$ is a parameter and $g(\cdot)$ is an exponential eigengap operator defined as:

$$g(\sigma_i) = e^{\lambda_{i+1}} - e^{\lambda_i} \tag{9}$$

Here, $\lambda_{i_{i=1}}^N$ are the eigenvalues of $\mathbf{L_Z}$ in increasing order. The eigengap, or the difference between the $i^{th}$ and $(i+1)^{th}$ eigenvalues, plays a crucial role. According to matrix perturbation theory Stewart (1990), a larger eigengap indicates a more stable subspace composed of the selected $k$ eigenvectors. Thus, the number of concepts can be determined by identifying the first extreme value of the eigengap [6].

## C PROOFS AND OPTIMIZATION

### C.1 PERMUTATION INVARIANCE OF REPRESENTATION MATRIX IN SEQMATRIX

**Theorem C.1** *In Drift2Matrix, the representation matrix obtained for a permuted input data is equivalent to the permutation-transformed original representation matrix. Specifically, let $\mathbf{Z}$ be feasible to $\Phi(\mathbf{S}) = \Phi(\mathbf{S})\mathbf{Z}$, then $\tilde{\mathbf{Z}} = \mathbf{P^T Z P}$ is feasible to $\Phi(\tilde{\mathbf{S}}) = \Phi(\tilde{\mathbf{S}})\tilde{\mathbf{Z}}$.*

**Proof C.2** *Given a permutation matrix $\mathbf{P}$, consider the self-representation matrix $\tilde{\mathbf{Z}}$ for the permuted data matrix $\mathbf{SP}$. The objective for $\mathbf{SP}$ becomes:*

$$\min_{\tilde{\mathbf{Z}}} \left\| \Phi(\mathbf{SP}) - \Phi(\mathbf{SP})\tilde{\mathbf{Z}} \right\|^2 + \Omega(\tilde{\mathbf{Z}}), \quad s.t. \ \tilde{\mathbf{Z}} = \tilde{\mathbf{Z}}^{\mathrm{T}} \geq 0, \ diag(\tilde{\mathbf{Z}}) = 0 \tag{10}$$

*By the properties of kernel functions and permutation matrices, we have $\Phi(\mathbf{SP}) = \Phi(\mathbf{S}P)$. Substituting this into the objective function for $\tilde{\mathbf{Z}}$, we have:*

$$\min_{\tilde{\mathbf{Z}}} \left\| \Phi(\mathbf{SP}) - \Phi(\mathbf{S})\mathbf{P}\tilde{\mathbf{Z}} \right\|^2 + \Omega(\tilde{\mathbf{Z}}) \quad s.t. \ \tilde{\mathbf{Z}} = \tilde{\mathbf{Z}}^{\mathrm{T}} \geq 0, \ diag(\tilde{\mathbf{Z}}) = 0 \tag{11}$$

*Since $P$ is a permutation matrix, $\mathbf{PP}^{\mathrm{T}} = \mathbf{I}$, the identity matrix. We apply the transformation $\mathbf{P}\tilde{\mathbf{Z}}\mathbf{P}^{\mathrm{T}}$ to the objective function:*

$$\min_{\tilde{\mathbf{Z}}} \left\| \Phi(\mathbf{S}) - \Phi(\mathbf{S})\mathbf{P}\tilde{\mathbf{Z}}\mathbf{P}^{\mathrm{T}} \right\|^2 + \Omega(\tilde{\mathbf{Z}}) \quad s.t. \ \tilde{\mathbf{Z}} = \tilde{\mathbf{Z}}^{\mathrm{T}} \geq 0, \ diag(\tilde{\mathbf{Z}}) = 0 \tag{12}$$

---

[6] *In this paper, we simply initialize the number of concepts for each window as 3 to obtain the initial representation $\mathbf{Z}$, and then estimate k.*

*For the function to be minimized, $\mathbf{P}\tilde{\mathbf{Z}}\mathbf{P}^{\mathrm{T}}$ must be the optimal representation matrix for $\mathbf{S}$, which is $\mathbf{Z}$. Therefore, $\mathbf{P}\tilde{\mathbf{Z}}\mathbf{P}^{\mathrm{T}} = \mathbf{Z}$, or equivalently, $\tilde{\mathbf{Z}} = \mathbf{P}^{\mathrm{T}}\mathbf{Z}\mathbf{P}$.*

*This result shows that the self-representation matrix $\mathbf{Z}$ for $\mathbf{S}$ transforms to $\mathbf{P}^{\mathrm{T}}\mathbf{Z}\mathbf{P}$ for the permuted data matrix $\mathbf{SP}$, demonstrating the invariance of the representation matrix under permutations of the data matrix.*

### C.2 MANIFOLD STRUCTURE PRESERVATION IN DRIFT2MATRIX

**Theorem C.3** *Drift2Matrix reveals nonlinear relationships among time series in a high-dimensional space while simultaneously preserving the local manifold structure of series.*

**Proof C.4** *Optimization problem Eq. 3 can be converted to the form of a matrix trace:*

$$\min_{\mathbf{Z}} \frac{1}{2}\mathrm{Tr}(\mathcal{K} - 2\mathcal{K}\mathbf{Z} + \mathbf{Z}^{\mathrm{T}}\mathcal{K}\mathbf{Z}) + \gamma||\mathbf{Z}||_{\boxed{k}}, \tag{13}$$

*In the above, the negative term $-\mathrm{Tr}(\mathcal{K}\mathbf{Z})$ can be transformed into*

$$\min_{\mathbf{Z}} -\mathrm{Tr}(\mathcal{K}\mathbf{Z}) = \min_{\mathbf{Z}} \sum_{i=1}^{N}\sum_{j=1}^{N} -\Phi(\mathbf{S}_i)^{\mathrm{T}}\Phi(\mathbf{S}_j)\mathbf{Z}_{ij} = \min_{\mathbf{Z}} \sum_{i=1}^{N}\sum_{j=1}^{N} -\mathcal{K}(\mathbf{S}_i, \mathbf{S}_j)\mathbf{Z}_{ij} \tag{14}$$

*where $\mathcal{K}(\mathbf{S}_i, \mathbf{S}_j)$ indicates the similarity between $\mathbf{S}_i$ and $\mathbf{S}_j$ in kernel space. It can be seen from Eq. 14 that a large similarity (small distance) $\mathcal{K}(\mathbf{S}_i, \mathbf{S}_j)$ tends to cause a large $\mathbf{Z}_{ij}$, and vice versa. This is in fact an kernel extension of preserving local manifold structure in linear space, i.e., $\min_{\mathbf{Z}} \sum_{i=1}^{N}\sum_{j=1}^{N}||\mathbf{X}_i - \mathbf{X}_j||^2\mathbf{Z}_{ij}$ Nie et al. (2014); Zhan et al. (2018). Suppose a small weight is given to this negative term, which means that the self-representation of the data will take into account the contribution of all other data. Conversely, it will only consider the contribution of other data that are nearest neighbours to the data, thus further enhancing the sparsity of the self-representation $\mathbf{Z}$ while maintaining the local manifold structure.*

### C.3 OPTIMIZATION OF NONCONVEX PROBLEM

The optimization problem of Eq. 3 can be solved by the Augmented Lagrange method with Alternating Direction Minimization strategy Lin et al. (2011). Normally, we require $\mathbf{Z}$ in Eq. 3 to be nonnegative and symmetric, which are necessary for defining the block diagonal regularizer. However, the restrictions on $\mathbf{Z}$ will limit its representation capability. Thus, we introducing an intermediate-term $\mathbf{V}$ and transform Eq. 3 to:

$$\min_{\mathbf{Z},\mathbf{V}} \frac{1}{2}\mathrm{Tr}(\mathcal{K} + \mathbf{V}^{\mathrm{T}}\mathcal{K}\mathbf{V}) - \alpha\mathrm{Tr}(\mathcal{K}\mathbf{V}) + \frac{\beta}{2}||\mathbf{V} - \mathbf{Z}||^2 + \gamma||\mathbf{Z}||_{\boxed{k}},$$
$$= \min_{\mathbf{Z},\mathbf{V}} \frac{1}{2}||\Phi(\mathbf{S}) - \frac{\alpha}{2}\Phi(\mathbf{S})\mathbf{V}||^2 + \frac{\beta}{2}||\mathbf{V} - \mathbf{Z}||^2 + \gamma||\mathbf{Z}||_{\boxed{k}} \tag{15}$$
$$s.t.\ \mathbf{Z} = \mathbf{Z}^{\mathrm{T}} \geq 0, \mathrm{diag}(\mathbf{Z}) = 0, \mathbf{1}^{\mathrm{T}}\mathbf{Z} = \mathbf{1}^{\mathrm{T}}$$

The above two models Eq. 3 and Eq. 15 are equivalent when $\beta > 0$ is sufficiently large. As will be seen in optimization, another benefit of the relaxation term $||\mathbf{Z} - \mathbf{V}||^2$ is that it makes the objective function separable. More importantly, the subproblems for updating $\mathbf{Z}$ and $\mathbf{V}$ are strongly convex, making the final solutions unique and stable.

Consider that $||\mathbf{Z}||_{\boxed{k}} = \sum_{i=N-k+1}^{N} \lambda_i(\mathbf{L}_{\mathbf{Z}})$ is a nonconvex term. Drawing from the eigenvalue summation property presented in Dattorro (2010), we reformulate it as $\sum_{i=N-k+1}^{N} \lambda_i(\mathbf{L}_{\mathbf{Z}}) = \min_{\mathbf{W}} < \mathbf{L}_{\mathbf{Z}}, \mathbf{W} >$, where $0 \preceq \mathbf{W} \preceq \mathbf{I}, \mathrm{Tr}(\mathbf{W}) = k$, see Appendix A for detail. So Eq. 15 is equivalent to

$$\min_{\mathbf{Z},\mathbf{V},\mathbf{W}} \frac{1}{2}||\Phi(\mathbf{S}) - \frac{\alpha}{2}\Phi(\mathbf{S})\mathbf{V}||^2 + \frac{\beta}{2}||\mathbf{V} - \mathbf{Z}||^2$$
$$+ \gamma < \mathrm{Diag}(\mathbf{Z}\mathbf{1}) - \mathbf{Z}, \mathbf{W} > \tag{16}$$
$$s.t.\ \mathbf{Z} = \mathbf{Z}^{\mathrm{T}} \geq 0, \mathrm{diag}(\mathbf{Z}) = 0, 0 \preceq \mathbf{W} \preceq \mathbf{I}, Tr(\mathbf{W}) = k$$

Eq. 16 contains three variables. Due to the fact that $\mathbf{W}$ is independent of $\mathbf{V}$, it is possible to combine them into a single super-variable denoted as $\{\mathbf{W}, \mathbf{V}\}$, while treating $\{\mathbf{Z}\}$ as the remaining variable. Consequently, we can iteratively update $\{\mathbf{W}, \mathbf{V}\}$ and $\mathbf{Z}$ to solve Eq. 16.

First, we set $\mathbf{Z} = \mathbf{Z}^i$, and update $\{\mathbf{W}^{i+1}, \mathbf{V}^{i+1}\}$ by

$$\{\mathbf{W}^{i+1}, \mathbf{V}^{i+1}\} = \arg \min_{\mathbf{W}, \mathbf{V}} \frac{1}{2} ||\Phi(\mathbf{S}) - \frac{\alpha}{2}\Phi(\mathbf{S})\mathbf{V}||^2$$
$$+ \frac{\beta}{2} ||\mathbf{V} - \mathbf{Z}||^2$$
$$+ \gamma < \mathrm{Diag}(\mathbf{Z1}) - \mathbf{Z}, \mathbf{W} >$$
$$s.t.\ 0 \preceq \mathbf{W} \preceq \mathbf{I}, \mathrm{Tr}(\mathbf{W}) = k$$

This process is tantamount to independently updating $\mathbf{W}^{i+1}$ and $\mathbf{V}^{i+1}$:

$$\mathbf{W}^{i+1} = \arg \min_{\mathbf{W}} < \mathrm{Diag}(\mathbf{Z1}) - \mathbf{Z}, \mathbf{W} >,$$
$$s.t.\ 0 \preceq \mathbf{W} \preceq \mathbf{I}, \mathrm{Tr}(\mathbf{W}) = k \tag{17}$$

and

$$\mathbf{V}^{i+1} = \arg \min_{\mathbf{V}} \frac{1}{2} ||\Phi(\mathbf{S}) - \frac{\alpha}{2}\Phi(\mathbf{S})\mathbf{V}||^2 + \frac{\beta}{2} ||\mathbf{V} - \mathbf{Z}||^2 \tag{18}$$

Then, setting $\mathbf{W} = \mathbf{W}^{i+1}$ and $\mathbf{V} = \mathbf{V}^{i+1}$, we update $\mathbf{Z}$ by

$$\mathbf{Z}^{i+1} = \arg \min_{\mathbf{Z}} \frac{\beta}{2} ||\mathbf{V} - \mathbf{Z}||^2 + \gamma < \mathrm{Diag}(\mathbf{Z1}) - \mathbf{Z}, \mathbf{W} >$$
$$s.t.\ \mathbf{Z} = \mathbf{Z}^{\mathrm{T}} \geq 0, \mathrm{diag}(\mathbf{Z}) = 0 \tag{19}$$

The three subproblems presented in Eq. 17-Eq. 19 are convex and possess explicit solutions. For Eq. 17, $\mathbf{W}^{i+1} = \mathbf{U}\mathbf{U}^{\mathrm{T}}$, where $\mathbf{U} \in \mathcal{R}^{N \times k}$ is composed of the $k$ eigenvectors corresponding to the smallest $k$ eigenvalues of $\mathrm{Diag}(\mathbf{Z1}) - \mathbf{Z}$. For Eq. 18, the solution is straightforwardly derived as:

$$\mathbf{V}^{i+1} = (\Phi(\mathbf{S})^{\top}\Phi(\mathbf{S}) + \beta\mathbf{I})^{-1}(\alpha\Phi(\mathbf{S})^{\top}\Phi(\mathbf{S}) + \beta\mathbf{Z})$$
$$= (\mathcal{K} + \beta\mathbf{I})^{-1}(\alpha\mathcal{K} + \beta\mathbf{Z}) \tag{20}$$

Eq. 19 is equivalent to

$$\mathbf{Z}^{i+1} = \arg \min_{\mathbf{Z}} \frac{1}{2} ||\mathbf{Z} - \mathbf{V} + \frac{\gamma}{\beta}(\mathrm{diag}(\mathbf{W})\mathbf{1}^{\mathrm{T}} - \mathbf{W})||^2$$
$$s.t. \mathbf{Z} = \mathbf{Z}^{\mathrm{T}} \geq 0, \mathrm{diag}(\mathbf{Z}) = 0 \tag{21}$$

The solution to this problem can be expressed in closed form as follows.

**Proposition C.5** *Consider the matrix $\mathbf{A} \in \mathbb{R}^{n \times n}$. Let's denote $\hat{\mathbf{A}} = \mathbf{A} - \mathrm{Diag}(\mathrm{diag}(\mathbf{A}))$. With this definition, the solution to the following optimization problem:*

$$\min_{\mathbf{Z}} \frac{1}{2} ||\mathbf{Z} - \mathbf{A}||^2, s.t.\ \mathrm{diag}(\mathbf{Z}) = 0, \mathbf{Z} \geq 0, \mathbf{Z} = \mathbf{Z}^{\top}, \tag{22}$$

*is given by $\mathbf{Z}^* = \left[ \left( \hat{\mathbf{A}} + \hat{\mathbf{A}}^{\top} \right) / 2 \right]_+.$*

**Proof C.6** *It is evident that problem Eq. 22 is equivalent to*

$$\min_{\mathbf{Z}} \frac{1}{2} ||\mathbf{Z} - \hat{\mathbf{A}}||^2, s.t.\ \mathbf{Z} \geq 0, \mathbf{Z} = \mathbf{Z}^{\top}. \tag{23}$$

*The constraint $\mathbf{Z} = \mathbf{Z}^{\top}$ suggests that $||\mathbf{Z} - \hat{\mathbf{A}}||^2 = \left\| \mathbf{Z} - \hat{\mathbf{A}}^{\top} \right\|^2.$*

*Thus*

$$\frac{1}{2} ||\mathbf{Z} - \hat{\mathbf{A}}||^2 = \frac{1}{4} ||\mathbf{Z} - \hat{\mathbf{A}}||^2 + \frac{1}{4} \left\| \mathbf{Z} - \hat{\mathbf{A}}^{\top} \right\|^2$$
$$= \frac{1}{2} \left\| \mathbf{Z} - \frac{\hat{\mathbf{A}} + \hat{\mathbf{A}}^{\top}}{2} \right\|^2 + c(\hat{\mathbf{A}})$$

*where $c(\hat{\mathbf{A}})$ only depends on $\hat{\mathbf{A}}$. Hence Eq. 23 is equivalent to*

$$\min_{\mathbf{Z}} \frac{1}{2} \left\| \mathbf{Z} - \left( \hat{\mathbf{A}} + \hat{\mathbf{A}}^\top \right)/2 \right\|^2, s.t. \mathbf{Z} \geq 0, \mathbf{Z} = \mathbf{Z}^\top,$$

*which has the solution $\mathbf{Z}^* = \left[ \left( \hat{\mathbf{A}} + \hat{\mathbf{A}}^\top \right)/2 \right]_+$.*

## D  DOMAIN AGNOSTIC WINDOW SIZE SELECTION

To segment time series, a fixed window slides over all the series and generates $b$ non-overlapping segments for each of the $N$ time series. A kernel representation for $N$ subseries under each segmentation is then learned to model concept behaviors. The quality of the representation heavily hinges on the quality of the time series segmentation. To this end, we aim to find the optimal window size to ensure that the representation learning algorithm identifies as many highly similar or repetitive clusters across different segments[7] as possible. The repetitiveness facilitates the generation of similar concepts at different windows for tracking. Note that the number of identified concepts may vary across windows. We propose a heuristic solution involving the segmentation of a segment-score function, defined as:

$$WS(w) = \frac{1}{w} \cdot \max\{C_p, 1 \leq p \leq b\} \tag{24}$$

Here, $w$ is the window size, $b$ is the number of non-overlapping segments, and $C_p$ is the number of concepts that can be discovered within window $p$ using Eq. 8 and Eq. 9. This implies that $\max\{C_p, 1 \leq p \leq b\}$ corresponds to the maximum number of concepts observed in $\mathbf{S}$. This score measures the *concept-consistency* – i.e., how the whole time series varies with various segmentation size. A small segmentation size $w$ (more segments) leads to a high concept-consistency score $WS(\cdot)$ indicating the presence of non-repeating concepts over time: time series is unstable when looking through a narrow segmentation. Conversely, a larger segmentation size (fewer segments) leads to a lower concept-consistency score, which corresponds to the cases with highly repetitive concepts. Broadly speaking, the $WS(\cdot)$ decreases and converges to zero when the segmentation size is too large to identify concepts.

Inspired by Bouguessa & Wang (2008), we suggest employing the Minimum Description Length (MDL) principle to determine the optimal window size within the available range. The core concept of the MDL principle revolves around encoding input data based on a specific model, with the aim of choosing the encoding that yields the shortest code length Rissanen (1998). Let ***WS*** be the set of all $WS(w)$ values for each window size in the available range. The MDL-selection strategy employed in our work bears resemblance to the MDL-pruning method described in Agrawal et al. (2005); Bouguessa & Wang (2008), where the data is split into two subsets (sparse and dense subsets), with one subset being discarded. In our case, we aim to divide ***WS*** into two groups $E$ and $F$. Here, $E$ encompasses the greater values of ***WS***, while the group $F$ encompasses the lower values. Following that, the border separating the two groups is chosen in order to get the optimal window size that minimizes the Minimum Description Length (MDL) criteria. The objective function can be defined according to the MDL criteria:

$$\begin{aligned} J(w) = \min_w \log_2(\mu_E) + \sum_{WS(w) \in E} \log_2(|WS(w) - \mu_E|) \\ + \log_2(\mu_F) + \sum_{WS(w) \in F} (|WS(w) - \mu_F|) \end{aligned} \tag{25}$$

In this equation, $\mu_E$, $\mu_F$ are the means of groups $E$ and $F$, respectively. The optimal window size $w$ can be found by iterating over each possible value within the range of window sizes and calculating $J_2(w)$. See Fig. 4 for an illustration. With a given sliding window of size $w$, the time series can be segmented into consecutive subseries, each spanning a length of $w$. Thus, we can represent $\mathbf{S}$ as a union of these subseries: $\mathbf{S} = \bigcup_{p=1}^b \mathbf{S}_p$. To ascertain the count of concepts $k$, we tally the distinct profile patterns present across all windows, from $W_1$ to $W_b$. Fig. 5 exhibits the selected window sizes for the respective datasets: the length of 78 (resp., 31, 227, 583, 50, 183, 69, 17) window used for the `SyD` (resp., `MSP, ELD, CCD, EQD, EOG, RDS, Stock1`) data.

---

[7] *In this paper, the terms "segment" and "window" are used interchangeably to refer to the same concept of dividing time series into non-overlapping intervals for analysis.*

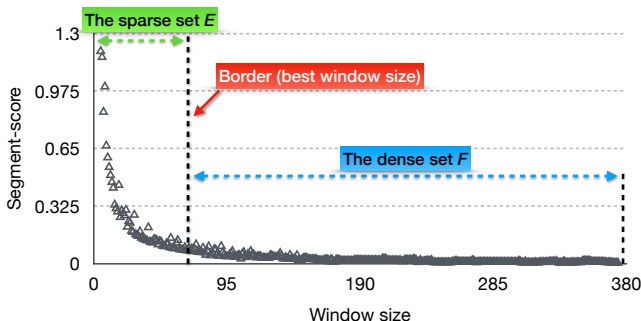

Figure 4: Partitioning of **WS** into two sets $E$ and $F$. The optimal window size is determined by the size corresponding to the border.

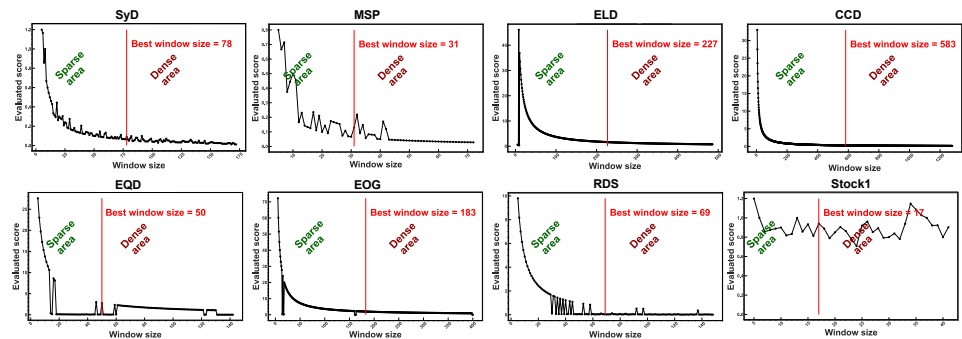

Figure 5: The best window size (red line) for the eight data sets

## E ALGORITHM OF DRIFT2MATRIX

---

**Algorithm 1** Drift2Matrix: Nonlinear concept identification

---

1: **Input:** A set of subseries $\mathbf{S} = \{S_i\}_{i=1}^N$, window sizes $\mathcal{P}$
2: **Output:** Optimal window size $w$ and a set of representations $\mathcal{Z} = \{\mathbf{Z}_p\}_{p=1}^b$
3: $w \leftarrow$ First value in $\mathcal{P}$, **WS** $\leftarrow \varnothing$
4: Scanning:
5: **for** each window $W_p$ in $\mathcal{P}$ **do**
6:     Obtain the set of subseries $\mathbf{S}_p$
7:     Update $\mathbf{Z}_p$, using Eq. 21
8:     Estimate concepts based on Eq. 8, Eq. 9
9:     Calculate the window score $WS(w)$ using Eq. 25
10:     $\mathcal{Z} \leftarrow \mathcal{Z} \cup \{\mathbf{Z}_p\}$
11:     **WS** $\leftarrow$ **WS** $\cup\, WS(w)$
12: **end for**
13: Iteration:
14: **while** —$new\ WS(w) - previous\ WS(w)$— $\geq \epsilon$ **do**
15:     $w \leftarrow$ Next value in $\mathcal{P}$
16:     Update $WS(w)$ and $\mathbf{Z}_p$ for the new $w$
17: **end while**
18: Determine the optimal window $w$ based on **WS**
19: Store the set of representations $\mathcal{Z}$
20: Discovering Concepts:
21: Identify distinct concepts $\mathbf{C}$ from $\mathcal{Z}$
22: Set number of concepts $k \leftarrow |\mathbf{C}|$

---

In this section, we delve into the detailed implementation of the Drift2Matrix algorithm, an approach designed for nonlinear concept identification and forecasting in co-evolving time series. The

Drift2Matrix algorithm operates in two distinct phases, each encapsulated in its own algorithmic structure.

The first phase, outlined in Algorithm 1, focuses on nonlinear concept identification. It takes a set of subseries and a range of window sizes as input to determine the optimal window size and a set of kernel-based representations. This phase involves scanning across different windows to estimate the concepts and evaluate window scores, leading to the identification of distinct concepts within the time series data.

The second phase, presented in Algorithm 2, builds upon the outputs of the first phase. It utilizes the optimal window size and the representations obtained to forecast the most probable concepts and the associated series values for each series within the time series data. This involves calculating transition probabilities between concepts and determining the most likely concept transitions, which are then used to forecast future values of the series.

---

**Algorithm 2** Drift2Matrix: Forecasting Concept and Series Values

1: **Input:** Optimal window size $w$ and a set of representations $\mathcal{Z} = \{\mathbf{Z}_p\}_{p=1}^b$ from Algorithm 1.
2: **Output:** Predicted concepts and series values $Pre\_S_i$ for each series $S_i$.
3: **for** each window $W_p$ and $W_{p+1}$ in $\mathcal{Z}$ **do**
4:     **for** each series $S_i$ in **S do**
5:         Calculate transition probabilities $P(\mathbf{C}_r \rightarrow \mathbf{C}_m | W_p \rightarrow W_{p+1}, S_i)$ using Eq. 4.
6:         Identify the concept $\mathbf{C}_m$ with the highest transition probability from $\mathbf{C}_r$.
7:     **end for**
8: **end for**
9: **for** each series $S_i$ in **S do**
10:     **for** $p = 1$ to (length of $\mathcal{Z}$) $- 1$ **do**
11:         Determine the most probable concept switch from $\mathbf{C}_r$ to $\mathbf{C}_m$.
12:         Calculate predicted values $Pre\_S_i$ under window $W_{p+1}$ using Eq. 6.
13:     **end for**
14: **end for**

---

## F    DETAIL OF DATASETS AND EXPERIMENTAL SETUP

We collected eight real-life datasets from various areas. The MSP dataset from online music player GoogleTrend event stream[8] contains 20 time series, each for the Google queries on a music-player spanning 219 months from 2004 to 2022. The Electricity dataset ELD comprises 1462 daily electricity load diagrams for 370 clients, extracted from UCI[9]. From the UCR's public repository[10], we obtained four time-series datasets – i.e., Chlorine concentration CCD, Earthquake EQD, Electrooculography signal EOG, and Rock dataset RDS. From Yahoo finance [11], we collected two datasets on stock. The dataset Stock1, encompasses daily OHLCV (open, high, low, close, volume) data for 503 S&P 500 stocks, spanning from 2012-01-04 to 2022-06-22. Meanwhile, Stock2 provides intra-day OHLCV data during market hours for 467 S&P 500 stocks, covering the period from 2017-05-16 to 2017-12-06. The four ETT (Electricity Transformer Temperature) datasets[12] consist of two hourly-level datasets (ETTh1, ETTh2) and two 15-minute-level datasets (ETTm1, ETTm2), each containing seven oil and load features of electricity transformers from July 2016 to July 2018. The Traffic dataset[13] describes road occupancy rates with hourly data recorded by sensors on San Francisco freeways from 2015 to 2016, while the Weather dataset[14] includes 21 weather indicators such as air temperature and humidity, recorded every 10 minutes in Germany throughout 2020.

---

[8] *http://www.google.com/trends/*

[9] *https://archive.ics.uci.edu/ml/datasets/*

[10] *https://www.cs.ucr.edu/%7Eeamonn/time_series_data_2018*

[11] *https://ca.finance.yahoo.com/*

[12] https://github.com/zhouhaoyi/ETDataset

[13] http://pems.dot.ca.gov

[14] https://www.bgc-jena.mpg.de/wetter/

Table 4: Computing Resources Used for Experiments

| Component | Specification |
|---|---|
| CPU | Intel(R) Core(TM) i7-9800X CPU @ 3.80GHz, 8 cores, 16 threads |
| Memory | 125 GB RAM |
| GPUs | 2x NVIDIA GeForce RTX 2080 Ti, each with 11 GB memory |
| GPU Driver | Version: 545.23.08 (CUDA 12.3) |
| Operating System | Ubuntu 22.04.4 LTS (GNU/Linux 6.5.0-45-generic x86_64) |
| Repetitions | All experiments repeated 3 times with different seeds |

For stock datasets, the value of interest we aim to predict is the implied volatility of each stock[15]. Given that true volatility remains elusive, we approximated it using an estimator grounded in realized volatility. We employed the conventional volatility estimator Li & Hong (2011), defined as: $\mathcal{V}_t = \sqrt{\sum_{t=1}^{n}(r_t)^2}$, where $r_t = \ln(c_t/c_{t-1})$ and $c_t$ represents the closing price at time $t$. For `Stock1`, we utilized daily data to gauge monthly volatility, while for `Stock2`, we used 1-hour intra-day data to determine daily volatility. We conducted online forecasting tests on the `Stock2` dataset, segmenting the time series with an 11-day sliding window. Given the constrained intra-day data span (7 months) and our volatility forecasting strategy, extending the sliding window would compromise the available data for assessment. At any time point $t$, the observable data encompasses a period quadruple the window size preceding time $t$, e.g., at time point t=110, our model is trained with $\mathbf{S}\,[66:109]$ and forecasts $\mathbf{S}\,[110:121]$.

Furthermore, we crafted a Synthetic `SyD` dataset comprising 500 simulated time series, each generated by a combination of following five nonlinear functions. The synthetic dataset allows the controllability of the structures/numbers of concepts and the availability of ground truth. To make a 780-steps long time series, we randomly choose one of the five functions ten times; every time, this function produces 78 sequential values – which are considered a concept. Table 3 summarizes the statistics of the datasets.

Table 3: Data statistics

| Data | | # of series | Length of series |
|---|---|---|---|
| **SyD** | | 500 | 780 |
| Real | MSP | 20 | 219 |
| | ELD | 370 | 1,462 |
| | CCD | 166 | 3,480 |
| | EQD | 139 | 512 |
| | EOG | 362 | 1,250 |
| | RDS | 50 | 2,844 |
| | Stock1 | 503 | 126 |
| | Stock2 | 467 | 143 |
| | ETTh1 | 7 | 17,420 |
| | ETTh2 | 7 | 17,420 |
| | ETTm1 | 7 | 69,680 |
| | ETTm2 | 7 | 69,680 |
| | Traffic | 862 | 17,544 |
| | Weather | 21 | 52,696 |

$$\begin{cases} g_1(t) = \cos(4\pi t/5) + \cos(\pi(t-50)) + t/100 \\ g_2(t) = \sin(\pi t/3 - 3) - \sin(\pi t/6) + t/100 \\ g_3(t) = 1 - \sin(\pi t/2 - 3) \times \cos(\pi(t-3)/6) \times \cos(\pi(t-13)) + t/100 \\ g_4(t) = \sin(\pi t/2 - 3) \times \cos(\pi(t-3)/6) \times \cos(\pi(t-13)) + t/100 \\ g_5(t) = \cos(3\pi t/5) + \sin(2\pi t/5 - t) + t/100 \end{cases}$$

In the kernel representation learning process, we used the Gaussian kernel of the form $\mathcal{K}(S_i, S_j) = exp(-||S_i - S_j||^2/d_{max}^2)$, where $d_{max}$ is the maximal distance between series. Parameters $\alpha, \gamma$ in Eq. 3 and $\beta$ in Eq. 15 are selected over [2,4,6,8,10,20], [0.1,0.4,0.8,1,4,10] and [5,10,20,40,60,100] respectively and set to be $\alpha = 4, \gamma = 0.8, \beta = 60$

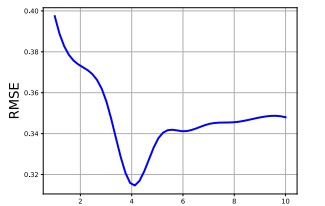 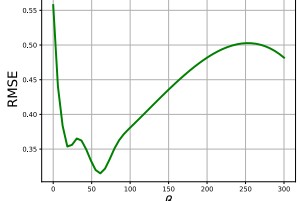

Figure 6: The effect of parameters $\alpha$, $\beta$ on `SyD`

for the best performance. Fig. 6 shows the impact of varying $\alpha$ and $\beta$ on the `SyD` dataset, while the effect of $\gamma$ can be found in Fig. 17. Table 4 summarizes the computing resources used for our experiments.

Our learning mode of kernel-induced representation can be summarized in two ways depending on the specific scenario:

- When applied to a new time series dataset, Drift2Matrix first re-adaptively determines the most suitable segmentation size $W_1, \ldots, W_b$ and learns the concept profiles within each

---

[15]We chose to validate our model through forecasting volatility as it provides a quantifiable and objective measure to assess the model's capability to understand and adapt to market changes, rather than through investment decisions or predicting market trends, which could be influenced by subjective interpretations and external market conditions and fall outside the purview of this study.

segment through kernel-induced representation. This ensures optimal alignment for concept identification.

- When receiving new data points in an online learning setting (with a fixed segmentation size), Drift2Matrix quickly updates the kernel representation matrix for the new segment, enabling efficient adaptation without recalculating the segmentation size.

For both modes, the subsequent steps remain the same: by counting the number of distinct profiles across all segments, we determine the number of distinct concepts present in the entire co-evolving time series dataset. Based on these discovered concepts, we can predict concept drift probabilities—both within a single series and through joint probabilities across the ecosystem.

## G    Parameters Setting of Comparing Methods

For all comparison methods, we set the observable historical data and prediction steps to be the same as for Drift2Matrix. For the ARIMA model, we determined the optimal parameter set using AIC; for the INFORMER model, we configured it with 3 encoder layers, 2 decoder layers, and 8 attention heads, with a model dimension of 512. We trained the model for 10 epochs with a learning rate of 0.001, a batch size of 32, and applied a dropout rate of 0.05; for the N-BEATS model, we adopted three stack modes with 1024 hidden layer units while setting the batch size to 10 for more training examples. For the other comparison methods, we performed fine-tuning on parameters to arrive at the optimal settings for each dataset.

## H    Additional Experiments

### H.1    Handling Noise and Outliers in Representation Matrix

To evaluate Drift2Matrix's ability to handle noise, we introduced artificial noise into the $\mathtt{SyD}$ dataset. We approached the issue of noise and outliers from two perspectives:

1. **Noise in the Representation Matrix**: Drift2Matrix captures relationships among series, where those belonging to the same concept form a highly correlated submatrix, visible as bright blocks in the heatmap. Self-representation learning naturally identifies noise or outliers, which appear as darker areas due to weak or no correlation. An example of this phenomenon is shown in Fig. 7(a), where a missing connected block can be seen in the lower right corner of the matrix. Importantly, the overall structure of the representation matrix $Z$ remains intact despite the presence of noise.

2. **Reconstructing a Clean Representation Matrix** $Z$: To address noise, we modify the objective function of self-representation learning. In a linear space, the objective function is adjusted from:

$$\min_Z \frac{1}{2}\|\mathbf{S} - \mathbf{SZ}\|_2^2 + \Omega(Z)$$

to the following form:

$$\min_{Z,\mathbf{E}} \Omega(Z) + \lambda\|\mathbf{E}\|_{2,1}, \quad \text{s.t.} \quad \mathbf{S} = \mathbf{SZ} + \mathbf{E}$$

Here, $\mathbf{S}$ represents the series matrix, which is composed of authentic samples from the underlying concepts, and outliers are denoted by $\mathbf{E}$. This modification allows us to obtain a noise-reduced representation matrix, with the noise captured in $\mathbf{E}$. Figure 7(b-c) provides an example of this, with $\mathbf{E}$ highlighting the noise or outliers, offering useful insights in certain domains.

### H.2    The Outputs of Concept Identification with Other Datasets and complete results

To evaluate the capability of our model to identify the concepts, we also used the $\mathtt{MSP}$ (resp., $\mathtt{ELD}$, $\mathtt{CCD}$, $\mathtt{EQD}$, $\mathtt{EOG}$, $\mathtt{RDS}$. In Fig. 8, for each row, we have the distinct concepts exhibited by co-evolving series in the $\mathtt{MSP}$, $\mathtt{ELD}$, $\mathtt{CCD}$, $\mathtt{EQD}$, $\mathtt{EOG}$ and $\mathtt{RDS}$ datasets, respectively. As can be seen, we discovered 4 different concepts in the $\mathtt{MSP}$ time series, 3 in $\mathtt{ELD}$, 4 in $\mathtt{CCD}$, 3 in $\mathtt{EQD}$, 3 in $\mathtt{EOG}$,

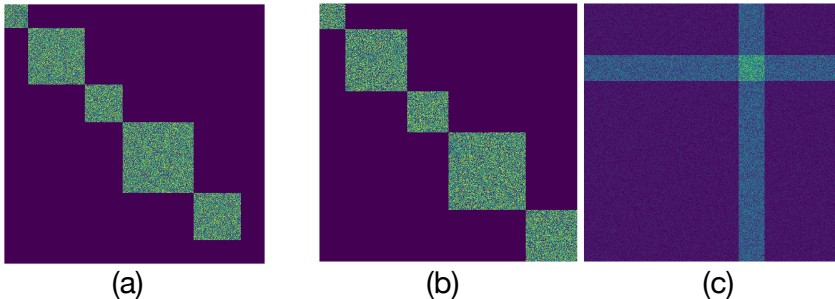

(a)                  (b)                  (c)

Figure 7: (a) Representation matrix with noise. A missing connected block in the lower right corner of the matrix due to noise data. (b-c) Reconstructed representation matrix. This allows us to obtain a noise-reduced representation matrix (b), with the noise captured in (c).

and 5 in `RDS`. For these real cases, we lack the ground truth for validating the obtained concepts. Instead, we validate the value and gain of the discovered concepts for time series forecasting as they are employed in the forecasting Eq. 6. Fig. 9 illustrates the forecasted outcomes for six arbitrarily selected time series from the respective datasets, offering a demonstrative insight into the notable efficacy of our model in forecasting time series.

### H.3 COMPREHENSIVE COMPARISON

In this section, we present a detailed comparison of Drift2Matrix and Auto-D2M with thirteen different models. The results are summarized in Table 7 which includes the performance metrics for each model across all datasets used in our experiments. This comparison highlights the strengths and weaknesses of Drift2Matrix relative to other state-of-the-art models, showcasing its superior ability to handle complex time series data and capture concept drift. Among the seventeen models, ten are forecasting models (ARIMA Box (2013), KNNR Chen & Paschalidis (2019), INFORMER Zhou et al. (2021), N-BEATS Oreshkin et al. (2019), CARD Wang et al. (2023), ETSformer Woo et al. (2022), TimesNet Wu et al. (2022), SparseTSF Lin et al. (2024), FITS Xu et al. (2024), Dlinear Zeng et al. (2023)), the other seven are concept-drift models (MSGARCH Ardia et al. (2019), SD-Markov Bazzi et al. (2017), OrBitMap Matsubara & Sakurai (2019), Cogra Miyaguchi & Kajino (2019), FEDformer Zhou et al. (2022), OneNet Wen et al. (2024) and FSNet Pham et al. (2022)).

Furthermore, we conducted additional Type I and Type II error evaluations for concept detection on the `SyD`, `Stock1`, and `Stock2` datasets. To contextualize these results, we define **True Positive (TP)**, **False Positive (FP)**, **True Negative (TN)**, and **False Negative (FN)** in the context of our concept detection methodology:

- **True Positive (TP):** The model correctly detects the presence of a ground truth concept within a window.
- **False Positive (FP):** The model incorrectly detects a concept within a window where no concept actually exists.
- **True Negative (TN):** The model correctly identifies that no concept exists within a window.
- **False Negative (FN):** The model fails to detect a ground truth concept within a window.

The evaluation results are summarized in Tables 5 and 6 below:

Table 5: Evaluation of Concept Detection: TP, FP, TN, and FN Counts.

| Dataset | Total Tests | Ground Truth Concepts | TP | FP | TN | FN |
|---|---|---|---|---|---|---|
| SyD | $500 \times 10 = 5000$ | 5 | 4900 | 50 | 4950 | 100 |
| Stock1 | $503 \times 7 = 3521$ | 5 | 3200 | 100 | 3421 | 321 |
| Stock2 | $467 \times 13 = 6071$ | 5 | 5000 | 200 | 5871 | 1071 |

These results demonstrate the robustness of Drift2Matrix in accurately detecting concept drift across datasets with different levels of complexity and domain specificity.

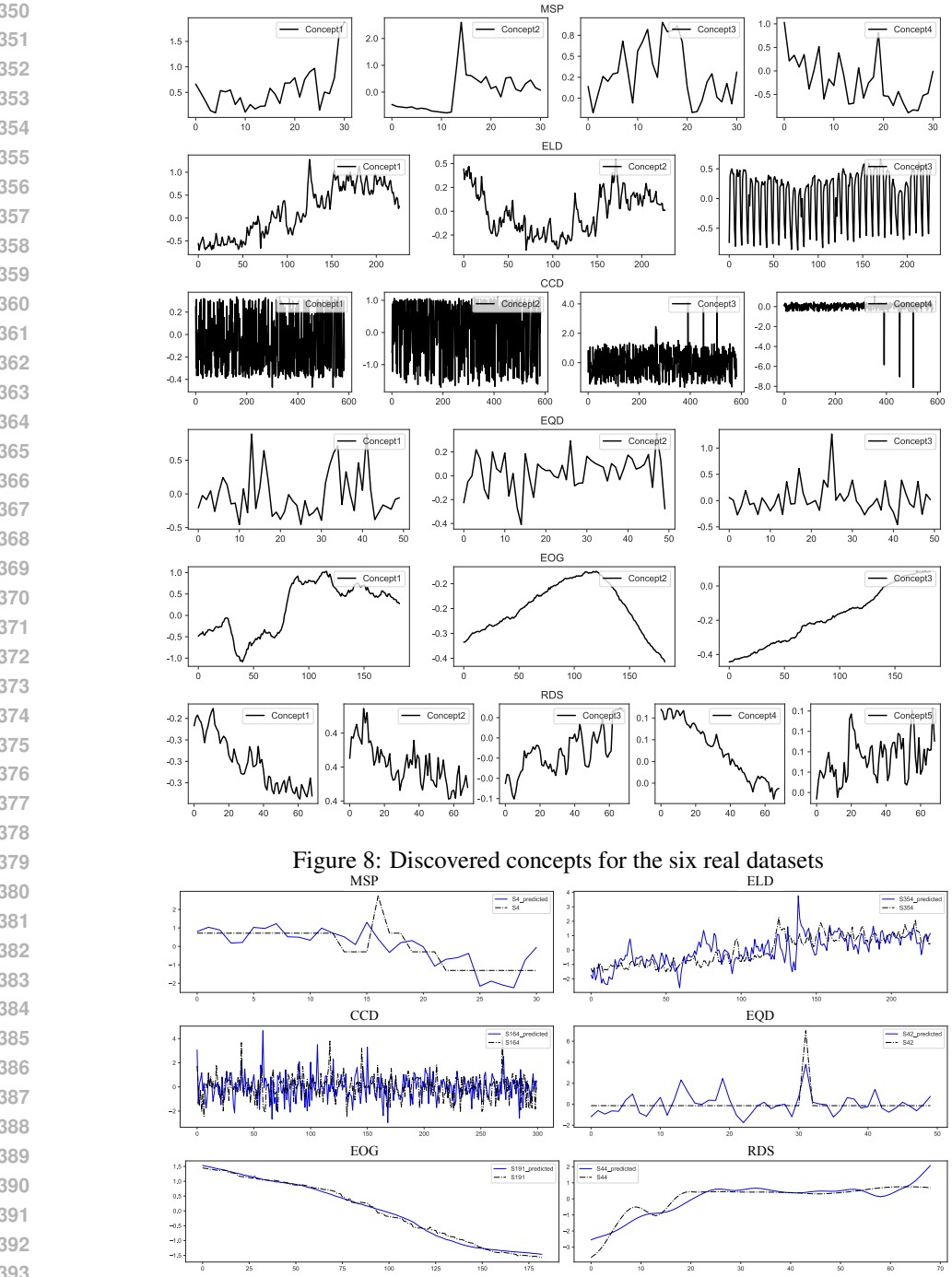

Figure 8: Discovered concepts for the six real datasets

Figure 9: True (black) and forecasted (blue) values for the six time series, each from a real dataset.

Table 6: Type I and Type II Errors for Concept Detection.

| Dataset | Type I Error (FPR) | Type II Error (FNR) |
|---------|--------------------|--------------------|
| SyD | 1.00% | 2.00% |
| Stock1 | 2.87% | 9.14% |
| Stock2 | 3.34% | 17.67% |

Table 7: Models' forecasting performance, in terms of RMSE

| Datasets | Horizon | Forecasting models | | | | | | | | | |
|---|---|---|---|---|---|---|---|---|---|---|---|
| | | ARIMA | KNNR | Informer | N-Beats | CARD | ETSformer | TimesNet | SparseTSF | FITS | Dlinear |
| SyD | 78 | 1.761 | 1.954 | 0.966 | 0.319 | 0.796 | 1.009 | 0.811 | 0.773 | 0.753 | 0.752 |
| MSP | 31 | 6.571 | 4.021 | 2.562 | 0.956 | 2.112 | 2.677 | 2.151 | 1.495 | 1.475 | 1.466 |
| ELD | 227 | 2.458 | 2.683 | 2.735 | 1.593 | 2.255 | 2.858 | 2.297 | 2.841 | 2.629 | 2.589 |
| CCD | 583 | 8.361 | 6.831 | 3.746 | 1.692 | 3.089 | 3.914 | 3.146 | 2.178 | 2.153 | 2.086 |
| EQD | 50 | 5.271 | 3.874 | 4.326 | 1.681 | 3.567 | 4.520 | 3.633 | 2.508 | 2.459 | 2.347 |
| EOG | 183 | 3.561 | 3.452 | 4.562 | 2.487 | 3.761 | 4.767 | 3.831 | 3.207 | 3.048 | 2.829 |
| RDS | 69 | 6.836 | 6.043 | 5.682 | 2.854 | 4.685 | 5.937 | 4.772 | 3.886 | 3.649 | 3.504 |
| Stock1 ($\times10^{-2}$) | 17 | 2.635 | 2.348 | 2.127 | 1.035 | 1.754 | 2.224 | 1.786 | 1.323 | 1.291 | 1.228 |
| Stock2 ($\times10^{-2}$) | 11 | 2.918 | 2.761 | 1.064 | 0.607 | 0.877 | 1.117 | 0.894 | 0.765 | 0.704 | 0.702 |
| ETTh1 | 96 | 1.209 | 0.997 | 0.966 | 0.933 | 0.939 | 0.974 | 0.949 | 0.932 | 0.920 | 0.917 |
| | 192 | 1.267 | 1.034 | 1.005 | 1.023 | 1.082 | 1.037 | 1.087 | 1.123 | 1.071 | 1.069 |
| | 336 | 1.297 | 1.057 | 1.035 | 1.048 | 1.101 | 1.087 | 1.103 | 1.107 | 1.086 | 1.076 |
| | 720 | 1.347 | 1.108 | 1.088 | 1.115 | 1.147 | 1.103 | 1.148 | 1.176 | 1.168 | 1.152 |
| ETTh2 | 96 | 1.216 | 0.944 | 0.943 | 0.892 | 0.933 | 1.001 | 0.942 | 0.945 | 0.952 | 0.946 |
| | 192 | 1.250 | 1.027 | 1.015 | 0.979 | 0.995 | 1.102 | 1.004 | 1.090 | 1.062 | 1.053 |
| | 336 | 1.335 | 1.111 | 1.088 | 1.040 | 1.072 | 1.134 | 1.076 | 1.143 | 1.127 | 1.190 |
| | 720 | 1.410 | 1.210 | 1.146 | 1.101 | 1.131 | 1.182 | 1.139 | 1.121 | 1.109 | 1.141 |
| ETTm1 | 96 | 0.997 | 0.841 | 0.853 | 0.806 | 0.810 | 0.895 | 0.821 | 0.856 | 0.837 | 0.829 |
| | 192 | 1.088 | 0.898 | 0.898 | 0.827 | 0.846 | 0.902 | 0.859 | 0.879 | 0.862 | 0.949 |
| | 336 | 1.025 | 0.886 | 0.885 | 0.852 | 0.885 | 0.905 | 0.896 | 0.904 | 0.905 | 0.916 |
| | 720 | 1.070 | 0.921 | 0.910 | 0.903 | 0.963 | 0.932 | 0.975 | 0.986 | 0.969 | 0.957 |
| ETTm2 | 96 | 0.999 | 0.820 | 0.852 | **0.804** | 0.825 | 0.885 | 0.830 | 0.854 | 0.851 | 0.848 |
| | 192 | 1.072 | 0.874 | 0.902 | 0.829 | 0.849 | 0.902 | 0.861 | 0.897 | 0.874 | 0.861 |
| | 336 | 1.117 | 0.905 | 0.892 | 0.852 | 0.863 | 0.959 | 0.867 | 0.955 | 0.943 | 0.938 |
| | 720 | 1.176 | 0.963 | 0.965 | 0.897 | 0.928 | 1.006 | 0.928 | 0.973 | 0.967 | 0.960 |
| Traffic | 96 | 1.243 | 1.006 | 0.895 | 0.893 | 0.919 | 0.921 | 0.920 | 0.958 | 0.939 | 0.930 |
| | 192 | 1.253 | 1.021 | 0.910 | 0.920 | 0.956 | 0.957 | 0.953 | 0.987 | 0.971 | 0.964 |
| | 336 | 1.260 | 1.028 | 0.916 | **0.895** | 0.902 | 0.996 | 0.929 | 0.957 | 0.941 | 0.937 |
| | 720 | 1.285 | 1.060 | 0.968 | 0.949 | 0.986 | 1.056 | 0.998 | 0.995 | 0.985 | 0.984 |
| Weather | 96 | 1.013 | 0.814 | 0.800 | 0.752 | 0.760 | 0.841 | 0.790 | 0.766 | 0.754 | 0.741 |
| | 192 | 1.021 | 0.867 | 0.861 | 0.798 | 0.832 | 0.908 | 0.854 | 0.913 | 0.903 | 0.899 |
| | 336 | 1.043 | 0.872 | 0.865 | 0.828 | 0.857 | 0.905 | 0.884 | 0.901 | 0.918 | 0.903 |
| | 720 | 1.096 | 0.917 | 0.938 | 0.867 | 0.895 | 0.956 | 0.918 | 0.942 | 0.940 | 0.939 |

| Datasets | Horizon | Concept-aware models | | | | | | | | |
|---|---|---|---|---|---|---|---|---|---|---|
| | | MSGARCH | SD-Markov | OrbitMap | Cogra | FEDformer | OneNet | FSNet | Drift2Matrix | Auto-D2M |
| SyD | 78 | 1.264 | 0.936 | 0.635 | 1.251 | 1.260 | 0.317 | 0.433 | **0.315** | *0.313* |
| MSP | 31 | 2.641 | 3.234 | 1.244 | 2.898 | 2.849 | 0.751 | 1.148 | **0.663** | *0.659* |
| ELD | 227 | 2.425 | 2.439 | 1.835 | 2.587 | 2.635 | **1.101** | 1.425 | 1.644 | *1.669* |
| CCD | 583 | 5.712 | 3.462 | 1.753 | 3.604 | 3.616 | **1.298** | 1.678 | 1.387 | *1.392* |
| EQD | 50 | 4.213 | 3.573 | 1.386 | 3.949 | 3.944 | **1.386** | 1.938 | 1.392 | *1.388* |
| EOG | 183 | 3.566 | 3.571 | 3.251 | 4.067 | 4.013 | 1.337 | 2.044 | **1.198** | *1.191* |
| RDS | 69 | 5.924 | 4.587 | 4.571 | 5.135 | 5.779 | 1.865 | 2.546 | **1.198** | *1.689* |
| Stock1 ($\times10^{-2}$) | 17 | 2.366 | 2.146 | 1.003 | 2.137 | 2.258 | 0.923 | 0.953 | **0.878** | *0.902* |
| Stock2 ($\times10^{-2}$) | 11 | 2.129 | 1.669 | 0.747 | 1.367 | 1.352 | 0.312 | 0.477 | **0.303** | *0.317* |
| ETTh1 | 96 | 1.073 | 1.025 | **0.909** | 0.909 | 0.928 | 0.916 | 0.928 | 0.913 | *0.907* |
| | 192 | 1.092 | 1.037 | 0.991 | 0.996 | 1.034 | **0.975** | 0.995 | 0.979 | *0.977* |
| | 336 | 1.123 | 1.094 | 1.039 | 1.041 | 1.102 | 1.028 | 1.045 | **1.018** | *1.015* |
| | 720 | 1.146 | 1.108 | 1.083 | 1.095 | 1.122 | 1.082 | 1.102 | **1.073** | *1.085* |
| ETTh2 | 96 | 0.974 | 0.956 | 0.894 | 0.901 | 0.997 | 0.889 | 0.909 | **0.885** | *0.879* |
| | 192 | 0.952 | 0.935 | 0.976 | 0.987 | 0.993 | **0.968** | 0.971 | 0.977 | *0.970* |
| | 336 | 1.094 | 1.009 | 1.052 | 1.065 | 1.072 | 1.039 | **1.019** | 1.044 | *1.037* |
| | 720 | 1.148 | 1.131 | 1.120 | 1.131 | 1.119 | 1.135 | 1.147 | 1.115 | *1.121* |
| ETTm1 | 96 | 0.832 | 0.803 | 0.778 | 0.780 | 0.796 | **0.777** | 0.788 | 0.781 | *0.777* |
| | 192 | 0.854 | 0.814 | 0.810 | 0.819 | 0.827 | 0.813 | 0.842 | **0.805** | *0.801* |
| | 336 | 0.895 | 0.857 | 0.820 | 0.838 | 0.857 | **0.819** | 0.898 | 0.822 | *0.819* |
| | 720 | 0.919 | 0.895 | 0.868 | 0.890 | 0.899 | 0.868 | 0.964 | **0.864** | *0.854* |
| ETTm2 | 96 | 0.958 | 0.864 | 0.821 | 0.824 | 0.830 | 0.812 | 0.832 | 0.810 | *0.802* |
| | 192 | 0.982 | 0.885 | 0.832 | 0.849 | 0.858 | 0.830 | 0.854 | **0.825** | *0.824* |
| | 336 | 1.003 | 0.977 | 0.842 | 0.854 | 0.870 | **0.841** | 0.864 | 0.847 | *0.839* |
| | 720 | 1.134 | 1.021 | 0.906 | 0.921 | 0.935 | 0.896 | 0.901 | **0.886** | *0.876* |
| Traffic | 96 | 1.224 | 1.058 | 0.883 | 0.898 | 0.906 | 0.884 | 0.895 | **0.880** | *0.874* |
| | 192 | 1.242 | 1.083 | 0.895 | 0.908 | 0.925 | **0.883** | 0.905 | 0.888 | *0.880* |
| | 336 | 1.341 | 1.189 | 0.908 | 0.922 | 0.924 | 0.901 | 0.929 | 0.937 | *0.926* |
| | 720 | 1.337 | 1.201 | 0.946 | 0.964 | 0.969 | 0.940 | 0.946 | **0.932** | *0.923* |
| Weather | 96 | 0.974 | 0.951 | 0.744 | 0.759 | 0.785 | 0.745 | 0.775 | **0.737** | *0.737* |
| | 192 | 0.999 | 0.982 | 0.775 | 0.793 | 0.802 | 0.776 | 0.794 | **0.771** | *0.769* |
| | 336 | 1.028 | 1.009 | 0.806 | 0.825 | 0.869 | 0.801 | 0.804 | **0.791** | *0.786* |
| | 720 | 1.093 | 1.027 | 0.841 | 0.863 | 0.871 | **0.833** | 0.864 | 0.840 | *0.832* |

While forecasting series task is not our main focus, we provide a comparison of Drift2Matrix with other models. Results for the extended Auto-D2M, are included but not part of the comparison.

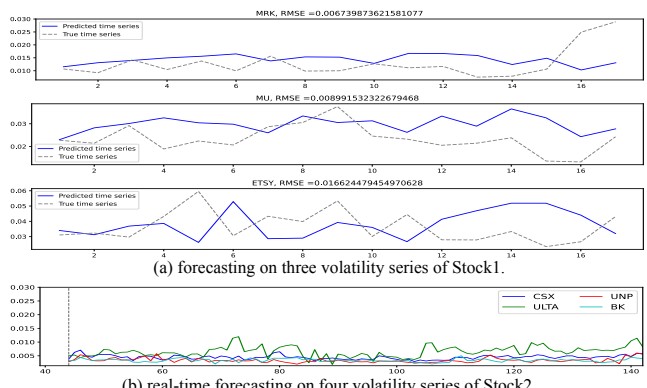

(a) forecasting on three volatility series of Stock1.

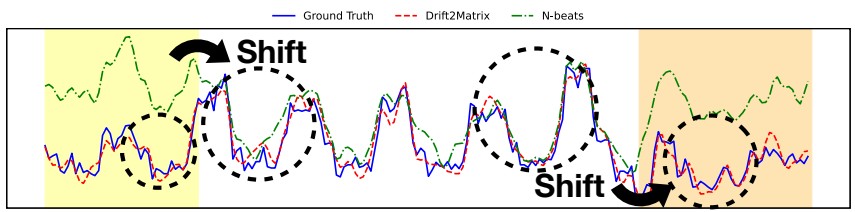

(b) real-time forecasting on four volatility series of Stock2.

Figure 10: Results of N-BEATS model. (a) forecasting on three series of `Stock1`. (b) online forecasting on four series of `Stock2`. Please also see our results shown in Fig 2(c) and Fig 3(a).

Figure 11: Visualization of N-BEATS and Drift2Matrix on `ETTh1`.

### H.4 Forecasting results of N-BEATS on Stock1, Stock2, ETTm2

The forecasting results of N-BEATS are depicted in Fig. 10. Compared to our forecasted result, shown in Fig 2(c) and Fig. 3, N-BEATS falls short in capturing complex concept transitions within multiple time series, revealing the limitations of a model geared solely for single time series forecasting. Additionally, Fig. 11 illustrates visualizations of N-BEATS and Drift2Matrix on the ETTh1 dataset, highlighting the strengths of Drift2Matrix in handling co-evolving time series with concept drift.

### H.5 Case Study of Kernel Representation on Motion Segmentation sequences

We apply the proposed method to motion segmentation on the Hopkins155 database. Hopkins155 is a standard motion segmentation dataset consisting of 155 sequences with two or three motions. These sequences can be divided into three concepts, i.e., indoor checkerboard sequences (104 sequences), outdoor traffic sequences (38 sequences), and articulated/nonrigid sequences (13 sequences). This dataset provides ground-truth motion labels and outlier-free feature trajectories (x-, y-coordinates) across the frames with moderate noise. The number of feature trajectories per sequence ranges from 39 to 556, and the number of frames from 15 to 100. Under the affine camera model, the trajectories of one motion lie on an affine subspace of dimensions up to three.

Fig. 13 shows the results on the four random sequences – i.e., pepople1, cars10, 1R2TCR, and 2T3RTCR, our kernel-induced representation achieves good results on imbalance-concept sequence (people 1) and performs well on noticeable perspective distorted sequences (1R2TCR and 2T3RTCR). In 1R2TCR and 2T3RTCR, the camera often has some degree of perspective distortion so that the affine camera assumption does not hold; in this case, the trajectories of one motion lie in a nonlinear subspace.

### H.6 Distribution of RMSE Values Across all Datasets

Fig. 12 illustrates the distribution of RMSE values for each model across all datasets, and it is evident that our model achieves the most favourable outcomes overall.

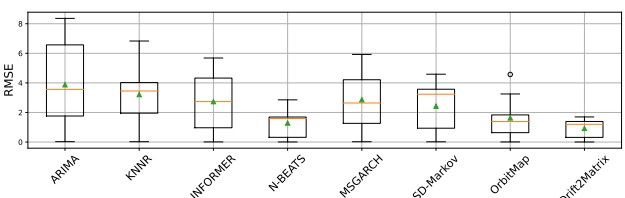

Figure 12: Box Plot of RMSE Values for Each Model Across Datasets

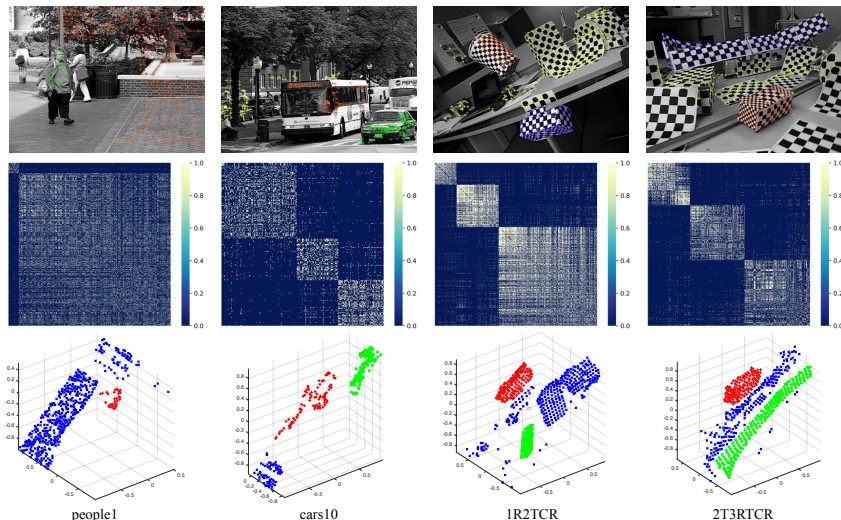

Figure 13: Results on random four sequences (pepople1, cars10, 1R2TCR, 2T3RTCR) of Hopkins155 database. The top row shows images from the four sequences with superimposed tracked points. The second row is the heatmap of representation matrices yielded by Drift2Matrix. The bottom row is a projection of the representation results into a 3D space.

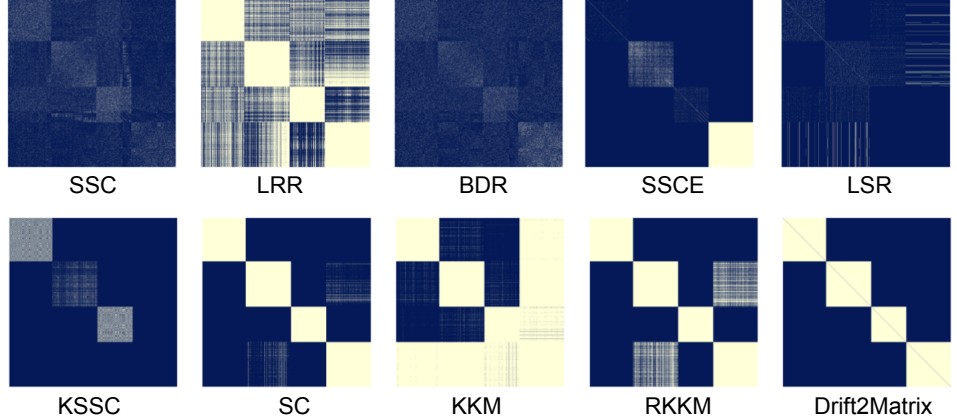

Figure 14: The heatmap of representation matrices (binarized) learned on SyD by various SOTA and Drift2Matrix.

## H.7 COMPLEXITY ANALYSIS AND EXECUTION TIME EVALUATION

Solving the optimization problem involves iterative updates to $\mathbf{W}$, $\mathbf{V}$, and $\mathbf{Z}$. The update for $\mathbf{W}$ requires computing the smallest $k$ eigenvalues and eigenvectors of a matrix, with a complexity of $\mathcal{O}(kn^2)$. Updating $\mathbf{Z}$ involves matrix addition, transposition, and element-wise truncation, resulting in a complexity of $\mathcal{O}(n^2)$. The update for $\mathbf{V}$, which involves matrix inversion $\left((\mathcal{K} + \beta \mathbf{I})^{-1}\right)$,

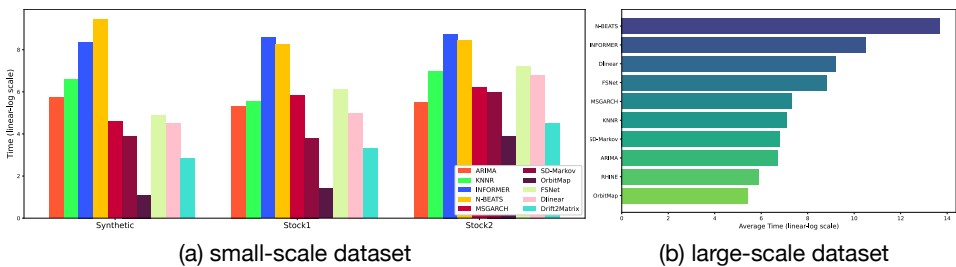

Figure 15: Computation time on small/large scale series datasets.

dominates the computation with a complexity of $\mathcal{O}(n^3)$. Over $T$ iterations, the total computational complexity is $\mathcal{O}(Tn^3)$, with matrix inversion being the primary bottleneck. The space complexity is $\mathcal{O}(n^2)$ due to the kernel matrix.

To improve scalability, we employ low-rank kernel approximations, reducing the complexity to $\mathcal{O}(Tn^2d)$, where $d \ll n$. Parallelization and efficient inversion techniques, such as the Woodbury formula, further enhance computational efficiency.

Additionally, we further reduce complexity by employing the Nyström method in combination with self-representation learning. A straightforward approach selects a smaller subset of time series, specifically $k$ concept prototypes, by choosing $m$ representative time series from the dataset. Based on self-representation learning principles, these concept prototypes are expressed as a linear combination of all time series, residing within the subspace spanned by the dataset $\mathbf{S}$. To avoid computing the full kernel matrix, the solution is restricted to a smaller subspace $\widehat{\mathbf{S}} \subset \mathbf{S}$, satisfying two criteria: (1) $\widehat{\mathbf{S}}$ must be small enough for computational efficiency, and (2) it must sufficiently cover the dataset to minimize approximation errors.

Building on kernel-induced representation learning, we use the initial representation matrix to identify concepts efficiently. Concept prototypes and their associated time series are selected from these concepts to construct $\widehat{\mathbf{S}}$. With $\widehat{\mathbf{S}}$ containing $m(m \ll n)$ time series, we define two kernel matrices: $\widehat{\mathcal{K}} \in \mathbb{R}^{m \times m}$ for kernel similarities among the $m$ selected time series, and $\widetilde{\mathcal{K}} \in \mathbb{R}^{n \times m}$ for similarities between the entire dataset and the selected time series. Using the Nyström method, the full kernel matrix $\mathcal{K}$ is approximated as $\mathcal{K}^* \approx \widetilde{\mathcal{K}} \widehat{\mathcal{K}}^{-1} \widetilde{\mathcal{K}}^{\mathrm{T}}$. This reduces computational costs significantly as only $\widetilde{\mathcal{K}}$ needs computation ($\widehat{\mathcal{K}}$ is a subset of $\widetilde{\mathcal{K}}$). This approach provides an efficient and scalable solution for large-scale co-evolving time series.

Fig. 15 presents the execution time comparison on small-scale datasets (with short sequence lengths: `SyD`, `Stock1`, `Stock2`) and the average runtime on large-scale datasets (with long sequence lengths: `ETTh1`, `ETTh2`, `ETTm1`, `ETTm2`, `Traffic`, `Weather`). Note that the vertical axis uses a linear-log scale.

## H.8 ABLATION STUDY

### H.8.1 ABLATION STUDY ON THE NUMBER OF SERIES

Fig. 16 illustrates the effect of the number of series on model performance. For datasets with a smaller number of series, the experimental results remain relatively stable. In contrast, for datasets with a larger number of series, a significant reduction in the series count leads to a decline in accuracy, indicating that fewer nonlinear inter-series correlations are available for the model to leverage. For the EOG dataset, the RMSE initially decreases before increasing. This suggests that the initial reduction in series eliminates redundant information, improving performance. However, as the reduction continues, the loss of critical information ultimately degrades the model's accuracy.

### H.8.2 ABLATION STUDY ON REGULARIZATIONS

We conduct ablation experiments to validate the efficacy of Drift2Matrix's kernel representation learning. We focus particularly on the regularization and kernelization techniques employed in Eq. 3. Our approach is compared against existing self-representation learning/subspace clustering methods

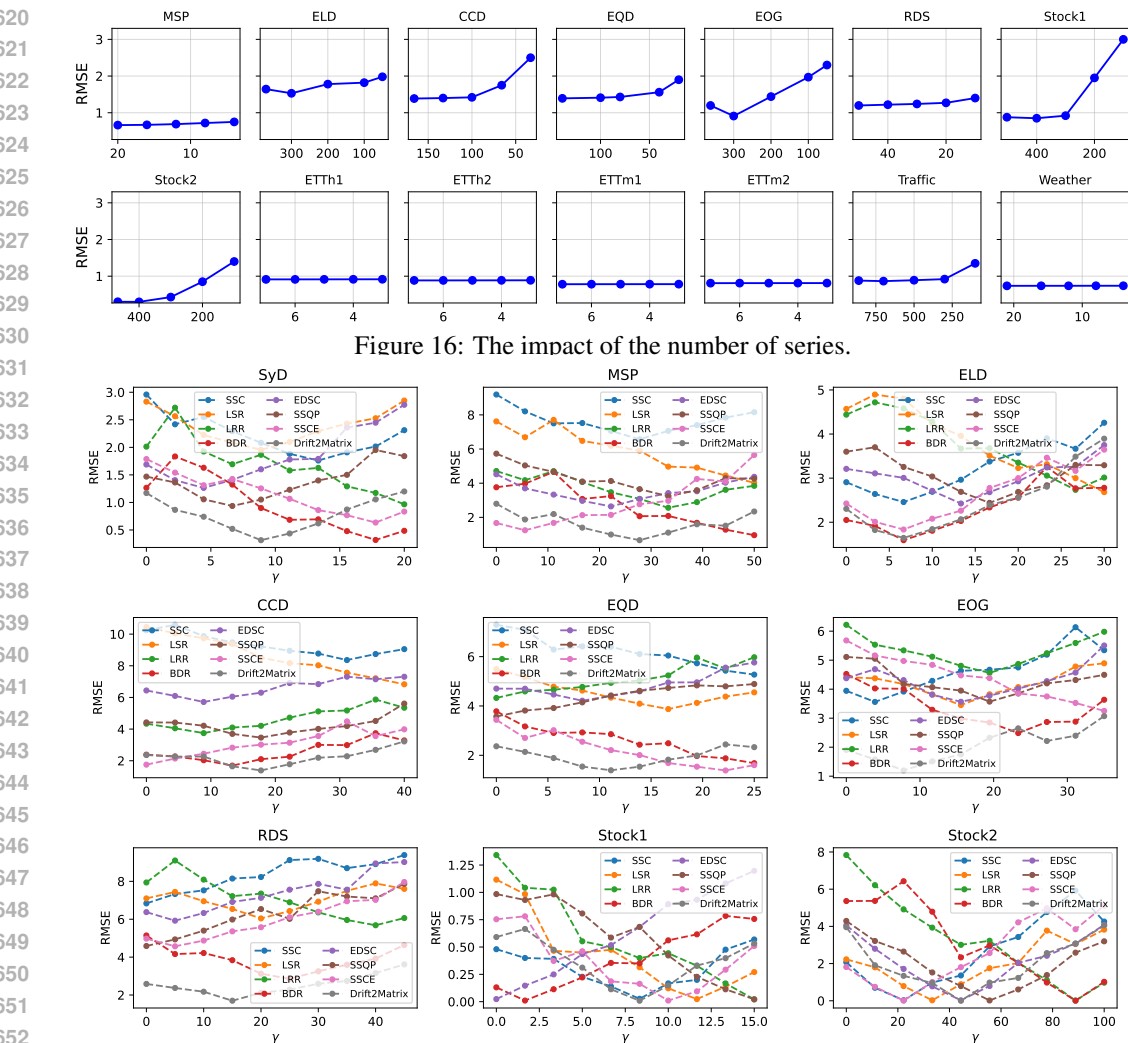

Figure 16: The impact of the number of series.

Figure 17: RMSE across datasets for different regularizations at varying $\gamma$

Lu et al. (2018a); Bai & Liang (2020); Elhamifar & Vidal (2013); Ji et al. (2014); Liu et al. (2012) - SSC, LSR, LRR, BDR, EDSC, SSQP, and SSCE, under varying values of $\gamma$. The comparative results are illustrated in Fig. 17. This comparison clearly demonstrates that our model achieves superior performance, outperforming the other methods in terms of RMSE across different datasets for a range of $\gamma$ values.

### H.8.3 ABLATION STUDY ON KERNEL-BASED METHODS

We also extend ablation experiments to other kernel-based method to validate the representation capability of Drift2Matrix. We present the representation (binarized version) produced by Drift2Matrix and other SOTA methods on the predicted window of SyD in Fig. 14. Drift2Matrix yields a block diagonal matrix with dense within-cluster scatter and sparse between-cluster separation, revealing the underlying cluster structure. SSC, BDR, SSCE, and LSR perform poorly (i.e., unclearly identified the strong and weak correlations) when the subspaces are nonlinear or overlap. LRR gets improved for those weakly-correlated points (in the light area) but still cannot accurately predict the representation for highly-correlated points (in the dark area), making the cluster undistinguished from each other (Note that undistinguished clusters can lead to bad representations). Although kernel-based methods are adept at handling nonlinear data, they are helpless in the case of potentially locally manifold structures, e.g., SC, KKM, and RKKM fail to distinguish the second cluster from the fourth cluster, and KSSC only finds three clusters.

Table 8: Ablation study of Drift2Matrix with different kernel functions, in terms of RMSE, for the nine datasets

| Dataset | Drift2Matrix (Linear) | Drift2Matrix (Polynomial) | Drift2Matrix (Sigmoid) | Drift2Matrix |
|---------|----------------------|---------------------------|------------------------|--------------|
| SyD | 1.324 | 1.325 | 0.638 | **0.315** |
| MSP | 3.673 | 2.975 | 1.382 | **0.663** |
| ELD | 1.853 | 2.655 | 2.903 | **1.644** |
| CCD | 4.390 | 3.395 | 2.429 | **1.387** |
| EQD | 5.409 | 4.405 | 4.015 | **1.392** |
| EOG | 3.200 | 3.205 | 2.517 | **1.198** |
| RDS | 6.705 | 4.710 | 3.715 | **1.699** |
| Stock1 | $2.103 \times 10^{-2}$ | $2.056 \times 10^{-2}$ | $9.905 \times 10^{-3}$ | $\mathbf{8.780 \times 10^{-3}}$ |
| Stock2 | $2.650 \times 10^{-2}$ | $2.061 \times 10^{-2}$ | $6.073 \times 10^{-3}$ | $\mathbf{3.032 \times 10^{-3}}$ |

### H.8.4 ABLATION STUDY ON KERNEL FUNCTIONS

In this section, we present an ablation study conducted to evaluate the impact of different kernel functions on the performance of the Drift2Matrix model. This study aims to ascertain the effectiveness of various kernels in capturing nonlinear relationships in time series data.

We explore three different kernel functions in our ablation study: Linear, Polynomial, Sigmoid. These kernels are chosen for their distinct properties in mapping data to higher-dimensional spaces. Each kernel function offers different properties and captures various aspects of the data. The linear kernel is straightforward and effective for linear relationships, while the polynomial kernel can model more complex, non-linear interactions. The sigmoid kernel, inspired by neural networks, and the Gaussian kernel, a popular choice for capturing the locality in data, add more flexibility to the model.

- **Linear Kernel**: Simple yet effective for linear relationships, represented as

$$\mathcal{K}(\mathbf{S}, \mathbf{S}) = \mathbf{S}^\top \mathbf{S} \tag{26}$$

- **Polynomial Kernel**: Captures complex, non-linear interactions, given by

$$\mathcal{K}(\mathbf{S}, \mathbf{S}) = (\mathbf{S}^\top \mathbf{S} + c)^d \tag{27}$$

    where $c$ is a constant and $d$ is the degree of the polynomial.

- **Sigmoid Kernel**: Inspired by neural networks, takes the form of

$$\mathcal{K}(\mathbf{S}, \mathbf{S}') = \tanh(\xi \mathbf{S}^\top \mathbf{S}' + c) \tag{28}$$

    where $\xi$ and $c$ are the parameters of the sigmoid function.

In our experiments, the parameters for each kernel function were tuned to optimize the model's performance. For the Polynomial and Sigmoid kernels, we varied the degree $d$ and the constants $\xi$ and $c$ to explore their effects on the model's forecasting accuracy.

As shown in Table 8, the Drift2Matrix model with the Gaussian kernel achieves the best performance across all datasets, indicating its effectiveness in capturing the complex nonlinear relationships in time series data. The ablation study highlights the Gaussian kernel's ability to adaptively handle various types of data distributions, making it a versatile choice for time series analysis. However, the choice of the kernel function may depend on the specific characteristics of the dataset, and therefore, a careful consideration of kernel properties is necessary for optimal model performance. Besides, from a technical perspective, multiple kernel learning (MKL) offers a way to learn an appropriate consensus kernel by combining several predefined kernel matrices, thus integrating complementary information and identifying a suitable kernel for the given task, i.e.,

$$\mathcal{K} = \sum_{m=1}^{M} \beta_m \mathcal{K}_m, \quad \text{subject to} \quad \sum_{m=1}^{M} \beta_m = 1$$

where $\mathcal{K}_m$ represents individual kernels and $\beta_m$ are the non-negative weights that sum to 1. This formulation allows MKL to find an optimal combination of kernels.

