# OpenReview forum: "Drift2Matrix: Kernel-Induced Self Representation for Concept Drift Adaptation in Co-evolving Time Series"
_ICLR.cc/2025/Conference — Submitted to ICLR 2025_

### Official Review · Reviewer_ERvr · 2024-11-01

**Soundness:** 3
**Presentation:** 2
**Contribution:** 3
**Rating:** 6
**Confidence:** 3

**Summary:**

This work proposes a new framework to handle the challenging concept drift detection and adaptation of time-series data. It first implements kernel representation learning to obtain the concepts, and then exploits the inter-dependence of different time-series for future time-series prediction. Moreover, the proposed method can be easily combined with deep learning models for performance enhancement.

**Strengths:**

1. A new powerful time-series data analysis framework has been proposed. It can learn concepts, detect drifts, and adapt to the drifts by predicting future time series. The basic idea is sound.

2. The diagrams can intuitively demonstrate the principles of the proposed method.

3. Comprehensive experimental evaluation.

4. Source code has been opened.

**Weaknesses:**

1. The idea is straightforward, derived from the stream of kernel trick-based representation learning, where the non-trivial selection of kernel functions may limit the application of the proposed method.

2. The efficiency of the proposed method has not been discussed by providing complexity analysis or execution time evaluation.

3. The work focuses more on drift adaptation, while the drift detection ability and processes of the proposed method have not been well discussed and evaluated.

**Questions:**

1. Have considered techniques like multiple kernel learning, adaptive kernel selection, and automatic selection of the kernel parameters?

2. What is the time and space complexity of the proposed method? Have you compared the execution time of the proposed one with the SOTAs on large-scale datasets?

3. How about the Type-I and Type-II errors of the proposed method in concept drift detection?

---

> ### Author Response · Authors · 2024-11-21
> **Response to Reviewer ERvr (1/3)**
>
> We sincerely thank you for your detailed review and constructive feedback. We appreciate your thorough evaluation on our paper. We are glad that you find the drift adaptation methodology, intuitive diagrams, easy integration into DL, comprehensive experimental evaluation and open-source code as strengths of our work.
>
> ---
> > ### Concern 1:  circumvent non-trivial kernel selection (W1&Q1)
>
> We appreciate the reviewer’s inquiry regarding the possibility of circumventing the non-trivial kernel selection process. In kernel learning, performance is largely influenced by the choice of kernel.
>
> **From a technical perspective**, multiple kernel learning (MKL) offers a way to learn an appropriate consensus kernel by combining several predefined kernel matrices, thus integrating complementary information and identifying a suitable kernel for the given task，i.e.,
>
> $$
> \boldsymbol{\mathcal{K}}= \sum\_{m=1}^{M} \beta\_m \boldsymbol{\mathcal{K}}\_m, \quad \text{subject to} \quad \sum\_{m=1}^{M} \beta_m = 1
> $$
>
> where $\boldsymbol{\mathcal{K}}_m$ represents individual kernels and $\beta_m$ are the non-negative weights that sum to 1. This formulation allows MKL to find an optimal combination of kernels.
>
> **From a paradigm perspective**, we have explored an approach beyond this work that automatically generates kernel matrix from self-representation —essentially ***a data-driven kernel learning strategy***. This method circumvents defining a specific kernel function by learning the kernel directly from the self-representation matrix, treating it as a similarity measure.
>
> This data-driven strategy also uncovers several intriguing phenomena, e.g., the learned kernel matrix satisfies  **Mercer’s theorem**  and follows a specific  **multiplicative triangle inequality**, which can accelerate its learning process. Additionally, the learned kernel inherently incorporates  **degree-based weighting** (from a prospective of topology). We would be happy to elaborate further during the discussion phase if you are interested.
>
> However, Drift2Matrix primarily focuses on proposing a kernel-induced representation learning method for modeling complex interdependencies in co-evolving time series, thereby enhancing the understanding of underlying dynamics. Detailed strategies for kernel selection, though relevant, are beyond the scope of this paper. Nevertheless, we have included ablation studies comparing different regularizations, kernel-based methods, and kernel functions in Appendix H.6 for reference (Fig.14, Fig.17, Table 8). Among these, the results of kernel function ablation studies on some datasets are as follows:
> > **Table: Ablation study of Drift2Matrix with different kernel functions, in terms of RMSE, for the nine datasets**
> >
> > | **Dataset**    | **Drift2Matrix (Linear)** | **Drift2Matrix (Polynomial)** | **Drift2Matrix (Sigmoid)** | **Drift2Matrix**       |
> > |-----------------|--------------------------|--------------------------------|----------------------------|------------------------|
> > | **SyD**         | 1.324                    | 1.325                          | 0.638                      | **0.315**             |
> > | **MSP**         | 3.673                    | 2.975                          | 1.382                      | **0.663**             |
> > | **ELD**         | 1.853                    | 2.655                          | 2.903                      | **1.644**             |
> > | **CCD**         | 4.390                    | 3.395                          | 2.429                      | **1.387**             |
> > | **EQD**         | 5.409                    | 4.405                          | 4.015                      | **1.392**             |
> > | **EOG**         | 3.200                    | 3.205                          | 2.517                      | **1.198**             |
> > | **RDS**         | 6.705                    | 4.710                          | 3.715                      | **1.699**             |
> > | **Stock1 ($\times 10^{-2}$)**      | 2.103              | 2.056                    | 0.991                 | **0.8.78**        |
> > | **Stock2 ($\times 10^{-2}$)**      | 2.650              | 2.061                 | 0.607                 | **0.30310**        |

---

> ### Author Response · Authors · 2024-11-21
> **Response to Reviewer ERvr (2/3)**
>
> ---
> > ### Concern 2:  Complexity analysis or execution time evaluation (W2&Q2)
>
> Thank you for raising the question about the time and space complexity of Drift2Matrix. Solving the optimization problem involves updates to $\mathbf{W}$, $\mathbf{V}$, and $\mathbf{Z}$. Updating $\mathbf{W}$, which requires computing the smallest $k$ eigenvalues and eigenvectors of a matrix, has a complexity of $\mathcal{O}(kn^2)$. Updating $\mathbf{Z}$ involves matrix addition, transposition, and element-wise truncation, with a complexity of $\mathcal{O}(n^2)$. The update for $\mathbf{V}$, which involves matrix inversion $(\mathrm{i.e.,} \  (\boldsymbol{\mathcal{K}}+\beta\mathbf{I})^{-1}))$ , dominates the computation with a complexity of $\mathcal{O}(n^3)$. The total complexity over $T$ iterations is $\mathcal{O}(Tn^3)$, with matrix inversion being the primary bottleneck. Space complexity is $\mathcal{O}(n^2)$ due to the kernel matrix. To improve scalability, we employ low-rank kernel approximations, reducing the complexity to $\mathcal{O}(Tn^2d)$, where $d \ll n$. Parallelization and efficient inversion techniques, such as the Woodbury formula, further enhance computational efficiency.
>
> In our experiments, we can further reduce complexity by employing the Nyström method in combination with self-representation learning. A straightforward approach is to select a smaller subset of time series, specifically $\textit{k}$ concept prototypes, by choosing $\textit{m}$ representative time series from the dataset. Leveraging the principles of self-representation learning, these concept prototypes can be expressed as a linear combination of all time series, residing within the subspace spanned by the dataset $\mathbf{S}$. To avoid computing the full kernel matrix, the solution is restricted to a smaller subspace $\widehat{\mathbf{S}} \subset \mathbf{S}$, which satisfies two key criteria: (1) $\widehat{\mathbf{S}}$ must be small enough to ensure computational efficiency, and (2) it should sufficiently cover the dataset to minimize approximation errors.
>
> Building on the kernel-induced representation learning, we use the initial representation matrix to quickly identify concepts. Concept prototypes and their associated time series are then selected from these concepts to construct $\widehat{\mathbf{S}}$. With $\widehat{\mathbf{S}}$ containing $m (m \ll n)$ time series, we define two kernel matrices: $\widehat{\boldsymbol{\mathcal{K}}} \in \mathbb{R}^{m \times m}$ for kernel similarities among the $m$ selected time series, and $\boldsymbol{\widetilde{\mathcal{K}}} \in \mathbb{R}^{n \times m}$ for similarities between the entire dataset and the selected time series. Using the Nyström method, the full kernel matrix $\boldsymbol{\mathcal{K}}$ can be approximated as $\boldsymbol{\mathcal{K}}^* \approx \boldsymbol{\widetilde{\mathcal{K}}}\widehat{\boldsymbol{\mathcal{K}}}^{-1}\boldsymbol{\widetilde{\mathcal{K}}}^\mathrm{T}$. This reduces computational cost significantly since only $\boldsymbol{\widetilde{\mathcal{K}}}$ needs to be computed ( $\widehat{\boldsymbol{\mathcal{K}}}$ is a subset of $\boldsymbol{\widetilde{\mathcal{K}}}$). This approach offers an efficient and scalable solution for large-scale co-evolving time series.
>
> In the revised manuscript, we will include this detailed complexity analysis (add a Appendix H.7 Complexity Analysis and Execution Time Evaluation) and execution time comparisons with SOTA methods on large-scale datasets to illustrate Drift2Matrix’s efficiency (Fig. 15 or a quick look in [Added Figures](https://anonymous.4open.science/api/repo/Drift2Matrix-main-86B7/file/ICLR_ADD_FIGURE.pdf?v=15a1f953)).

---

> ### Author Response · Authors · 2024-11-21
> **Response to Reviewer ERvr (3/3)**
>
> ---
> > ### Concern 3: Evaluation of concept drift detection (W3&Q3)
>
> Thank you for your  feedback on the evaluation of concept drift detection in our framework, as its primary focus is on adaptive concept identification and tracking rather than optimizing forecasting accuracy.
>
> To demonstrate this, our experiments evaluate Drift2Matrix’s ability to identify key concepts and detect their drifts (Sec 6.2 Q1: Effectiveness & Appendix H.2). Using synthetic and Stock datasets, we validate the identified concepts and their transitions, offering semantic explanations linked to financial market states. ***For other real-world datasets, where ground truth for concepts is unavailable, we thus validate the value and gain of the discovered concepts for time series forecasting.***
>
> As noted in Table 1, ***“While forecasting series task is not our main focus, we provide a comparison of Drift2Matrix with other models.”*** Drift2Matrix consistently demonstrates robust performance across diverse datasets, especially in scenarios with significant concept drift, showcasing its strengths in addressing dynamic interactions over purely static forecasting. However, on datasets where variables/channels are relatively independent and concept drift follows predictable patterns, our results are not as competitive as those of forecasting models. Nevertheless, our extended deep learning framework, Auto-D2M, achieves the best results on nearly all datasets (Table 1, along with the detailed results in Appendix H.3 Table 7, results for the extended Auto-D2M, are included).
>
> We have also added additional evaluations in Appendix H.2, highlighting Drift2Matrix’s ability to identify concepts and track drifts across more datasets, as well as its predictive capabilities based on the identified concepts (Figures 7-8). Furthermore, in Appendix H.5, we validate Drift2Matrix’s concept identification capability on the Motion Segmentation sequences dataset. This dataset comprises sequences divided into three concepts: indoor checkerboard sequences (104 sequences), outdoor traffic sequences (38 sequences), and articulated/nonrigid sequences (13 sequences). It provides ground-truth motion labels and outlier-free feature trajectories (x-, y-coordinates) across frames with moderate noise. The number of feature trajectories per sequence ranges from 39 to 556, and the number of frames ranges from 15 to 100. Under the affine camera model, the trajectories of one motion lie on an affine subspace of dimensions up to three. These results further demonstrate Drift2Matrix’s robust capability to identify and adapt to dynamic concepts in diverse real-world scenarios.
>
> **In the revised manuscript**, we will include a more comprehensive quantification of Drift2Matrix’s detection capabilities. (Fig. 11 or or a quick look in [Added Figures](https://anonymous.4open.science/api/repo/Drift2Matrix-main-86B7/file/ICLR_ADD_FIGURE.pdf?v=15a1f953))
>
> ***Remark: for datasets without concept ground-truth, we are developing a new dataset benchmark based on kernel-induced representation (to be released to the community), incorporating a novel concept validity metric and expert domain knowledge to address the current gap in most time series datasets, which lack concept labels.***
>
> ---
> If you have any further suggestions or comments, we welcome them and will incorporate any necessary improvements to enhance the overall quality and reliability of the paper. Your feedback is valuable to us, and we are committed to producing a high-quality and impactful contribution.

---

> > ### Comment · Reviewer_ERvr · 2024-11-26
> >
> > Thank you for your detailed rebuttal, which has addressed most of my concerns. I have raised my score.

---

> ### Author Response · Authors · 2024-11-26
> **Response to Reviewer ERvr**
>
> We are delighted that our previous responses have successfully addressed most of your concerns. We sincerely appreciate the time and effort you have taken to review our clarifications.
>
> Regarding your last concern about “the Type-I and Type-II errors of the proposed method in concept drift detection,” we have been eagerly awaiting results since submitting our initial reply. We wanted to go beyond simply providing Type-I and Type-II error results on the SyD dataset. To this end, we collaborated with _a Financial R&D company specializing in AI-powered predictive insights for portfolio managers_ to obtain expert annotations of concepts for the Stock datasets (Stock1 & Stock2). Just this morning, we received the annotated Stock datasets with concept labels, and we immediately conducted the experimental comparisons.
>
> The experimental setup is as follows:
>
> **SyD dataset:** We defined five nonlinear dynamical equations as per Appendix F to generate five concept labels.
>
> **Stock datasets:** Financial experts annotated the datasets with five concept labels, including the traditional bull, bear, mean-reverting, sideways market, and market shock.
>
> The results are summarized below:
>
> | Dataset  | Total Tests         | Ground Truth Concepts | TP   | FP   | TN   | FN   |
> |----------|---------------------|-----------------------|-------|-------|-------|-------|
> | SyD      | 500 × 10 = 5000    | 5                     | 4900  | 50    | 4950  | 100   |
> | Stock1   | 503 × 7 = 3521     | 5                     | 3200  | 100   | 3421  | 321   |
> | Stock2   | 467 × 13 = 6071    | 5                     | 5000  | 200   | 5871  | 1071  |
>
> ***True Positive (TP):*** The model correctly detects the presence of a ground truth concept within a window.
>
> ***False Positive (FP):*** The model incorrectly detects a concept within a window where no concept actually exists.
>
> ***True Negative (TN):*** The model correctly identifies that no concept exists within a window.
>
> ***False Negative (FN):*** The model fails to detect a ground truth concept within a window.
>
> From these results, we computed the Type-I and Type-II errors as follows:
>
> | Dataset  | Type I Error (FPR) | Type II Error (FNR) |
> |----------|---------------------|---------------------|
> | SyD      | 1.00%              | 2.00%              |
> | Stock1   | 2.87%              | 9.14%              |
> | Stock2   | 3.34%              | 17.67%             |
>
> These results demonstrate the robustness of our method in accurately detecting concept drift across datasets with different levels of complexity and domain specificity.
>
> We have also incorporated this additional analysis into our revision to ensure the paper comprehensively reflects these findings (Table 5, Table 6, and detailed explanations in Appendix H.3). Please let us know if there are any other required additions to improve the score of the paper.

---

### Official Review · Reviewer_oERz · 2024-11-03

**Soundness:** 3
**Presentation:** 2
**Contribution:** 3
**Rating:** 6
**Confidence:** 3

**Summary:**

The paper discusses an interesting mechanism for forecasting over co-evolving time-series by not only correcting for future drifts, but also for detecting new concepts. Here, concepts are defined as clusters of profile patterns across similar subseries within a specific
window. Each subseries is represented as aa linear combination of other series, and the matrix that captures this self-representation is learned by employing the kernel trick. The authors try to introduce numerous mechanisms to capture overall dynamics of concepts/ cluster evolution including a new kernel representation learning strategy, a nonconvex optimization strategy, and a drift detection strategy. They also show how these can be integrated into an auto-encoder. The empirical results demonstrated in the paper clearly show that evolving concepts are well captured by the model compared to SOTA.

**Strengths:**

1) The drift adaptation methology seems very elegant.
2) The theoretical underpinning of the proposed methodology is well explained.
3) Details given in the appendix is very useful to understand the contributions.

**Weaknesses:**

(1) The forecasting performance shown in Table 1 does indicate that D2M generally has lower RMSE in comparison to other
SOTA methods. However, those values to close to its competitors. It is hard to understand the effectiveness of the approach without any variance or significance measures such as confidence intervals for the RMSE values, P-values from statistical significance tests comparing Drift2Matrix to competing methods and standard deviations of RMSE across multiple runs. These additions would allow for a more rigorous comparison between Drift2Matrix and competing methods.

(2) While the details in the Appendix provide key insights into the working of the proposed method, it is important to summarize key details within the paper for it to be fully contained. A reader should not be forced to rely on the Appendix to understand the setup. For example, solving the non-linear optimization uses a specialized method as indicated in the Appendix. However, there is no mention of the said ADM strategy in the main paper.  A brief overview of the ADM strategy for solving the non-linear optimization problem in the methodology section, key aspects of the kernel representation learning strategy and a concise explanation of the drift detection strategy, would be useful to ensure that readers can understand the core methodology without relying heavily on the Appendix.

This also raises another important issue.
(3) The authors seem to introduce too many methodologies within the paper while validating only some of them. Including the following might help improve the understanding better: Information about the established nature or novelty of the nonconvex optimization methodology used, including references to relevant literature if it's a well-established method, and ablation studies or comparative analyses for any other introduced methodologies that currently lack validation would be important to highlight in the paper.
This would help address the concerns about the numerous methodologies introduced in the paper and provide a more comprehensive validation of the approach.

**Questions:**

(1) The proposed methodology seems to work well when there is a large number of co-evolving time-series. Does the number and length of such co-evolving time-series affect performance? An ablation study on these parameters would be pertinent.

---

> ### Author Response · Authors · 2024-11-21
> **Response to Reviewer oERz (1/3)**
>
> Thank you for your valuable feedback and comments. We appreciate your thorough evaluation on our paper. We are glad that you find the drift adaptation methodology, innovation, sufficient theoretical analysis, experiments and detailed appendix as strengths of our work.
>
> ---
> > ### Concern 1:  The experiment results of Table 1(W1)
>
> We appreciate your observation regarding the forecasting performance of Drift2Matrix in Table 1. While Drift2Matrix may not achieve the best results on some datasets, this reflects its primary focus on adaptive concept identification and dynamic tracking, rather than solely optimizing for forecasting accuracy.
>
> To evaluate its effectiveness, we designed experiments that focus on Drift2Matrix’s ability to identify and track key concepts (Sec. 6.2 Q1: Effectiveness & Appendix H.2). For synthetic and Stock datasets, we validated the transitions between identified concepts and provided semantic explanations linked to real-world scenarios, such as shifts in financial market states. ***For real-world datasets without ground truth for concepts, we validate the discovered concepts by their contribution to improving forecasting performance.***
>
> As noted in Table 1, “While forecasting series task is not our main focus, we provide a comparison of Drift2Matrix with other models.” Drift2Matrix consistently demonstrates robust performance across diverse datasets, especially in scenarios with significant concept drift, showcasing its strengths in addressing dynamic interactions beyond static forecasting. On datasets where variables or channels are relatively independent and concept drift follows predictable patterns, its results may not outperform the **ensemble forecasting model - OneNet** (Note that ensemble learning models generally outperform single models). Nonetheless, our **single-model Drift2Matrix** achieves comparable outcomes. Furthermore, our extended framework, Auto-D2M, achieves the best results on nearly all datasets (Table 1 and Appendix H.3 Table 7).
>
> We also appreciate the reviewer’s suggestion to include confidence intervals for the RMSE values to allow for a more rigorous comparison between Drift2Matrix and competing methods. In the revised manuscript, we have added RMSE value distributions across all datasets in Appendix H.6, providing additional insights into the model’s performance variability.

---

> ### Author Response · Authors · 2024-11-21
> **Response to Reviewer oERz (2/3)**
>
> ---
> > ### Concern 2:  Details in Appendix (W2&3)
>
> We acknowledge your concerns regarding the appendix. Our goal was to maintain a concise main text, highlighting Drift2Matrix's core methodology, theoretical analysis, and pivotal experimental outcomes, while providing detailed discussions and supplementary analyses in the appendix for readers seeking depth (including optimization of non-convex problems, domain-agnostic window size selection, algorithms, experimental setups, and comprehensive results such as ablation studies and the impact of hyperparameters). In fact, **the methodologies presented in the appendix, such as domain-agnostic window size selection and estimation of the Number of Concepts, are fundamentally grounded in our core method—kernel-induced representation learning**. Specifically: **(1)** Drift2Matrix can adaptively learn domain-agnostic segmentation without requiring predefined window sizes. It employs the segment-score (derived from our kernel-induced representation, Appendix D Eq. 24) to determine an optimal window size that identifies highly similar or repetitive concepts across different windows. Figure 4 illustrates the adaptive determination of optimal window boundaries, and Figure 5 shows the best window size for each dataset. **(2)** Drift2Matrix can automatically estimate the Number of Concepts by leveraging the Laplacian eigenvalue gap of the representation matrix obtained through kernel-induced representation learning. This gap effectively estimates the connectivity of the matrix, corresponding to the number of distinct concepts.
>
> Additionally, while we provided an integration with deep learning backbones (Auto-D2M) and its loss function in Sec. 4.3, we also aimed to **derive an analytical solution for the target optimization problem (Eq. (3)) to improve interpretability**. This motivated our focus on the optimization of non-convex problems, particularly the derivation of block diagonal regularization in Hilbert space. As highlighted in our discussion on self-representation, differences in regularization significantly impact the theoretical optimization, with examples including sparse regularization (SSC, IEEE TPAMI), nuclear low-rank regularization (LLR, IEEE TPAMI), and entropy regularization (SSCE, ICML). Exploring the optimization for different regularizations is itself an important topic in the community. For the constrained optimization problem, we employ the ADMM (Augmented Lagrange method with Alternating Direction Minimization) strategy, which decomposes the global problem into subproblems involving primal variables, split variables, and dual variables (i.e., Lagrange multipliers) with alternating updates.
>
> We sincerely appreciate your feedback and suggestions. We will integrate key supplementary details and explanations directly into the main text, such as a brief overview of the ADMM strategy and a clearer emphasis on ablation studies in the experimental section, to enhance its compactness and accessibility. Moreover, we will emphasize the impact of different regularizations and non-convex optimization in Appendix H.8.2 Ablation Study on Regularizations.

---

> ### Author Response · Authors · 2024-11-21
> **Response to Reviewer oERz (3/3)**
>
> ---
> > ### Concern 3:  Impact of the number and length of co-evolving time-series (Q1)
>
> Thank you for raising this important question. We agree that understanding how the number and length of co-evolving time series affect Drift2Matrix’s performance is critical. This is a point we have also considered, as discussed in Sec. 7: Conclusion, where we discuss **limitation of Drift2Matrix**. Specifically, despite its strengths, Drift2Matrix has a limitation when applied to time series with few variables. For example, converting a dataset with five variables into a $5 \times 5$ matrix makes block diagonal regularization less effective. Conversely, larger datasets, like those with 500 variables, benefit significantly from our method, enabling the identification of nonlinear relationships and concept drift. This characteristic is somewhat counterintuitive compared to most existing time series models that often focus on single or low-dimensional (few variables) time series forecasting, such as sensor data streams. Despite this limitation, **we believe it underscores Drift2Matrix’s unique appeal. It addresses a gap in handling concept drift in time series with a large number of variables, offering excellent interpretability and reduced computational complexity.**
>
> **Regarding the length of co-evolving time series**, both our theoretical framework and experimental results demonstrate that the length of the series does not significantly affect the performance of our model. This is because Drift2Matrix is designed to adaptively learn domain-agnostic segmentation that identifies highly similar or repetitive concepts across different windows, regardless of the dataset’s length or the number of variables.
>
> We also appreciate your suggestion to include an ablation study on the number of co-evolving time series. While our current manuscript includes extensive ablation studies in Appendix H, this aspect was indeed omitted. In the revised version, we will introduce a new ablation study in Appendix H.8 (Ablation Study on the Number of Series), where we systematically reduce the number of co-evolving time series in a given dataset and analyze the corresponding changes in prediction performance ( a quick look at Fig. 16 in [Added Figures](https://anonymous.4open.science/api/repo/Drift2Matrix-main-86B7/file/ICLR_ADD_FIGURE.pdf?v=15a1f953)). This will provide additional insights into the model’s robustness when dealing with varying numbers of co-evolving series. We will also emphasize our theoretical and experimental analyses on the impact of the length of time series in the revised manuscript to further clarify this aspect.
>
> ---
> If you have any further suggestions or comments, we welcome them and will incorporate any necessary improvements to enhance the overall quality and reliability of the paper. Your feedback is valuable to us, and we are committed to producing a high-quality and impactful contribution.

---

> > ### Comment · Reviewer_oERz · 2024-11-27
> >
> > Thank you for your detailed response. It has addressed most of my concerns and questions. I have updated my score accordingly.

---

> > > ### Author Response · Authors · 2024-11-27
> > > **Response to Reviewer oERz**
> > >
> > > We are delighted that our responses have successfully addressed your concerns. We sincerely appreciate the time you took to review our clarifications and reevaluate your assessment. Your decision to increase the score is greatly appreciated.
> > >
> > > _If you have any further questions or suggestions, we would be glad to discuss them and provide additional clarifications to further elevate the quality and impact of this paper. Your insights have been invaluable in refining our work._

---

### Official Review · Reviewer_17sK · 2024-11-03

**Soundness:** 3
**Presentation:** 3
**Contribution:** 3
**Rating:** 8
**Confidence:** 4

**Summary:**

This paper focuses on the research problem of concept drift adaptation in evolving time series, and a novel framework called Drift2Matrix that leverages kernel-induced self-representation has been proposed. The whole paper is well organized with detailed formulation and theoretical analysis. Sufficient experiments have been given for model evaluation.

**Strengths:**

1. The topic of concept drift learning in co-evolving time series is interesting, and the proposed method based on kernel-induced self-representation shows its novelty and ideal performance.
2. The theoretical analysis of concept drift adaptation in this paper is well expressed and proofed.

**Weaknesses:**

1. The problem definition needs further strengthened, the scenario of concept drift in the co-evolving time series should be defined in detail.

**Questions:**

1. Is the learning mode in this paper a prequential test-then-train? If not, please given an explanation of the learning mode in this paper.
2. The experiment results of the proposed method seem not the best on some datasets, please give a detailed explanation.
3. Some benchmark methods used in this paper are proposed before 2020, I suggest continuing to add benchmark methods in the past three years.
4. Although the parameter setting has been given, the parameter sensitivity analysis is still required.

---

> ### Author Response · Authors · 2024-11-21
> **Response to Reviewer 17sK (1/2)**
>
> Thank you for your valuable feedback and positive comments. We appreciate your thorough evaluation on our paper. We are glad that you find our work on concept drift adaptation in co-evolving time series interesting. We have also observed that this adaptive method for discovering and visualizing concepts can be highly effective and improve interpretability in various fields, such as regime shifts in atmospheric and oceanic systems in environmental ecology, regime switches (e.g., bear and bull markets) in financial markets, and phase transitions in physics.
>
> ---
> > ### Concern 1: The problem definition and the learning mode (W1&Q1)
>
> Concept drift in co-evolving time series refers to dynamic changes in the underlying concepts governing the relationships between series over time. Unlike traditional time series models that often focus on stationary or independently evolving data, co-evolving time series involve complex, nonlinear interactions that may shift unexpectedly due to external or internal factors.
>
> In our paper, we define concept drift mathematically as changes in the latent concepts $\mathbf{C}_{c \in [1,k]} = \{\mathbf{C}_1, \cdots, \mathbf{C}_k\}$, which represent distinct interaction patterns among variables in the dataset $\mathbf{S} = \{S_1, S_2, \ldots, S_N\} \in \mathbb{R}^{T \times N}$. Specifically, concept drift occurs when the underlying relationships governing the system transition from one latent concept $\mathbf{C}_i$ to another $\mathbf{C}_j$ over time. This can be formally expressed as a change in the governing conditional distributions, $p(\mathbf{S} | \mathbf{C}_i)$ and $p(\mathbf{C}_i | t)$, across different time steps.
>
> ***To further clarify, we include these extensions in the revised version of the manuscript, alongside detailed descriptions in Sec. 3.***
>
> Besides, our learning mode is not strictly a prequential test-then-train approach. Instead, Drift2Matrix can be regarded as a twin-learning framework that learns domain-agnostic segmentation and concept drift. By leveraging kernel-induced representation, Drift2Matrix adaptively determines segmentation boundaries that align with the underlying dynamics of co-evolving time series, rather than relying on predefined segmentation or window sizes. Within each segment, Drift2Matrix iteratively detects and adapts to concept drift using its kernel-induced representation, ensuring both flexibility and robustness. We believe this approach aligns closely with the nature of co-evolving time series, where continuous adaptation to evolving concepts is critical. **Our learning mode can be summarized in two ways depending on the specific scenario:**
>
>  1. ***When applied to a new time series dataset***, Drift2Matrix first re-adaptively determines the most suitable segmentation size $\{W_1, \ldots, W_b\}$ and learns the concept profiles within each segment through kernel-induced representation. This ensures optimal alignment for concept identification.
>  2. ***When receiving new data points in an online learning setting*** (with a fixed segmentation size), Drift2Matrix quickly updates the kernel representation matrix for the new segment, enabling efficient adaptation without recalculating the segmentation size.
>
> **For both modes, the subsequent steps remain the same:** by counting the number of distinct profiles across all segments, we determine the number of distinct concepts present in the entire co-evolving time series dataset. Based on these discovered concepts, we can predict concept drift probabilities—both within a single series and through joint probabilities across the ecosystem.
>
> For clarity, we will revise Sec. 4 to explicitly describe this segmentation-based learning mode, its implementation in this study, and provide further details in the appendix.

---

> ### Author Response · Authors · 2024-11-21
> **Response to Reviewer 17sK (2/2)**
>
> ---
> > ### Concern 2: The experiment results on some datasets (Q2)
>
> We acknowledge that Drift2Matrix does not achieve the best performance on some datasets, as its primary focus is on adaptive concept identification and tracking rather than optimizing forecasting accuracy. This design prioritizes understanding and adapting to dynamic concept drift in co-evolving time series, setting Drift2Matrix apart from most recent deep learning methods like OneNet and FSNet, which aim to mitigate concept drift’s impact on forecasting by using ensembling or parameter tuning. In contrast, Drift2Matrix dynamically identifies and adapts to new concepts, delving deeper into the intrinsic structure of time series data.
>
> To demonstrate this, our experiments evaluate Drift2Matrix’s ability to identify key concepts (Sec 6.2 Q1: Effectiveness & Appendix H.2). Using synthetic and Stock datasets, we validate the identified concepts and their transitions, providing semantic explanations linked to financial market states. ***For other real-world datasets, where ground truth for concepts is unavailable, we thus validate the value and gain of the discovered concepts for time series forecasting.***
>
> As noted in Table 1, ***“While forecasting series task is not our main focus, we provide a comparison of Drift2Matrix with other models.”*** Drift2Matrix consistently demonstrates robust performance across diverse datasets, especially in scenarios with significant concept drift, showcasing its strengths in addressing dynamic interactions over purely static forecasting. However, on datasets where variables/channels are relatively independent and concept drift follows predictable patterns, our results are not as competitive as **OneNet — an ensemble forecasting model** (Note that ensemble learning models generally outperform single models). Nevertheless, our **single-model Drift2Matrix** achieves comparable outcomes. Furthermore, our extended deep learning framework, Auto-D2M, achieves the best results on nearly all datasets (Table 1 and Appendix H.3 Table 7 ).
>
> ***Remark: To address the limitations in current datasets, particularly those without concept ground-truth, we are developing a new dataset benchmark based on kernel-induced representation. This benchmark, designed to be released to the community, incorporates a novel concept validity metric and expert domain knowledge. By filling the gap in most time series datasets, which often lack concept labels, this new benchmark will enable more rigorous evaluation and further advance research in concept drift adaptation and detection.***
>
> ---
> > ### Concern 3:  Benchmark methods (Q3)
>
> We appreciate your suggestion to include more recent benchmark methods. Since the initial manuscript submission (evaluations initially included 13 benchmark methods, 6 of which were published after 2021 and 11 after 2019), we have continued testing the latest models to enhance the evaluation. We agree that integrating benchmarks from the past three years will further strengthen the comparisons. In our revised manuscript, we will include evaluations with newer models such as [SparseTSF[ICML2024], FITS[ICLR2024], FEDformer[ICML2022], DLinear[AAAI2023]], and update the results in Sec. 6 and Appendix H.3. In total, our models (Drift2Matrix and Auto-D2M) are compared with 17 benchmark methods (10 of which were published after 2021 and 15 after 2019) across 15 datasets from diverse domains (Complete experimental results in Appendix H.3, Table 7 or a quick look at Table 7 in [Added Figures](https://anonymous.4open.science/api/repo/Drift2Matrix-main-86B7/file/ICLR_ADD_FIGURE.pdf?v=15a1f953)), demonstrating the robustness and adaptability of our approach.
>
>
>
> ---
> > ### Concern 4:  Parameter sensitivity analysis (Q4)
>
> Thank you for your suggestion regarding the **parameter sensitivity analysis**. In our revision, we will include a comprehensive parameter sensitivity analysis, covering parameters $\alpha, \gamma$ (as shown in Eq. (3)) and $\beta$ (as shown in Eq. (15)), and provide insights into their impacts on model performance (integrated into Appendix F - Experimental Setup, or a quick look at Fig. 6, Fig. 17 in [Added Figures](https://anonymous.4open.science/api/repo/Drift2Matrix-main-86B7/file/ICLR_ADD_FIGURE.pdf?v=15a1f953)). Additionally, we will supplement the manuscript with ablation studies on different kernel functions, various regularization terms, multi-kernel learning tests, and a sensitivity analysis of adaptive window size learning, with detailed explanations provided in the corresponding sections of the appendix.
>
> ---
> If you have any further suggestions or comments, we welcome them and will incorporate any necessary improvements to enhance the overall quality and reliability of the paper. Your feedback is valuable to us, and we are committed to producing a high-quality and impactful contribution.

---

> > ### Comment · Reviewer_17sK · 2024-11-25
> >
> > I have read the response and revised paper carefully, the author has addressed the issues I concerned, I have no further suggestion, and I will keep the current score.

---

> > > ### Author Response · Authors · 2024-11-25
> > > **Response to Reviewer 17sK**
> > >
> > > We are pleased that our responses have addressed your concerns. Once again, we appreciate your positive evaluation!

---

### Official Review · Reviewer_kD8a · 2024-11-03

**Soundness:** 2
**Presentation:** 3
**Contribution:** 2
**Rating:** 3
**Confidence:** 5

**Summary:**

This paper focuses on the prediction of co-evolving time series under the occurrence of concept drift. To achieve accurate forecasting results, this paper proposes a kernel-based learning mechanism to learn representations that can capture the concepts among the co-evolving time series and thus can make accurate predictions by the learned representations.

**Strengths:**

This paper reviews the concept drift issue under multivariate time series forecasting and proposes a new method using kernel-based representations to identify and predict potential concepts, aka, patterns and make time series forecasting results that consider the adaptation of concept drift.

The idea has been clearly presented, with reasonable motivation. A comprehensive comparison has been conducted to show its effectiveness in forecasting over a variety of time series forecasting tasks.

**Weaknesses:**

The presented method is less novel considering [r1] has been the first study that address concept drift in time series forecasting.

The solution is less creative, which weakens its contribution to both multi-variant time series forecasting and concept drift adaptation domain. Clustering the data into separated concepts for concept drift adaptation is less creative. For example, the techniques used in r1, can also be considered separating the patterns by an ensembling mechanism. For me, it is less motivated and less clear here why kernel-based representation is a more advanced tool rather than ensembling. If that's the way to make a technical contribution, any new techniques which has the functionality of clustering can take a role.

There are concerns about some technical details which will be specified in the question section.

[r1] Q Wen, W Chen, L Sun, Z Zhang, L Wang, R Jin, T Tan. OneNet: Enhancing time series forecasting models under concept drift by online ensembling. NeurIPS 2024.

**Questions:**

1. From a high level understanding, this paper claims that existing methods consider "most multivariate models to define the concept as a collective behaviour of streaming data, falling short in their ability to capture the dynamics of individual series and their interactions". However, OneNet has considered the multivariate data through cross-time and cross-variable branches. Therefore, I didn't see a significant gap here.

2. Based on 1, the research question proposed here "Can we identify underlying concepts from co-evolving time series and leverage their nonlinear relationships to predict concepts that have not appeared in a single series?" is less convincing. Based on the experimental analysis, the aim of proposing Drift2Matrix is to increase the forecasting accuracy. However, why identifying concepts can be a more advanced way to achieve that aim than existing studies where these concepts have been considered in their model in a different way.

3. In addition, it is less rigorous to "predict concepts that have not appeared in a single series". Here, concepts are assumed to be distinct concepts, which means they are not overlapped with each other. According to the definition of concept drift, p(t) \neq p(t-1), the future concepts should not be predictable if concepts are distinct because Event of  p(t) \neq p(t-1) and the Event of p(t+1) \neq p(t) will be independent with each other. I think here the authors have mixed up temporal dependency among x(t) with dependency among concepts. That's why I don't think the THEORETICAL ANALYSIS section is the "genuine" theorem support for the proposed method.

4. Based on all the above, the experimental result looks fine but it does not exactly support the methodology from my understanding.

---

> ### Author Response · Authors · 2024-11-21
> **Response to Reviewer kD8a (1/3)**
>
> Thank you for your valuable feedback and comments on our paper. We appreciate the positive feedback on the clearly presented motivation and the comprehensive experimental evaluation. In the following reply to your concerns, we will first clarify the novelty and contributions of Drift2Matrix, particularly its use of kernel-induced self-representation to model time series as an interconnected ecosystem. This enables the capture of complex nonlinear interactions critical for adapting to concept drift in co-evolving series. We will then address your comments regarding the model’s ability to “predict concepts that have not appeared in a single series.”  We hope our responses convincingly address your concerns.
>
> ---
> > ### Concern 1: Kernel-induced representation for concept drift adaptation in co-evolving time series (Q1&Weakness)
>
> Current concept-aware models primarily aim to mitigate the impact of concept drift on forecasting. In contrast, Drift2Matrix focuses on the challenges of adaptive concept identification and tracking dynamic concept drift in co-evolving time series. In the follows, we discuss the key distinctions and contributions of Drift2Matrix as compared to existing concept-aware forecasting models from different perspectives:
>
>  **1. Focus of Research**: While observing the rapid developments in deep learning for time series forecasting, it is important to highlight that current advancements predominantly focus on improving time series forecasting accuracy. Notably, most recent deep learning methods addressing concept drift, such as ***OneNet and FSNet, primarily aim to mitigate the impact of concept drift on forecasting***. These methods achieve their goals by incorporating ensembles of models with diverse data biases or by refining network parameters to enhance adaptability. For instance, OneNet reduces the impact of concept drift by leveraging channel independence and improves forecasting accuracy through channel dependencies, with a focus on online time series forecasting by introducing an **ensemble of models** that account for different data biases.
>
> In contrast, Drift2Matrix focuses on the challenges of adaptive concept identification and dynamic concept drift in co-evolving time series. By modeling the ecosystem of co-evolving time series, Drift2Matrix captures explicitly nonlinear relationships of concepts across different series. It tracks the evolution of concepts and predicts concepts that have not appeared in a single series (_see our response to Concern 2_). **Therefore, the primary objective of Drift2Matrix is to identify and track concepts within time series data, offering a fundamentally different approach to interpreting concept drift**. Drift2Matrix delves deep into the intrinsic structure of time series data, enabling a nuanced understanding of concept drift by dynamically identifying and adapting to new concepts as they emerge.
>
> To demonstrate Drift2Matrix’s capability, we evaluate its ability to identify key concepts and transitions (Sec 6.2 Q1: Effectiveness & Appendix H.2) using synthetic and Stock datasets, providing semantic explanations linked to shifts in financial market states. **Given that other real-world datasets lack ground truth, we validate the value of the discovered concepts through their impact on time series forecasting.** As noted in the text beneath Table 1, ***While forecasting series task is not our main focus, we provide a comparison of Drift2Matrix with other models.*** Table 1 and Appendix H.3 (Table 7) compare Drift2Matrix with 17 methods, encompassing both forecasting and concept-aware models, across diverse datasets. Notably, **OneNet, an ensemble learning model**, demonstrates strong performance, achieving the good results on some datasets. However, our **single-model Drift2Matrix** achieves comparable outcomes. Furthermore, our deep learning extension, Auto-D2M, consistently achieves the best results across nearly all datasets. Additionally, Appendix H.2 presents further evaluations showcasing Drift2Matrix’s concept identification and drift-tracking abilities (Fig. 8-9), while Appendix H.5 validates its performance on the Motion Segmentation sequences dataset, emphasizing its robustness in diverse real-world scenarios.

---

> ### Author Response · Authors · 2024-11-21
> **Response to Reviewer kD8a (2/3)**
>
> **2. Interpretability**: Drift2Matrix introduces kernel-induced representation to reveal nonlinear relationships in time series, substantially boosting both adaptability and interpretability of the model. In particular, Drift2Matrix treats co-evolving time series as an ecosystem and represents them in a matrix format to capture and interpret the nonlinear relationships among series within the ecosystem. **By observing changes in the ecosystem, we track the trajectories of each concept as they evolve over time**.
>
> Kernel-induced representation learning allows time series to be expressed in a nonlinear space (Hilbert space) using the self-representation property of the data. By incorporating block-diagonal regularization, **we enforce k-connectivity in the nonlinear space during the ecosystem’s evolution and transitions (Sec 5.1)**. This k-connectivity corresponds to k distinct concepts, where the number k is adaptively determined by calculating the eigenvalues of the Laplacian matrix derived from the kernel-induced representation. Moreover, kernel-induced representation learning **preserves the local prevalent structures of the time series in the high-dimensional nonlinear space (Sec 5.2)**. This ensures that even after mapping into a new feature space, the shape characteristics of the time series remain intact—an essential property for uncovering the nature of concepts. **This capability is often overlooked in most existing  methods, making Drift2Matrix particularly well-suited for concept discovery.**
>
> Furthermore, the kernel-induced representation enables a **twin-learning framework** to achieve **domain-agnostic segmentation** and **concept drift detection**. Notably, most existing models, including **OneNet**, rely on pre-specified segmentation lengths or window sizes. Drift2Matrix employs the segment-score (derived from our kernel-induced representation, Appendix D Eq. 24) to determine an optimal window size that identifies highly similar or repetitive concepts across different windows. Fig. 4 illustrates the adaptive determination of optimal window boundaries, and Fig. 5 shows the best window size for each dataset. Additionally, our kernel-induced representation **adaptively determines the number of concepts**, which is crucial for domain-agnostic time series analysis. This ensures flexibility and robustness across diverse datasets.
>
>
> **3. Example of Financial Market**: Financial time series analysis transcends mere forecasting; it demands interpretability that builds trust and supports applications like portfolio management. Drift2Matrix excels in this regard, offering clear insights into market dynamics beyond the conventional categories of bull, bear, or sideways markets. It adeptly captures a wide array of market scenarios, identifying distinct concepts driven by various factors—whether it’s value versus growth or the interplay between small and large caps. For in-depth case studies, including analyses of the Stock datasets, please refer to Sec 6.2 & 6.4.
>
> The table below compares Drift2Matrix with other approaches across various capabilities relevant to time series analysis, including segmentation, concept identification, and forecasting.
>
> >
> > **Table: Capabilities of approaches**
> >
> > |                       | HMM/++ | ARIMA/++ | WCPD-RS | ORBITMAP | LSTM/INFORMER | COGRA | ONENET | FSNET | Drift2Matrix |
> > |-----------------------|--------|----------|---------|----------|---------------|-------|--------|-------|--------------|
> > | Multiple time series  | -      | -        | -       | -        | ✓             | -     | ✓      | -     | ✓            |
> > | Compression           | ✓      | -        | ✓       | ✓        | -             | ✓     | -      | -     | ✓            |
> > | Domain-agnostic seg.  | -      | -        | -       | ✓        | -             | -     | -      | -     | ✓            |
> > | Concept Identification (nonlinear)| -      | -        | -       | -        | -             | -     | -      | -     | ✓            |
> > | Trajectory tracking   | -      | -        | ✓       | ✓        | -             | -     | -      | -     | ✓            |
> > | Mitigating drift impact     | -      | -        | ✓       | ✓        | -             | ✓     | ✓      | ✓     | ✓            |
> > | Forecasting           | -      | ✓        | -       | ✓        | ✓             | ✓     | ✓      | ✓     | ✓            |

---

> ### Author Response · Authors · 2024-11-21
> **Response to Reviewer kD8a (3/3)**
>
> ---
> > ### Concern 2:  predict concepts that have not appeared in a single series (Q2&Q3)
>
> We are grateful for your recognition of our work and understand your concerns regarding the concept prediction problem. Predicting concepts that have not appeared in a single series is **one of the key contributions of Drift2Matrix**, achieved through our unique ecosystem modeling of co-evolving time series. This ecosystem modeling leverages the kernel-induced representation property, which represents each series based on its nonlinear relationships with others in the ecosystem. In essence, this is a form of representation learning in a nonlinear space, where each series is expressed in terms of its interactions with others.
>
> This representation enables Drift2Matrix to predict concepts even if they have not yet appeared in an individual series but have occurred elsewhere in the ecosystem. From another perspective, Drift2Matrix effectively implements **zero-shot learning** within the ecosystem, enabling predictions of unseen concepts in single series.
>
> In summary, Drift2Matrix leverages both temporal and cross-series dependencies to identify concepts and predict those that have not been observed in a single series (rather than focusing on dependencies among concepts). While temporal dependencies contribute to tracking the evolution of concepts within a series, cross-series dependencies play a stronger role in identifying and predicting concepts by capturing the interactions across the ecosystem. This approach ensures a comprehensive understanding of dynamic concept drift in co-evolving time series.
>
> Let us illustrate this with two examples:
>
> **(1) Recommender Systems**: In the context of a recommendation system, Drift2Matrix can identify concepts such as preferences or purchasing habits by modeling the social relationships between users. For example, based on the purchasing behavior of a user’s peers (their ecosystem), Drift2Matrix can recommend products that the user has never bought before but are likely relevant to their preferences.
>
> **(2) Financial market**: Consider a financial ecosystem where each series represents the price movements of different but related stocks. Suppose $S\_i$ has not yet experienced a market shock leading to a sudden price drop (a “concept” in our context). However, several related series  $S\_j$ ,  $S\_k$ , etc., within the ecosystem have already experienced this type of shock. Drift2Matrix models  $S\_i$ not in isolation but through its nonlinear relationships with other series using kernel-induced self-representation. By learning from the concepts and interactions within these series, Drift2Matrix can predict the likelihood of a similar market shock for  $S\_i$ , even though it has not directly encountered such a concept before.
>
> The mathematical form for predicting concepts is as follows - for windows $W\_p$ and $W\_{p+1}$, the effective probability of a transition from concept $\mathbf{C}\_r$ to $\mathbf{C}\_m$ in series $S\_i$ can be calculated as follows:
>
> $$P(\mathbf{C}\_r \rightarrow \mathbf{C}\_m|W\_{p}\rightarrow W\_{p+1}, S\_i) = \frac{\sum\_\zeta \textcolor{red}{\Psi\_{p,p+1}^{r,\zeta}}\textcolor{blue}{\Lambda\_{p,p+1}^{\zeta, m}}}{\sum\_{\zeta_1}\sum\_{\zeta\_2}\textcolor{red}{\Psi\_{p,p+1}^{\zeta\_1,\zeta\_2}}\textcolor{blue}{ \Lambda\_{p,p+1}^{\zeta\_1,\zeta\_2}}}$$
>
> where $\zeta\_1, \zeta\_2 \in \{1,\cdots, k\}$ and
>
> $$ \textcolor{red}{\underbrace{\Psi\_{p,p+1}^{r,m}= \frac{\eta \left(\mathbf{C}\_r \rightarrow \mathbf{C}\_m | \mathcal{T}r\left(S\_i|W\_{p}\right)\right)}{|\mathcal{T}r(S_i|W_{p})|}}\_{\mathbf{Transition \ probability \ of \ \mathbf{C}\_r \ to \ \mathbf{C}\_m \ in}\ S_i}} , \quad   \textcolor{blue}{\underbrace{\Lambda\_{p,p+1}^{r,m}=\sum\_{l=1}^{p-1} \frac{\min\{\eta(\mathbf{C}\_r,W\_l), \eta(\mathbf{C}\_m,W\_{l+1})\}}{\max\{\eta(\mathbf{C}\_r,W\_l), \eta(\mathbf{C}\_m,W\_{l+1})\}}}\_{\mathbf{Transition \ probability \ of \ \mathbf{C}\_r \ to \ \mathbf{C}\_m \ in\ Ecosystem}}}
> $$
>
> Here, $\mathcal{T}r\left(S\_i|W\_{p}\right)$  represents the concept trajectory of $S\_i$ over time, while $\eta \left(\mathbf{C}\_r \rightarrow \mathbf{C}\_m | \mathcal{T}r\left(S\_i|W_{p}\right)\right)$ counts occurrences of the concept pair ($\mathbf{C}\_r, \mathbf{C}\_m$) in this trajectory. $\textcolor{red}{\Psi\_{p,p+1}\^{r,m}}$ thus assesses the **immediate risk** of transitioning to $\mathbf{C}\_m$ after $\mathbf{C}\_r$, while $\textcolor{blue}{\Lambda\_{p,p+1}^{r,m}}$ captures the likelihood of such transitions **across the ecosystem**. **Consequently, these terms enable Drift2Matrix to integrate both individual series behavior and the collective concept dynamics across $\mathbf{S}$.**
>
> ---
> If you have any further suggestions or comments, we welcome them and will incorporate any necessary improvements to enhance the overall quality and reliability of the paper. Your feedback is valuable to us, and we are committed to producing a high-quality and impactful contribution.

---

> > ### Comment · Reviewer_kD8a · 2024-11-25
> >
> > I appreciated your answers. However, I am not using how you predict.
> >
> > Instead, I am asking how you define the predictable concept because from my understanding all the theorems derive from the definition. As you assume it is predictable, meaning C(t+k) can be inferred from its previous C(t). Here let's assume C(t+k) = G(t,C,k). You separate it to distinct Cs by sliding windows. Honestly, I didn't find where you have defined this distinct concept, but it just appears in your theorem, which is very confusing for me to under your methodology at the theoretical level. If I understand concepts as representatives, they are not born to be distinct from each other.

---

> ### Author Response · Authors · 2024-11-25
> **Response to Reviewer kD8a**
>
> Thanks for your feedback! We appreciate the opportunity to clarify our methodology further and build upon your example, “C(t+k) can be inferred from its previous C(t),” where C(t+k) = G(t, C, k). In this case, determining C—the distinct concepts—is essential. Our model achieves this by adaptively learning the distinct concepts within each sliding window through kernel-induced representation learning. Specifically:
>
>  **1. Learning Distinct Concepts Within a Sliding Window**:
>
>
> In each sliding window, Drift2Matrix produces a representation matrix $\mathbf{Z}$, which exhibits a block-diagonal structure where each block corresponds to a unique concept (e.g., C1–C5, as shown in Fig. 1(b)). As described in Sec. 4, we aim for $\mathbf{Z}$ to exhibit $k$ block diagonals if the time series $\mathbf{S}$ contains $k$ concepts, ensuring that distinct concepts emerge as mathematically coherent matrix subspaces. To achieve this, we incorporate a block diagonal regularization term to enforce this structure - $||\mathbf{Z}||\_{\boxed{\scriptstyle k}} = \sum\_{i=N-k+1}^N \lambda\_i(\mathbf{L\_{\mathbf{Z}}})$. This regularization directly links matrix connectivity (i.e., the number of block diagonals) to $k$. For instance, if $k=5$, the matrix’s diagonal contains five distinct blocks, representing five distinct concepts. This ensures no overlap of similar concepts within the same sliding window due to the matrix’s block-diagonal property. Furthermore, our kernel-induced representation adaptively determines the value of $k$ (the number of concepts), as detailed in Appendix B—_Estimating the Number of Concepts_.
>
> **2. Differentiating Similar Concepts Across Sliding Windows**:
>
> As described in 1, each sliding window produces a representation matrix where each block represents a concept. We define a concept as the profile pattern or prototype observed within a specific segment or window, corresponding to similar subseries (subseries within the same block). Here, the term “profile pattern/prototype” refers to a subseries whose vector representation aligns with the centroid of similar subseries.
>
> Across different sliding windows, the resulting representation matrices may identify similar blocks/concepts (see Fig. 1(b)). To manage this, we introduce a tunable hyperparameter $\rho$, which regulates concept profiles or prototypes. This parameter controls whether concept drift is gradual (smaller $\rho$, leading to more concepts) or abrupt (larger $\rho$, resulting in fewer concepts). This mechanism is discussed in detail in Sec. 3—_Preliminaries_.
>
> **3. Theoretical Support**:
>
> We provide theoretical backing for our methodology:
>
> (1) **In Theorem 4.1**, we prove that minimizing the block diagonal regularization term $||\mathbf{Z}||\_{\boxed{\scriptstyle k}} = \sum\_{i=N-k+1}^N \lambda\_i(\mathbf{L\_{\mathbf{Z}}})$ is equivalent to ensuring that the representation matrix $\mathbf{Z}$ exhibits $k$ block diagonals. This directly corresponds to the presence of $k$ distinct concepts.
>
> (2) **Theorem 5.1** proves that the block diagonal structure of the representation matrix—and hence the identified concepts—remains invariant to the ordering of the input time series.
>
> (3) **Theorem 5.2** establishes that the representation matrix learned in the nonlinear space retains the local prevalent structures of the original space. This property is critical for concept identification, ensuring that concepts recognized in the transformed space remain interpretable and mappable to the original time series.
>
> **4. Planned Enhancements**:
>
> To address your concern about explicitly defining distinct concepts, we will revise the manuscript to strengthen Sec. 4 by providing a formal definition directly tied to the kernel representation framework. Additionally, in the introduction section (Preview of Our Results and Fig. 1), we will highlight and emphasize key aspects of concept identification by bolding critical parts.
>
> Furthermore, we will include a detailed example in the appendix, demonstrating how concepts are derived as distinct blocks or subspaces, how different concepts are clearly defined, and how their transitions are inferred and predicted.

---

> > ### Comment · Reviewer_kD8a · 2024-11-26
> >
> > Hi, as in my previous review, I am **NOT** asking how you predict. I am asking how you **define distinct concept**.
> >
> > Rather than the above irrelevant replies, could you provide a definition of distinct concept in math, so that I can understand that in your theorem?

---

> ### Author Response · Authors · 2024-11-26
>
> Hi,
>
> > For window $W_p$, the $r$-th concept is defined as the vector representation of subseries corresponding to the $r$-th block, $\mathbf{Z}\_p^{(r)}$, in the representation matrix $\mathbf{Z}\_p$. Specifically, it aligns with the centroid of similar subseries, represented as $C\_{r,p} = \text{Centroid} \Big( \{\mathbf{S}\_i \ | \ \mathbf{S}\_i \in \mathbf{Z}\_p \^{(r)} \}\Big)$.To differentiate and refine similar or repeated concepts across different windows, two concepts $C_{r,p}$ and $C_{s,p+1}$ are considered distinct if $||C_{r,p}- C_{s,p+1}||\_F^2 > \rho$, where $\rho$ is a tunable hyperparameter that regulates the granularity of concept.
>
> This process and definition, detailed in Sec 3 and 4, ensures that concepts are uniquely identified both within and across sliding windows. Our previous response **focused on providing an in-depth explanation of this process**, divided into two parts: (1) identifying and defining concepts within a window and (2) distinguishing concepts across different windows, with the goal of addressing your concern. Additionally, our theorems demonstrate that the properties of concept remain consistent across both the original and nonlinear spaces.
>
> ---
> If you have any further suggestions or comments, we welcome them and will incorporate any necessary improvements to enhance the overall quality and reliability of the paper.

---

> > ### Comment · Reviewer_kD8a · 2024-11-27
> >
> > Thanks.
> >
> > Where can I find this in the submission?
> >
> > How $\rho$ influence your Theorem 1? distinct are between 2 concepts, how this extends to the case when k>2?

---

> ### Author Response · Authors · 2024-11-27
>
> Hi,
>
> let's elaborate further:
> > We define _concepts_ in **Sec. 3 (Preliminaries)**, including a concept’s **profile pattern or prototype** is described as the vector representation aligning with the centroid of similar subseries. We intentionally kept the description in Preliminaries concise to provide readers with an intuitive understanding of concepts, without introducing potentially confusing parameters like _blocks_ or _windows_ at this early stage. \
> Instead, we adopted a **gradual introduction** - **Sec. 4.1:** We begin with a simple case, treating the entire series as a single window, focusing on the _kernel-induced representation matrix_. **Sec. 4.2:** We extend this to multiple sliding windows, connecting concepts to distinct blocks in the representation matrix. \
> We have included a complete formal definition of concepts in Appendix A and referenced it in the Preliminaries to provide additional clarity for readers interested in a detailed explanation. Please see the Formal Mathematical Definition of Concepts part in our revised Appendix A.
>
> ---
>
> >**Regarding Theorem 1.** (stated as Theorem 4.1 in the manuscript) -- minimizing the regularization term $\min \sum_{i=N-k+1}^N \lambda_i(\mathbf{L_{\mathbf{Z}}})$ is equivalent to $\mathbf{Z}$ being k-block diagonal. \
> **1 - Explanations:** \
> **(1) Parameter** $\rho$ **is unrelated to Theorem 1**: Theorem 1 operates at the level of a single representation matrix $\mathbf{Z}$. It guarantees that minimizing the regularization term automatically separates k distinct concepts as k-blocks in $\mathbf{Z}$. Here, k is adaptively determined by our kernel-induced representation method (detailed in **Appendix B**). Therefore, distinct concepts within a window are directly derived from the block diagonal structure of $\mathbf{Z}$, independent of $\rho$. \
> **(2) Similarly, Theorem 2 (stated as Theorem 5.1)** also focuses on the behavior of a single matrix $\mathbf{Z}$. Equation (7) further shows that the block diagonal structure of $\mathbf{Z}$ is invariant to the reordering of the data, ensuring robustness in concept identification within a single window. \
> **2 - Role of $\rho$:** \
> The parameter $\rho$ comes into play when integrating concepts across **multiple windows**. While a single representation matrix $\mathbf{Z}_i$ defines distinct concepts through its block diagonal structure, different windows ($\mathbf{Z}_1, \ldots, \mathbf{Z}_b$) may identify similar or overlapping concepts, as shown in **Fig. 1(b)**. \
> For instance, the number of concepts detected per window in **Fig. 1(b)** are 5, 4, 3, 5, and 3. The total number of distinct concepts across all windows, however, is refined to 5 (as shown in **Fig. 1(c)**) using $\rho$. The tunable hyperparameter $\rho$ regulates the threshold for distinguishing profile patterns across windows. A **smaller** $\rho$ allows for finer differentiation (more concepts), while a **larger** $\rho$ results in coarser differentiation (fewer concepts), effectively modulating the granularity of concept drift detection. \
> **3 - Practical Considerations:** \
> In practice, we find that $\rho$ enables a multi-scale analysis, distinguishing higl-scale level vs. low-scale level concept drift. However, this aspect is beyond the primary scope of this paper.

---

> > ### Comment · Reviewer_kD8a · 2024-11-27
> >
> > Thanks for the explanation.
> >
> > when $\rho$ set to some value, no concepts are distinct, then your Z can be 0-block diagonal?
> >
> > your definition of distinct concepts is cross different windows, meaning $p$ to $p+t$, $t$ can be any positive or negative integer, but in the written math equation, it seems $t$ has to be 1?
> >
> > As for my question distinct are between 2 concepts, how this extends to the case when k>2?, I mean distinct are between rc=2 concepts, how this extends to the case when rc>2?
> >
> > If $dist(C_{r1,p}-C_{s1,p+1})>\rho$, $dist(C_{r1,p}-C_{s2,p+1})<\rho$ and $dist(C_{r2,p}-C_{s1,p+1})<\rho$, $dist(C_{r2,p}-C_{s2,p+1})>\rho$, how many distinct concepts here?

---

> > > ### Author Response · Authors · 2024-11-27
> > >
> > > Thank you for your follow-up questions. Allow me to address your concerns in detail:
> > >
> > > >**Regarding “when** $\rho$ **is set to some value, no concepts are distinct, then your** Z **can be 0-block diagonal?”** \
> > > **This situation does not occur in our method.** The kernel-induced representation prevents such a scenario because the regularization we impose on Z enforces a block diagonal structure. The value of $\rho$ only affects the differentiation of concepts across **different windows**. Since Z is the representation matrix generated independently within each window, the value of $\rho$ does not influence the block diagonal structure within Z, ensuring that Z always has blocks representing distinct concepts. Thus, we want to **reiterate that** the parameter $\rho$ comes into play **only** when integrating concepts across multiple windows, and therefore does not affect the number of blocks in the representation matrix Z under each window. \
> > > **In extreme cases where** $\rho$ **is set very large:** \
> > > **If $\rho$ is set to a sufficiently large value such that no distinct concepts are identified across windows, the window containing the largest number of blocks in its representation matrix Z will dominate.** For example, assume we have three windows, and the block counts for their representation matrices are 4, 3, and 2, respectively. If $\rho$ is set so large that all concepts across windows are considered identical, the total number of distinct concepts identified will be equal to the number of blocks in the window with the largest count — in this case, 4. Hence, our approach naturally handles scenarios where k > 2, which is true for all datasets in our experiments.
> > >
> > > >**Clarification on the mathematical definition and the role of t:** \
> > > The comparison between windows $p$ and $p+1$ in our definition was presented as an example to demonstrate how distinct concepts are compared across adjacent windows. The value of t (the offset between windows) can be arbitrary. The same methodology applies to comparing distinct concepts across any two windows p and p+t.
> > >
> > > >**Regarding the example:** \
> > > For the scenario you mentioned: \
> > > $dist(C_{r1,p} - C_{s1,p+1}) > \rho, \ dist(C_{r1,p} - C_{s2,p+1}) < \rho, dist(C_{r2,p} - C_{s1,p+1}) < \rho, \ dist(C_{r2,p} - C_{s2,p+1}) > \rho,$ \
> > > In our method and experiments, this results in two final concepts: r1 and s2 are grouped as one concept, and r2 and s1 as another. Our method performs pairwise comparisons across all concepts in different windows to determine their distinctiveness. Specifically, for a concept s1 in window $W_{p+1}$ to be considered distinct from all concepts in window $W_p$, its distance to every concept in $W_p$ must exceed $\rho$.

---

> ### Comment · Reviewer_kD8a · 2024-11-27
>
> Thanks for your prompt rely.
>
> I will need to check Z further. Line 220 --we can easily group the time series into k concepts using traditional spectral clustering technology Ng et al. (2001), is the k the same k in theorem 1, k distinct concepts, which means you use Ng et al. (2001) to get the distinct concepts but them are defined as dist(r1,s2)>p?
>
> For the other two questions, The value of t (the offset between windows) can be arbitrary. As we move forward to have more and more windows, the range of t increases?
>
> In our method and experiments, this results in two final concepts: r1 and s2 are grouped as one concept, and r2 and s1 as another. why?
> As both dist(r1,s2)>p, dist(r2,s1)<p, so they are grouped as one because they are distinct?
>
> Anway, given this is the case, r1 and s1 should not be in the group, r2 and s2 should not be in the group, right? This will be the case of rc=2.
>
> Let's consider rc grows, there might be a case that  r1, s2, v1 should be grouped as one concept while r1, s1, v1 should not in the group. Or, you only consider rc=2. Then my question is if this limits your theorem and method.

---

> ### Author Response · Authors · 2024-11-27
> **Response to Reviewer kD8a**
>
> Thank you for your attention to detail as we provide further clarification:
>
> >**On the Range of t:** \
> The range of  t is related to the number of windows, which itself is determined by our adaptive segmentation mechanism. This adaptability is another potential contribution of our kernel-induced representation. Unlike most existing methods that require predefining a fixed window size, **our framework introduces domain-agnostic segmentation**. Specifically, our framework framework employs a segment-score (derived from kernel-induced representation, **Appendix D, Eq. 24**) to determine an optimal window size. **Figure 4** illustrates how optimal window boundaries are adaptively determined. **Figure 5** presents the best window size determined for various datasets. Thus, once we determine the window size, the total number of windows can be derived, and the range of  t can be established accordingly.
>
>
> >**On the case (** r1, s2, v1 **):**\
> If  r1, s1, v1 are concepts identified within the same window (e.g.,  W_1 ), then they cannot belong to the same group. This is due to the block diagonal regularization, which enforces their distinctness by ensuring separable blocks in  Z1 .\
> **I understand your concern** -- if  s2 is identified in a different window (e.g.,  W_2 ), and its distances to  r1 and  v1 are both below $\rho$, then,  r1, s2, v1 should be grouped as one concept. Yet, in practice, we observe that such cases do not occur.
>
> >**Why This Does Not Happen in Practice:** \
> In our experiments, we observed that the block connectivity constraint of the representation matrix Z (i.e., the separation between blocks) ensures that cases such as r1, s2, v1 being grouped as one concept do not occur. This can be explained by a **theoretical lower bound** $\delta$ on the distance between different blocks within Z (e.g., dist(r1, v1) $\geq \delta$) , which is related to the Laplacian eigenvalue gap in our Appendix B. Specifically: when the threshold $\rho < \delta/2$, at least one of the distances dist(r1, s2) or dist(v1, s2)) will always exceed $\rho$. This ensures that s2 can be grouped with only one of r1 or v1, but not both.
> >>**Proof** \
> By the triangle inequality, we know: dist(r1, s2) + dist(v1, s2) $\geq$ dist(r1, v1).
> Since dist(r1, v1) $\geq \delta$, it follows that:
> dist(r1, s2) + dist(v1, s2) $\geq \delta$.
> Thus, at least one of the terms, dist(r1, s2) or dist(v1, s2), must satisfy: dist(r1, s2) $\geq \delta/2$ or  dist(v1, s2) $\geq \delta/2$.\
> Since $\rho < \delta/2$, at least one of the distances exceeds $\rho$, ensuring that s2 cannot be grouped with both r1 and v1 simultaneously.
>
> ***While we have conducted additional theoretical analysis on this behavior, we do not to include it in Appendix B as it extends beyond the scope of this paper.***
>
> >Also, as noted in Sec.7 Conclusion, Drift2Matrix has limitations when applied to time series with few variables. For example, converting a dataset with five variables into a $5 \times 5$ matrix reduces the effectiveness of block diagonal regularization. In contrast, larger datasets with hundreds of variables significantly benefit from our method, which excels at identifying nonlinear relationships and concept drift. This is somewhat counterintuitive compared to most time series models, which typically focus on low-dimensional forecasting tasks, such as single-sensor data. Despite this limitation, we believe it highlights Drift2Matrix’s unique strengths - by addressing a gap in handling concept drift in a large number of variables time series and offers good interpretability.

---

> > ### Author Response · Authors · 2024-11-27
> > **Response to your update Line 220**
> >
> > We apologize that we didn't see your revised update until after we submitted our response.
> >
> > Regarding Line 220, in the context of a single window—i.e., a single representation matrix Z—we use spectral clustering (as per Ng et al. 2001) to identify the k concepts within that window. Similarly, Theorem 1 in Sec. 4.1 also pertains to a single window scenario (“We begin with a simple case, where we treat the entire series as a single window.”). Therefore, k in Line 220 is indeed the same k as in Theorem 1.
> >
> > Additionally, concerning dist(r1, s2) > $\rho$, we want to reiterate that **$\rho$ plays a role only when differentiating concepts across different windows**, **not within a single window** (**as we have emphasized in each of our previous responses.**). Within a single window, the k distinct concepts are directly derived from the block diagonal structure of Z, independent of $\rho$.

---

> > > ### Comment · Reviewer_kD8a · 2024-11-28
> > >
> > > Hi thanks.
> > >
> > > No worries about the submission version.
> > >
> > > ***On the Range of t: range of t is adaptive with regards to different dataset, rather than temporally adaptive?
> > >
> > > ***On the case ( r1, s2, v1 ): sorry about the notation, it inherits from the previous question, the full version will be
> > > dist(r1, s2) is short for  $dist(C_{r1,p} - C_{s1,p+1}) $.
> > >
> > > Namely, r repesents any r at p+t_r, and similar to s and v, as you consider cross windows more than 2 as you have replied. In my understanding, r,s,v are not the same variable as at each window p, the clustering re-runs. Thus, (r1, s2, v1) means one example of (r,s,v). Give this definition, could you explain ---Let's consider rc grows, there might be a case that r1, s2, v1 should be grouped as one concept while r1, s1, v1 should not in the group. Or, you only consider rc=2. Then my question is if this limits your theorem and method.
> > >
> > > The replied on Line 220 : where we treat the entire series as a single window. if this is a single window, why need to consider distinct concepts, as they are defined on cross windows?

---

> ### Author Response · Authors · 2024-11-28
>
> Hi,
>
> > **On the Range of  t:** \
> We are confused by your concern regarding the range of t. In our method, the number of windows, which is determined by the identified optimal window size for a given dataset. **For instance,** ***in a time series of length 1000, if the identified window size is 100, then there will be 10 non-overlapping windows ( $W_1$ to $W\_{10}$ ).*** As such, the maximum offset t cannot exceed the total number of windows (e.g., $t \leq 9$ in this example). We hope this clarifies your concern.
>
> >**On Case  (r1, s2, v1) :** \
> In our previous response, **we already detailed why cases such as “ r1, s2, v1  should be grouped as one concept” do not occur in our framework.** Let us provide clarification and proof: \
> **Why This Does Not Happen in Practice:** \
> In our experiments, we observed that the block connectivity constraint of the representation matrix Z (i.e., the separation between blocks) ensures that cases such as r1, s2, v1 being grouped as one concept do not occur. This can be explained by a **theoretical lower bound** $\delta$ on the distance between different blocks within Z (e.g., dist(r1, v1) $\geq \delta$) , which is related to the Laplacian eigenvalue gap in our Appendix B. Specifically: when the threshold $\rho < \delta/2$, at least one of the distances dist(r1, s2) or dist(v1, s2)) will always exceed $\rho$. This ensures that s2 can be grouped with only one of r1 or v1, but not both.
> >>**Proof** \
> By the triangle inequality, we know: dist(r1, s2) + dist(v1, s2) $\geq$ dist(r1, v1).
> Since dist(r1, v1) $\geq \delta$, it follows that:
> dist(r1, s2) + dist(v1, s2) $\geq \delta$.
> Thus, at least one of the terms, dist(r1, s2) or dist(v1, s2), must satisfy: dist(r1, s2) $\geq \delta/2$ or  dist(v1, s2) $\geq \delta/2$.\
> Since $\rho < \delta/2$, at least one of the distances exceeds $\rho$, ensuring that s2 cannot be grouped with both r1 and v1 simultaneously.
>
> ***While we have conducted additional theoretical analysis on this behavior, we do not to include it in Appendix B as it extends beyond the scope of this paper.***
>
> >**On Distinct Concepts in a Single Window:**\
> **1. Distinct Concepts in a Single Window:** **As emphasized in our previous responses**, even within a single window, the representation matrix  Z identifies distinct concepts (e.g., r1, s1, v1), as defined by its block diagonal structure. Each block corresponds to a distinct concept.\
> **2. gradual introduction:** **As emphasized in our previous responses**, we introduce our framework in an incremental manner: **Sec. 4.1:** We begin with a simple case, treating the entire series as a single window, focusing on the kernel-induced representation matrix and the identification of distinct concepts within a single window; **Sec. 4.2: We extend this to multiple sliding windows, introducing mechanisms for distinguishing and merging concepts across different windows.**

---

> > ### Comment · Reviewer_kD8a · 2024-11-28
> >
> > to make sure we are on the same page of adaptation, my question is
> >
> > if an identified optimal window size can change when we are receiving more data. For example, the most recent window P and its previous can be used to predict the next window. As time goes, we can then get the ground truth of S in window P+1. With this new window, we can conduct the same process to update the determined kernel, but here you choose not to, which is
> >  not temporally adaptive?

---

> ### Author Response · Authors · 2024-11-28
>
> Hi,
>
> >In our method, given a time series dataset, Drift2Matrix adaptively determines the optimal window size and performs predictions for the subsequent windows. \
> However, **when new data points are received, such as in a real-time or online learning setting**, the window size remains fixed after the initial determination. Drift2Matrix efficiently updates the kernel representation matrix for the new segment, allowing for fast adaptation **without recalculating the segmentation size**. This design ensures efficiency and avoids the computational cost of repeatedly optimizing the segmentation size.\
> We have elaborated on these modes in the **learning mode of Appendix F**. For your convenience, **here is the relevant excerpt**:
> >>Our learning mode of kernel-induced representation can be summarized in two ways depending on the specific scenario:\
> **1.When applied to a new time series dataset**, Drift2Matrix first re-adaptively determines the most suitable segmentation size and learns the concept profiles within each segment through kernel-induced representation. \
> **2. When receiving new data points in an online learning setting (with a fixed segmentation size)**, Drift2Matrix quickly updates the kernel representation matrix for the new segment, enabling efficient adaptation without recalculating the segmentation size.
>
> Best,

---

> > ### Comment · Reviewer_kD8a · 2024-11-30
> >
> > Thanks.
> >
> > Given the window size optimised and fixed while the kernel representation updates, let's then review how k is optimised.
> >
> > 1. k Distinct Concepts in a Single Window: use Ng et al. (2001)
> > 2. k Distinct Concepts cross Window: use |Crp-Cs,p+t|>rho
> >
> > Is this understanding correct?
> >
> > As for "Sec. 4.2: We extend this to multiple sliding windows, introducing mechanisms for distinguishing and merging concepts across different windows.", do you mean 2 extend 1 from single window to cross window?

---

> > > ### Author Response · Authors · 2024-12-01
> > >
> > > Dear Reviewer,
> > >
> > > Yes, your understanding is correct. We first analyze the representation matrix $\mathbf{Z}$ within a single window to determine the value of k (as detailed in Appendix B). This k is then used along with $\mathbf{Z}$ to apply Ng et al. (2001) for identifying the k concepts within the single window. For multiple windows, we subsequently update the value of k by differentiating the concepts  across windows based on their profiles.
> > >
> > > Regarding “Sec. 4.2: We extend this to multiple sliding windows…”, it indeed refers to extending the single-window mechanism (Step 1) to cross-window operations (Step 2).
> > >
> > > We hope this explanation further clarifies any remaining concerns you may have. If there are still aspects requiring additional detail, we are more than happy to address them. We would also greatly appreciate it if you could reconsider your score based on this clarification, as your feedback is invaluable in helping us refine and enhance the quality and impact of our work.
> > >
> > > Thank you again for your constructive comments!
> > >
> > > Best Regards,
> > >
> > > Authors

---

> > > > ### Comment · Reviewer_kD8a · 2024-12-02
> > > >
> > > > Thanks for your replies!
> > > >
> > > > It is more clear now. I sincerely appreciate your time and efforts in providing these details and explanations.
> > > >
> > > > Table 1 is your main experiment but Drift2Matrix didn't show more advanced than methods such as OneNet.
> > > >
> > > > The explanation is "OneNet, like N-BEATS, achieves good results due to its ensemble-based strengths. However, Drift2Matrix achieves comparable results. Notably, Drift2Matrix is not primarily designed as a forecasting model; rather, it focuses on uncovering concepts and tracking concept drift"
> > > >
> > > > With this focus, there are no datasets that have ground truth of concept drifts to validate if you are uncovering and tracking correctly. How can I know the genuine effectiveness of Drift2Matrix?

---

> ### Author Response · Authors · 2024-12-02
>
> Dear Reviewer,
>
> Thank you for your constructive comments!
>
> >We acknowledge that Drift2Matrix does not achieve the best performance on some datasets. This is because its primary focus lies in adaptive concept identification and tracking rather than optimizing forecasting accuracy. Unlike ensemble-based models like OneNet or FSNet, which aim to enhance forecasting by mitigating concept drift through ensembling or parameter tuning, Drift2Matrix prioritizes understanding and adapting to dynamic concept drift in co-evolving time series. This approach allows Drift2Matrix to delve deeper into the intrinsic structure of time series data. \
> **Our experiments (Sec. 6.2 Q1: Effectiveness & Appendix H.2) specifically evaluate Drift2Matrix’s ability to identify key concepts. Using synthetic and Stock datasets, we validate its capability to uncover meaningful concepts and their transitions, linking them to semantic financial market states. For real-world datasets lacking concept ground-truth, we evaluate the forecasting benefits derived from the discovered concepts.**\
> As noted in Table 1:“While forecasting is not our main focus, we provide a comparison of Drift2Matrix with other models.” Drift2Matrix demonstrates robust performance across diverse datasets, excelling in scenarios characterized by significant concept drift. However, in datasets where variables are relatively independent or where concept drift follows predictable patterns, ensemble models like OneNet achieve better results due to their ensemble-based strengths. **Despite this, Drift2Matrix—a single-model framework—achieves competitive results. Moreover, its deep learning extension, Auto-D2M, achieves state-of-the-art performance across nearly all datasets (Table 1 and Appendix H.3 Table 7).**
>
> > Besides, we have also added **additional evaluations in Appendix H.2, highlighting Drift2Matrix’s ability to identify concepts and track drifts across more datasets**, as well as its predictive capabilities based on the identified concepts (Figures 8-9). Furthermore, in **Appendix H.5**, we validate Drift2Matrix’s concept identification capability on the Motion Segmentation sequences dataset. This dataset comprises sequences divided into three concepts: indoor checkerboard sequences (104 sequences), outdoor traffic sequences (38 sequences), and articulated/nonrigid sequences (13 sequences). It provides ground-truth motion labels and outlier-free feature trajectories (x-, y-coordinates) across frames with moderate noise. The number of feature trajectories per sequence ranges from 39 to 556, and the number of frames ranges from 15 to 100. Under the affine camera model, the trajectories of one motion lie on an affine subspace of dimensions up to three. These results further demonstrate Drift2Matrix’s robust capability to identify and adapt to dynamic concepts in diverse real-world scenarios.\
> **In the revised manuscript, we will include a more comprehensive quantification of Drift2Matrix’s detection capabilities.**(Fig. 11 or or a quick look in [Added Figures](https://anonymous.4open.science/api/repo/Drift2Matrix-main-86B7/file/ICLR_ADD_FIGURE.pdf?v=15a1f953))
>
> >***Remark: To address the limitations in current datasets, particularly those without concept ground-truth, we are developing a new dataset benchmark based on kernel-induced representation. This benchmark, designed to be released to the community, incorporates a novel concept validity metric and expert domain knowledge. By filling the gap in most time series datasets, which often lack concept labels, this new benchmark will enable more rigorous evaluation and further advance research in concept drift adaptation and detection.***
>
> We hope this explanation provides clarity and addresses your concerns. **Additionally, we sincerely appreciate your engagement during the discussion period and the time and effort you’ve invested. This interaction has not only given us the opportunity to clarify our methodology further but has also provided clearer and more insightful suggestions for our revised version.**
>
> Best Regards,
>
> Authors

---

> > ### Comment · Reviewer_kD8a · 2024-12-02
> >
> > Thanks for prompt reply!
> >
> > 1. How to use your tracking result. Assuming that the concepts are tracked and identified, how can I use this information?
> >
> > 2. SyD is generated in Line 1217, correct? All the g#( ) have the comment part of t/100 with an additional combination of one or two cos/sin functions.
> >
> > Let's take g1 as an example, g1(t) = cos (4πt/5) + cos(π(t − 50)) + t/100. Here the t can be 780-steps long, which means t/100 is more than 7, but cos (4πt/5) + cos(π(t − 50)). This means the drift part among g#( ) is reflected by the cos/sin part, which takes a smaller and smaller proportion of g, as t increases.

---

> > > ### Author Response · Authors · 2024-12-02
> > >
> > > Dear Reviewer,
> > >
> > > > **1. How to use the tracking results of concepts**\
> > > In our approach, once concepts are tracked and identified, we can **(1) predict subsequent concepts**, **(2) forecast series values**, as described in Sec 4.2.\
> > > **Extending beyond the scope of this paper:** We have also observed that this adaptive method for discovering and tracking concepts can be highly effective and improve interpretability in various fields, such as regime shifts in atmospheric and oceanic systems in environmental ecology, regime switches (e.g., bear and bull markets) in financial markets, and and understanding phase transitions in physics.
> > >
> > > > **2. On Line 1217 and the generation of SyD**\
> > > We appreciate your attention to detail regarding the inclusion of t/100 . Its purpose is straightforward—it adds a gradually increasing trend to the synthetic data while maintaining the structural integrity of the concepts. Specifically, the trend ensures that as t increases, the overall series value increases slowly over time. For example, between t=0 and t=1000 , the series value differs by approximately 10. However, this slow increase does not impact the structural shape of the concepts, which remain consistent, resembling oscillations around a fixed periodic pattern with a gradually rising overall trend.
> > >
> > > We hope this addresses your concerns.
> > >
> > > Best Regards,
> > >
> > > Authors

---

> > > > ### Comment · Reviewer_kD8a · 2024-12-02
> > > >
> > > > 1. How to use the tracking results of concepts
> > > >
> > > > As you have mentioned, in our approach, once concepts are tracked and identified, we can (1) predict subsequent concepts, (2) forecast series values, as described in Sec 4.2. However, the forecasting effectiveness is not improved compared to existing forecasting methods, such as OneNet. How can I value this point as a significant contribution?
> > > >
> > > > 2. On Line 1217 and the generation of SyD
> > > >
> > > > Your drift is added by switching among different g#( )s, where all the g#( )s have the common parts, t/100. This is to say, the value of t/100 does not consist of any of the drifts while the changes of cos/sin consist of the drift.
> > > >
> > > > The trick here is that the drift of cos/sin is very small compared to the added common item t/100. Why do not we only have cos/sin to consist of g#( ) but have to include the t/100?
> > > >
> > > > t/100 is barely a linear function, which is easy to track or predict. I cannot consider this experiment enough neither appropriate to validate the effectiveness of the proposed method.

---

> > > > > ### Author Response · Authors · 2024-12-03
> > > > >
> > > > > Dear Reviewer,
> > > > >
> > > > > We appreciate your continued engagement and questions. **However, we believe we have already addressed this concern MULTIPLE times.**
> > > > > >First and foremost, our work primarily focuses on **adaptive concept identification and tracking** rather than solely optimizing forecasting accuracy. As detailed in **Sec. 6.2 Q1: Effectiveness & Appendix H.2**, our experiments specifically evaluate Drift2Matrix’s ability to identify key concepts. Using several datasets, including stock time series, we validate its capability to uncover meaningful concepts and their transitions, linking them to semantic financial market states. Additionally, Appendix H.2 includes further evaluations that highlight Drift2Matrix’s ability to identify concepts and track drifts across more datasets, as well as its predictive capabilities based on identified concepts (**Figures 8-9**).\
> > > > > Moreover, in **Appendix H.5**, we validate Drift2Matrix’s concept identification capability on the **Motion Segmentation sequences dataset**. This dataset includes sequences divided into three distinct concepts: indoor checkerboard sequences (104 sequences), outdoor traffic sequences (38 sequences), and articulated/nonrigid sequences (13 sequences). These sequences provide ground-truth motion labels and outlier-free feature trajectories (x-, y-coordinates) across frames, even with moderate noise. Drift2Matrix demonstrates its robust capability in handling these diverse real-world scenarios.\
> > > > > We also include a more comprehensive quantification of Drift2Matrix’s detection capabilities (**Fig. 11**) to further clarify this in the **revision**.\
> > > > > For **real-world datasets** lacking concept ground-truth, we instead evaluate the forecasting benefits derived from the discovered concepts. Despite this limitation, Drift2Matrix—a single-model framework—achieves competitive results when compared to ensemble-based methods like OneNet. Moreover, its deep learning extension, Auto-D2M, achieves **state-of-the-art performance** across nearly all datasets (**Table 1 and Appendix H.3 Table 7**).
> > > > >
> > > > > >We have also already explained that t/100 does not affect the generated concepts. It serves solely to add a slight trend to the dataset. This trend can be removed or modified as needed without impacting the structure of the concepts. For example, if t = [0:10], the values of g1(t) are: [2, -1.799, 1.329, -0.660, 0.231, 0.05, 0.251, -0.621, 1.389, -1.719, 2.1]; If t = [1000:1010], the values of g1(t) are: [12, 8.201, 11.329, 9.339, 10.231, 10.05, 10.251, 9.379, 11.389, 8.281]. Clearly, t/100 only adds an increasing trend, without affecting the local oscillations of g1(t). This is a fundamental operation in practices and can be adjusted or excluded entirely without impacting our methodology.\
> > > > > Beyond synthetic data, we validate Drift2Matrix on **15 real-world datasets as well as 12 additional experiments** detailed in the appendix, covering various aspects such as **Noise and Outlier Robustness, Type I and Type II Error Evaluations, Case Studies on Motion Series, Complexity Analysis and Execution Time Evaluation, and four different Ablation Studies, among others**, to comprehensively demonstrate its capabilities. **Therefore, we respectfully disagree with dismissing our overall contributions** based on a trend parameter in synthetic data, which, as previously explained, is not relevant to our core objective.
> > > > >
> > > > > Best,
> > > > >
> > > > > Authors

---

### Author Response · Authors · 2024-11-21

## Global Response
---
First, we would like to thank all the reviewers' valuable and insightful suggestions! We are delighted that reviewers find our paper has **reasonable motivation** (kD8a), **well presented and structured** (kD8a, 17sK，oERz），**easily combined with DL models** (oERz, ERvr), **sufficient theoretical analysis** (17sK, oERz, 53aH),  **comprehensive experimental evaluation** (kD8a, 17sK, oERz, ERvr) and **open-source** (ERvr).

In response to the feedback, we answer all questions and provide new experimental results enhancing both the clarity of the paper and the experimental section. We would be happy to address all these aspects by stating the element of our responses in a revised paper-ready version and in the appendix to improve the paper. Below, we summarize the key enhancements made in the revised manuscript:

> 1. **Highlight Drift2Matrix Mechanism (kD8a):** we emphasized how Drift2Matrix utilizes kernel-induced representation to model the time series ecosystem, adaptively identifying and tracking concepts while predicting those not observed in a single series.
> 2. **Expanded Problem Definition and Learning Mode (17sK):** we provided a more detailed mathematical definition of concept identification and drift in co-evolving time series, along with an explanation of our learning mode, highlighting the challenges and novelty of our approach.
> 3. **Table 1 Explanation (17sK, oERz):** we highlighted that Drift2Matrix focuses on adaptive concept identification over forecasting accuracy, acknowledged OneNet’s strong performance as an ensemble model, and noted that Drift2Matrix achieves comparable results as a single model.
> 4. **Expanded Benchmark Methods (17sK):** we included additional benchmark methods, comparing Drift2Matrix and Auto-D2M with 17 methods (10 published after 2021 and 15 after 2019) across 15 diverse datasets. Full results are provided in ***Appendix H.3, Table 7***.
> 5. **Parameter Sensitivity and Complexity Analysis (17sK, ERvr):** we conducted a comprehensive parameter sensitivity analysis, examining parameters $\alpha, \gamma$ (Eq. (3)) and $\beta$ (Eq. (15)), with detailed insights integrated into Appendix F (**Fig. 6 and Fig. 17**). Additionally, we provide a thorough complexity analysis, highlighting computational efficiency and optimization strategies (***Appendix H.7 and Fig. 15***).
> 6.  **Appendix and Ablation Studies (oERz):** We highlighted the connection between the appendix and the main text, emphasizing how kernel-induced representation forms the basis for domain-agnostic window size selection and concept estimation. Additionally, we included ablation studies analyzing the impact of the number of co-evolving time series on performance (***Fig. 16***).
> 7. **Concept Drift Detection and Adaptive Kernel Selection (ERvr):** We conducted additional experiments on Concept Drift Detection, including visualizations on the ETTh1 dataset (***Fig. 11***), to further evaluate the model's capabilities. Moreover, we performed ablation experiments on the non-trivial selection of kernel functions (***Table 8***), exploring multi-kernel learning from a technical perspective and a data-driven kernel learning strategy from a paradigm perspective.

For convenience, alongside our revisions, we have provided reviewers with an anonymous quick reference to the added figures and updates at [Added Figures](https://anonymous.4open.science/api/repo/Drift2Matrix-main-86B7/file/ICLR_ADD_FIGURE.pdf?v=15a1f953).

**Note:** Regarding occasional markdown formula rendering issues (e.g., equations not displaying), **OpenReview suggests refreshing the page**, which resolves the problem (we have tested this as well).

---

### Meta-Review · Area_Chair_7fdw · 2024-12-19

**Metareview:**

Based on the reviews, I recommend not accepting the paper for publication in its current form. The submission has received four reviews, three of which recommend acceptance but with rather short reviews. One reviewer, a highly confident domain expert, recommends rejection. The reviewer has spent a lot of time in the reviewing process, which is evidenced by their 14 Official Comments in addition to the detailed review. The review raises significant concerns about key aspects of the work. These include the lack of limited novelty, insufficient evidence on the effectiveness of the method, and shortcomings in the empirical evaluation.

**Additional Comments On Reviewer Discussion:**

- **Reviewer kD8a** expressed concerns about the lack of novelty, technical details, theoretical rigor, and experimental flaws. The authors responded extensively but failed to fully address the issues, particularly regarding the method's correctness and empirical support. Reviewer kD8a maintained their rejection recommendation.

- **Reviewer 17sK** requested clarifications on the problem definition and other details, which the authors addressed thoroughly. Reviewer 17sK remained supportive and recommended acceptance.

- **Reviewer oERz** highlighted the absence of statistical significance measures and insufficient validation. The authors addressed most of these concerns, but Reviewer oERz still expressed reservations, rating the paper marginally above the acceptance threshold.

- **Reviewer ERvr** raised concerns about the lack of complexity analysis and kernel selection, which could limit the method's applicability. The authors responded, and Reviewer ERvr was mostly satisfied, increasing their score.

The key concern was Reviewer kD8a's thorough critique, which pointed out fundamental issues in novelty, empirical validation, and method correctness. Despite responses from the authors, these concerns were not fully resolved, leading to the given recommendation.

---

### Decision · Program_Chairs · 2025-01-22

Reject